



# Isotopic fractionation corrections for the radiocarbon composition of $CO_2$ in the soil gas environment must include diffusion and mixing

Jocelyn E. Egan[1], David R. Bowling[2], David A. Risk[3]

[1]Department of Earth Sciences, Dalhousie University, Halifax, Nova Scotia, B3H 4R2, Canada
[2]School of Biological Sciences, University of Utah, Salt Lake City, Utah, 84112, USA
[3]Department of Earth Sciences, St. Francis Xavier University, Antigonish, Nova Scotia, B2G 2W5, Canada

*Correspondence to*: Jocelyn E. Egan (jocelyn.egan@dal.ca)

**Abstract.** Earth system scientists working with radiocarbon in organic samples use a stable carbon isotope ($\delta^{13}C$) correction to account for mass-dependent fractionation caused primarily by photosynthesis. Although researchers apply this correction

routinely, it has not been evaluated for the soil gas environment, where both diffusive gas transport and diffusive mixing are important. Towards this end we applied an analytical soil gas transport model across a range of soil diffusivities and biological $CO_2$ production rates, allowing us to control the radiocarbon ($\Delta^{14}C$) and stable isotope ($\delta^{13}C$) compositions of modeled soil $CO_2$ production and atmospheric $CO_2$. This approach allowed us to assess the bias that results from using the conventional correction method for estimating $\Delta^{14}C$ of soil production. We found that the conventional correction is

inappropriate for interpreting the radio-isotopic composition of $CO_2$ from biological production, because it does not account for diffusion and diffusive mixing. The resultant $\Delta^{14}C$ bias associated with the traditional correction is highest (up to 150 ‰) in soils with low biological production and/or high soil diffusion rates. We propose a new solution for radiocarbon applications in the soil gas environment that fully accounts for diffusion and diffusive mixing.

## 1 Introduction

Understanding the age of soil-respired carbon is important for process-based carbon cycle studies in the face of climate change. By knowing the age of respired carbon, we can make predictions about the rate at which carbon stocks will be transformed under current and future climates. In the case of Arctic environments underlain by permafrost, this is especially important as "ancient" carbon sources previously stored may become available to decomposers as the permafrost thaws.

Radiocarbon allows us to measure the age of *soil-respired $CO_2$* ($CO_2$ diffusing from the soil surface to the atmosphere, aka

soil flux as in Cerling et al., 1991), but the traditional calculation and reporting convention for radiocarbon was not established for soil gas-phase sampling, but rather for solid (organic matter) sample analysis. The validity of this convention has never been explicitly tested for soil-respired $CO_2$.

The traditional radiocarbon reporting convention (Stuiver and Polach, 1977), uses a mass-dependent correction based on the



isotopic composition of wood. Its purpose is to correct for biochemical fractionation against the radiocarbon isotopologue ($^{14}CO_2$) abundance during photosynthesis, which is assumed to be twice as strong as for $^{13}CO_2$ based on their respective departures in molecular mass from $^{12}CO_2$. The classical reference describing these conventional calculations is Stuiver and Polach (1977).

In the soil gas environment, researchers have different implicit expectations for fractionation processes. They generally assume that $^{14}C$ of $CO_2$ is *not* biochemically fractionated in the gas phase, between the points of $CO_2$ *production* (biological production of $CO_2$ by soil organisms and roots) and measurement (subsurface or flux chamber samples). This assumption is reasonable based on the short residence time of $CO_2$ (minutes to days) in the soil profile before emission to the atmosphere. However, soil gas isotopic signatures depart in predictable ways from the signature of production because of physical

fractionation. It has been recognized for decades that $\delta^{13}C$ of $CO_2$ at any point in the soil profile will never equal the isotopic signature of production, because of transport fractionations that alter produced $CO_2$ before it is measured (Cerling et al., 1991). This theory extends readily to $^{14}C$. Overall, soil researchers have two implicit expectations, which is that physical fractionations are present, and that biochemical fractionations are absent. These expectations seem clearly out of step with the classical reporting conventions for $^{14}C$ that compensates only for biochemical fractionation. To date, no study has

examined radiocarbon reporting conventions with these implicit expectations in mind. We believe the traditional radiocarbon convention should be re-evaluated for soil gas studies.

The principal objective of this study is to determine a way to properly interpret measurements of radiocarbon from the soil gas environment. Using theory and physical modelling of soil gas transport we assess traditional techniques, propose and validate an alternative approach for specific use cases, and consider uncertainties that would result from the misapplication

of traditional approaches.

**2 Theory**

To understand why the Stuiver and Polach (1977) convention may be a poor fit for soil gas studies, we can look at our current understanding of the stable isotopic composition of *soil $CO_2$* (pore space $CO_2$, mole fraction with respect to dry air). We use delta notation to present the stable isotopic composition of $CO_2$:

$$\delta^{13}C = \left(\frac{R_s}{R_{VPDB}} - 1\right) 1000, \tag{1}$$

where $\delta^{13}C$ is the isotopic composition in ‰ (see Table 1 for a full list of abbreviations), $R_s$ is the $^{13}C/^{12}C$ ratio of the sample, and $R_{VPDB}$ is the $^{13}C/^{12}C$ ratio of the international standard, Vienna Pee Dee Belemnite.

The isotopic composition of soil $CO_2$ is different from that of soil-respired $CO_2$ (Cerling et al., 1991). The change in $\delta^{13}C$ of

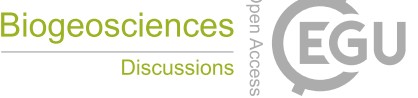



soil $CO_2$ with depth is influenced by 1) mixing of atmospheric and biological (or biogeochemical) sources of isotopically-distinct $CO_2$, which may occur via diffusion (no bulk gas flow) or advection (bulk gas flow) and 2) kinetic fractionation by diffusion. The effect of these is illustrated in Fig. 1. In panel (a) two depth profiles of $\delta^{13}C$ of $CO_2$ were modelled as a steady-state environment are shown (the model will be described later). The profiles differ only in soil diffusivity; all other characteristics were held constant, including rates of production, and $\delta^{13}C$ of $CO_2$ in the atmosphere (-8 ‰; circle) and biological production (-25 ‰; square with dashed line). In the resultant depth profile with higher soil diffusivity in panel (a), the $\delta^{13}C$ of soil $CO_2$ ranges from -8 to -14.2 ‰. In the depth profile representing a soil with lower diffusivity, the $\delta^{13}C$ of soil $CO_2$ ranges from -8 to -20.5 ‰. We stress again these two isotopic depth profiles differ only due to differences in transport as a result of their varying soil diffusivities. In the depth profile with lower soil diffusivity, atmospheric $CO_2$ does not penetrate downwards as readily, so the profile shape is much steeper near the soil-atmosphere boundary, and is more reflective of the production source composition, -25 ‰, at depth. In the depth profile with higher soil diffusivity, atmospheric air of -8 ‰ more readily mixes from the surface downward by diffusion, so the near surface isotopic composition will be more reflective of the atmosphere due to mixing of these end-members near the soil surface.

Importantly, the soil $CO_2$ never equals the $\delta^{13}C$ of production (-25 ‰) at any depth, in either profile in Fig. 1(a). It is not possible to directly measure $\delta^{13}C$ of production *in situ*, because diffusion and mixing alter the character of $CO_2$ immediately after its production. From the site of production in the soil, $^{12}CO_2$ diffuses somewhat faster through the soil than $^{13}CO_2$, because the former has lower mass. This diffusive difference leads to isotopic fractionation, and results in depth profiles of $\delta^{13}C$ of soil $CO_2$ that are isotopically enriched (less negative) as compared to the source of production. Work by Cerling (1984) and later by Cerling et al. (1991), demonstrated that the mass differences between the two isotopologues led to a difference in diffusion rate of each in air, amounting to a fractionation of 4.4 ‰ (note that this applies only to binary diffusion of $CO_2$ in air and will differ if $CO_2$ diffuses in other gases). As a result, the $\delta^{13}C$ of soil $CO_2$ measured at any depth will be enriched by a minimum of 4.4 ‰ relative to the biological production $CO_2$ source. However, the $\delta^{13}C$ of soil-respired $CO_2$ can be considerably more enriched than 4.4 ‰ relative to production due to mixing with the atmosphere as shown in Fig. 1(a).

A convenient theoretical formulation for correcting $\delta^{13}C$ for both diffusion fractionation and diffusive mixing was introduced by Davidson (1995), following on the work of Cerling (1984) and Cerling et al. (1991). This approach allows one to combine measurements of $CO_2$ and its isotopic composition within the soil and the air above it, to infer the isotopic composition of $CO_2$ produced in the soil. This only applies when transport within the soil is purely by diffusion (no bulk air movement). The Davidson (Davidson, 1995) solution uses the difference between the diffusion coefficients for $^{12}C$ and $^{13}C$ as follows:

$$\delta_J^{13} = \frac{C_s(\delta_s^{13}-4.4)-C_a(\delta_a^{13}-4.4)}{1.0044(C_s-C_a)}, \tag{2}$$



where $\delta_j^{13}$ is the $\delta^{13}$C composition of $CO_2$ from soil production (biological respiration within the soil), $C_s$ and $\delta_s^{13}$ are the mole fraction and isotopic composition of soil $CO_2$, and $C_a$ and $\delta_a^{13}$ are the mole fraction and isotopic composition of $CO_2$ in the air just above the soil. In Fig. 2(a) the mole fraction and isotopic composition of soil $CO_2$ at a 20 cm depth and of the air just above the soil was "sampled" from model-generated soil depth profiles and the (unrounded) values were used to

calculate the isotopic composition of production using Davidson's equation ($C_s$ = 9020 ppm, $\delta_s^{13}$ = -20.1745 ‰, $C_a$ = 380 ppm and $\delta_a^{13}$ = -8 ‰). The resulting $\delta_j^{13}$ (e.g. Eq. (2)) at this depth equals the true isotopic composition of production  (see inset box, 2(a)). However, because the Davidson approach accounts for diffusion and mixing, at any given soil depth, not just 20 cm, the modelled values of $C_s$ and $\delta_s^{13}$ in Fig. 1(a) and 2(a) will always yield (via Eq. (2)) the true isotopic composition of production, $\delta_j^{13}$ = -25 ‰ (dashed line). If $\delta^{13}$C of soil $CO_2$ were (erroneously) interpreted to represent the

$\delta^{13}$C of soil-respired $CO_2$, the error could be as large as the absolute value of ($\delta_a$- $\delta_j^{13}$) – 4.4 ‰. In Fig. 2(a), the error in interpreting $\delta^{13}$C of soil $CO_2$ at depth (~ 20 cm and ~ -20 ‰) as a value of production would be ~5 ‰, and the error increases in magnitude at shallower depths.  This Davidson (1995) $\delta_j^{13}$ approach has been shown to be robust when applied to field data from natural soils (Bowling et al., 2015; Breecker et al., 2012b; Liang et al., 2016).

While $^{14}$C is a radioactive isotope and thus decays with time, the half-life is sufficiently long so that $^{14}CO_2$ behaves similarly

to stable isotopes on the timescales at which diffusion occurs in a soil gas system.  Graham's law of effusion (rate of diffusion ~ $1/\sqrt{mass}$) provides us with an expected difference in diffusion rate of isotopologues.  In this way, $\delta^{13}$C diffusive fractionation theory can be applied to the radiocarbon isotopic composition, $\delta^{14}$C, so long as we account for the mass difference. The larger mass of $^{14}$C means that the diffusion fractionation factor is calculated to be 8.8 ‰ based on the atomic masses of $^{14}CO_2$, $^{12}CO_2$ and of bulk air (Southon, 2011).

We can show that $^{14}CO_2$ distribution in soils will be like that of $^{13}CO_2$, if we model its distribution through depth in the same synthetic soil gas environment. In Fig. 1(b) we present a modelled soil environment with defined atmospheric and production source $CO_2$ isotopic composition boundary conditions for $\delta^{14}$C, the $^{14}$C equivalent to $\delta^{13}$C:

$$\delta^{14}C = \left(\frac{A_s}{A_{abs}} - 1\right) 1000, \qquad\qquad\qquad (3)$$

where $\delta^{14}$C is the isotopic composition in ‰, $A_s$ is the measured activity of the sample, and $A_{abs}$ is the activity of the oxalic

acid standard (both unitless). As in Fig. 1(a), in panel (b) the profile with lower soil diffusivity, the downward penetration of atmospheric $CO_2$ into the soil profile is reduced, and as a consequence the isotopic depth profile more closely reflects (but does not equal) the composition of production (-200 ‰; dashed line). When the diffusion rate is high and transport is rapid, the atmospheric source is more readily able to penetrate the profile and mix with the production source. In both profiles, the measured value of soil $CO_2$ at a given depth will not equal the isotopic production value of -200 ‰, because of diffusion and

diffusive mixing. Similar profiles of $\delta^{14}$C of soil $CO_2$ with depth, highlighting the diffusive effects, have been presented by



Wang et al. (1994).

Since $\delta^{14}$C transport of soil $CO_2$ is like that of $\delta^{13}$C, it follows that we should apply corrections for $\delta^{14}$C like those in Eq. (2) in order to calculate the isotopic composition of production. The $\delta^{14}$C reformulation of Davidson's $\delta_J^{13}$ equation is as follows:

$$\delta_J^{14} = \frac{C_s(\delta_s^{14} - 8.8) - C_a(\delta_a^{14} - 8.8)}{1.0088(C_s - C_a)}, \tag{4}$$

where $\delta_J^{14}$ is the $\delta^{14}$C composition of soil production, $C_s$ and $\delta_s^{14}$ are the mole fraction and $\delta^{14}$C composition of the soil $CO_2$, and $C_a$ and $\delta_a^{14}$ are the mole fraction and $\delta^{14}$C composition of $CO_2$ in the air just above the soil. This Davidson reformulation for $\delta^{14}$C, $\delta_J^{14}$, was applied to a model-generated profile of soil $\delta^{14}$C at a 20 cm depth in Fig. 2(b), like in panel (a) for $\delta^{13}$C ($C_s$ = 9020 ppm, $\delta_s^{14}$ = -178.9879 ‰, $C_a$ = 380 ppm and $\delta_a^{14}$ = 138 ‰; see inset box, Fig. 2(b)). As was the case for $\delta^{13}$C in Fig. 2(a), the modelled values of $C_s$ and $\delta_s^{14}$ at any depth will yield the true isotopic composition of production, -200 ‰ (dashed line), because this approach accounts for diffusion and mixing.

The traditional approach for interpreting the $^{14}$C composition of soil $CO_2$ and soil-respired $CO_2$ (e.g., Trumbore, 2000) differs from the $\delta^{14}$C example above, because a $\delta^{13}$C correction is applied to account for mass-dependent isotopic fractionation of biochemical origin (Stuiver and Polach, 1977), ultimately converting $\delta^{14}$C to a variant called $\Delta^{14}$C. The derivation of the mass-dependent correction is provided in Stuiver and Robinson (1974), where observations are normalized to an arbitrary baseline value of -25 ‰ for $\delta^{13}$C (a value for terrestrial wood), and the $^{13}$C fractionation factors are squared to account for the $^{14}$C/$^{12}$C fractionation factor as follows:

$$
\begin{aligned}
A_{SN} &= A_s \left[ \frac{R_s(-25)}{R_s} \right]^2, \\
&= A_s \frac{\left[ \left(1 - \frac{25}{1000}\right) * R_{VPDB} \right]^2}{\left[ \left(1 + \frac{\delta^{13}C}{1000}\right) * R_{VPDB} \right]^2}, \\
&= A_s \frac{\left[ \left(1 - \frac{25}{1000}\right) \right]^2}{\left[ \left(1 + \frac{\delta^{13}C}{1000}\right) \right]^2}, \tag{5}
\end{aligned}
$$

where $A_{SN}$ is the normalized sample activity, $A_s$ is the sample activity, and $\delta^{13}$C is the isotopic composition of the sample (soil $CO_2$ in our case). As explained in Stuiver and Robinson (1974), the 0.975 term sometimes used in forms of $A_{SN}$ is equivalent to $\left(1 - \frac{25}{1000}\right)$, which we will retain for clarity. The equation for $\Delta^{14}$C, the $\delta^{13}$C corrected variant of $\delta^{14}$C, can then be created from Eq. (5) by substituting in delta notation for $\Delta^{14}$C of $\Delta^{14}$C = ($A_{SN}/A_{abs}$ -1)*1000 as follows Stuiver and Robinson (1974):



$$\Delta^{14}C = \left[ \left( 1 + \frac{\delta^{14}C}{1000} \right) \frac{\left(1 - \frac{25}{1000}\right)^2}{\left(1 + \frac{\delta^{13}C}{1000}\right)^2} - 1 \right] 1000. \tag{6}$$

Combining Eqs. (3) and (6):

$$\Delta^{14}C_{old} = \left[ \left( \frac{A_S}{A_{abs}} \right) \frac{\left(1 - \frac{25}{1000}\right)^2}{\left(1 + \frac{\delta^{13}C}{1000}\right)^2} - 1 \right] 1000. \tag{7}$$

For more information on the derivation of Eqs. (6) and (7) see Stuiver and Robinson (1974), page 88. In Eq. (7) we have
added the subscript "old" to highlight that this is the traditional mass-dependent correction - we will introduce a "new"
method with Eq. (15). The terms on the left-hand side of Eqs. (6) and (7) are identical. Note that $A_{abs}$ in our notation is
equivalent to $A_O$ in Stuiver and Robinson (1974).

This traditional approach uses $\delta^{13}C$ as an input parameter to make a mass-dependent correction to obtain $\Delta^{14}C$, but the
profiles of $\delta^{13}C$ and $\delta^{14}C$ of soil $CO_2$ (Fig. 1) highlight that both vary within the soil because of diffusion and mixing. This
makes it unclear what form of $\delta^{13}C$ should actually be used in the correction in the soil gas environment ($\delta^{13}C$ of the soil $CO_2$
is measured, but $\delta^{13}C$ of biological production is not). When $\Delta^{14}C_{old}$ is modelled through depth like $\delta^{13}C$ and $\delta^{14}C$ in Figs. 1
and 2 it also varies with depth as shown in Fig. 2(c). However, using a $\Delta^{14}C$ variant of Davidson's $\delta_J$ (as for $\delta^{14}C$ in Fig.
2(b)) at the same 20 cm depth does not correctly reproduce the specified model value for the $\Delta^{14}C$ of production of -200 ‰,
like it did for $\delta^{13}C$ and $\delta^{14}C$ ($C_s$ = 9020 ppm, $\Delta_s$= -187.0547 ‰, $C_a$ = 380 ppm and $\Delta_a$ = 100 ‰; see inset box, 2(c)).

For soil studies, there is clear conflict between conventions for expressing radiocarbon-$CO_2$, and our mathematical
understanding of soil gas transport. Although theory presented above suggests that the Stuiver and Polach (1977) approach is
inappropriate, these examples do not consider the full possible range of natural soil conditions (diffusivities and production
rates). For a more comprehensive exploration, and to test alternative proposed methodologies testing across a range of soil
parameters, we used a model approach to simulate realistic natural soil depth profiles of $^{12}CO_2$, $^{13}CO_2$, and $^{14}CO_2$. Once the
simulated profiles were generated, we "sampled" from them as if we were making field measurements, and determined the
radiocarbon isotopic composition of soil $CO_2$. Secondly, we adapted the traditional convention using Davidson's (1995)
theory and tested this new convention in the same way as the traditional one, by simulating soil profiles and "sampling"
them to extract the isotopic composition at depth. We then used actual field observations, where natural soil depth profiles
were sampled, to test the discrepancy between old and new formulations of $\Delta^{14}C$.





### 3 Methods

#### 3.1 Testing the Traditional Approach

We used an analytical gas transport model to simulate a range of natural soil profiles of $^{12}CO_2$, $^{13}CO_2$ and $^{14}CO_2$. The model is based on Fick's second law of diffusion:

$$\theta \frac{\partial Conc}{\partial t} = \frac{\partial}{\partial z}\left(D\,(z,t)\frac{\partial Conc}{\partial z}\right) + P(z,t)\,,$$ (8)

where $\theta$ is the soil air-filled pore space, $Conc$ is the concentration, $t$ is time, $D(z,t)$ is the soil gas diffusion function, and $P(z,t)$ is the biological production function, with the latter two dependent on both depth $z$ and time $t$.

The model was run in steady-state:

$$\frac{\partial Conc}{\partial t} = 0,$$ (9)

and both diffusion and production rates were constant with depth:

$$D(z) = D,$$ (10)
$$P(z) = P.$$ (11)

The following boundary conditions were used:

$$C(z = 0) = Conc_{atm},$$ (12)

$$\frac{\partial C}{\partial z}\Big|_{z=L} = 0,$$ (13)

where $Conc_{atm}$ is the concentration of $CO_2$ in air just above the soil and $L$ is the model lower spatial boundary, the point below which no production or diffusion occurs. Eq. (8) is solved analytically to yield the following equation:

$$Conc(z) = \frac{P/L}{D}\left(L \times z - \frac{z^2}{z}\right) + Conc_{atm}.$$ (14)

In the model, isotopologues of $CO_2$ are treated as independent gases, with their own specific concentration gradients and
diffusion rates (Cerling et al., 1991; Nickerson and Risk, 2009b; Risk and Kellman, 2008). We assume total $CO_2$ to be $^{12}CO_2$ because of its high abundance. The error associated with this assumption is less than 0.01% (Amundson et al., 1998). Eq. (14) is thus applied for $^{13}CO_2$ and $^{14}CO_2$. For the full derivation see Nickerson et al. (2014) Section 2.3.

The analytical gas transport model was applied across a range of soil diffusivity ($1 \times 10^{-7}$, $1 \times 10^{-6}$ and $1 \times 10^{-5}$ m$^2$ s$^{-1}$) and soil
production rates (0.5, 1, 2, and 4 µmol $CO_2$ m$^{-3}$ s$^{-1}$), with a depth resolution of 0.025 m. In addition it used $\Delta^{14}$C and $\delta^{13}$C compositions of soil biogeochemical production and atmospheric $CO_2$, all representing realistic conditions found in nature. The other model boundary conditions were as follows: $L = 0.8$ m, $z = 0.025$ m, $Conc_{atm} = 15833$ µmol m$^{-3}$ (~380 ppm), $\delta^{13}$C





of atmospheric $CO_2$ ($\delta_a$) = -8 ‰, $\delta^{13}C$ of production = - 30 ‰, $\Delta^{14}C$ of atmospheric $CO_2$ ($\Delta_a$) = 100 ‰, and $\Delta^{14}C$ of production = -200 ‰. The output of the model under these applied conditions were profiles of $^{12}CO_2$, $^{13}CO_2$, and $^{14}CO_2$ for each depth (z) down to the bottom boundary (L).

### 3.2 Testing the Traditional and Alternative Conventions

The model-generated soil pore space values of $^{12}CO_2$, $^{13}CO_2$, and $^{14}CO_2$ at each depth were then used to produce soil depth profiles of $\delta^{13}C$, $\delta^{14}C$, and $\Delta^{14}C$ of $CO_2$ as demonstrated earlier in Figs. 1 and 2. To simulate the traditional convention for radiocarbon (referred to as $\Delta^{14}C_{old}$ here), we input the soil pore space values into Eq. (7) to calculate the $\Delta^{14}C$ depth profiles.

Our proposed new approach is based on Davidson's (1995) theory. Rather than using the $\delta^{13}C$ soil pore space as a mass-
dependent correction in Fig. 2, we suggest instead using the value $\delta_j^{13}$ (Eq. (2)), the biological production of $\delta^{13}C$, in its place in the denominator of Eq. (7) as follows:

$$\Delta^{14}C_{new} = \left[ \left( \frac{A_S}{A_{abs}} \right) \frac{\left(1 - \frac{25}{1000}\right)^2}{\left(1 + \frac{\delta_j^{13}}{1000}\right)^2} - 1 \right] 1000. \tag{15}$$

The model-generated soil pore space isotope values were input into Eq. (15) to produce depth profiles of $\Delta^{14}C_{new}$. Values of $\Delta^{14}C_{new}$ through depth represent transport-fractionation-corrected soil $CO_2$ values of radiocarbon. Then to calculate the
radiocarbon composition of production, $\Delta_j^{14}$, the $\Delta^{14}C$ composition of the soil $CO_2$, $\Delta^{14}C_{new}$, can be placed into our adaption of Davidson (1995) for $^{14}C$ (Eq. (4)) as follows:

$$\Delta_j^{14} = \frac{C_s(\Delta^{14}C_{new} - 8.8) - C_a(\Delta_a^{14} - 8.8)}{1.0088(C_s - C_a)}, \tag{16}$$

where $\Delta_j^{14}$ is the $\Delta^{14}C$ composition of soil production, $C_s$ and $\Delta^{14}C_{new}$ are the mole fraction and $\Delta^{14}C$ composition of the soil $CO_2$, and $C_a$ and $\Delta_a^{14}$ are the mole fraction and $\Delta^{14}C$ composition of $CO_2$ in the air just above the soil.

### 3.3 Field Soil Profiles

To compare the model results to field results, we also sampled soil depth profiles in the field. We collected samples of soil $CO_2$ at 3 depths from the soil profile and from the air just above the soil, from 3 sites in Weyburn, Saskatchewan, for both stable and radio-isotope analysis. These sites were located in agricultural fields, with soils composed of mostly clay, with shallow organic layers.

To collect samples, we installed horizontal polyvinyl chloride (PVC) soil gas wells, 60 cm long, with 3-ply Gore-tex$^{TM}$ membranes covering 2 cm holes drilled every 10 cm along the length of PVC, at soil depths of 5 cm, 10 cm, 20 cm, 50 cm





and 75 cm. Details on the soil gas well design and sampling disturbance prevention are described in detail in Risk et al. (2013). We withdrew a minimum of two samples (one shallow and one deep) and one air sample at each of the three sites 6 times (approximately every 2 months) between August 2011 and May 2012 as described in Risk et al. (2013), by connecting a $N_2$-purged and evacuated 1 L stainless steel canister (Lab Commerce) to the well and allowing it to equilibrate for 15 mins.

The samples were sent to the University of Florida for cryogenic purification and then to the University of California Irvine Keck Accelerator Mass Spectrometer (AMS) facility to be graphitized and analyzed for $\Delta^{14}C$.

As described in Risk et al. (2013), on each visit triplicate samples were also collected in 10 ml $N_2$-purged and evacuated Exetainers vials (Labco, UK) at each depth as well as from the air above the soil. The triplicate samples were analyzed for

bulk $CO_2$ and $\delta^{13}C$ within 2 weeks of sampling using a GV Isoprime CF-IRMS and Multiflow gas bench (Isoprime, UK).

### 3.4 Interpreting Soil Profiles

Our primary goal was to correctly extract the isotopic composition of $CO_2$, produced by biological production. We applied two methods to extract this signal from the model-generated soil $CO_2$ profiles as well as soil $CO_2$ profiles from the field:

Method 1 follows the traditional correction, where Eq. (7) was used to calculate $\Delta^{14}C_{old}$, and, following convention, we

interpret this value as the radiocarbon composition of $CO_2$ from biological production.

Method 2, our proposed solution, instead determines the radiocarbon composition of $CO_2$ from biological production in two steps, as described in Section 3.2. First, the transport fractionation corrected radiocarbon composition of soil $CO_2$, is calculated using Eq. (15) ($\Delta^{14}C_{new}$). Then this properly interpreted soil $CO_2$ value can be used in Eq. (16) to calculate the

radiocarbon composition of production, $\Delta_J^{14}$.

### 4 Results

#### 4.1 Testing the Traditional Approach

In Fig. 3 we present the results from model scenarios that simulated natural soil mixing profiles between sources that are intended to represent a realistic range of values as a function of depth and soil properties. All the scenarios had the same

prescribed $\Delta^{14}C$ source values ($CO_2$ in the air just above the soil and soil biological production), and only soil diffusivities and production rates differed. Like the $\delta^{13}C$ and $\delta^{14}C$ soil $CO_2$ depth profiles in Fig. 1, the $\Delta^{14}C$ soil $CO_2$ profiles varied as a function of soil diffusivities and production rates. In the model scenarios in Fig. 3(a), the depth profiles with higher soil diffusivities had more enriched $\Delta^{14}C$ soil $CO_2$ (50-100 ‰ more enriched, depending on depth) than the scenarios with lower soil diffusion rates. In Fig. 3(b), the depth profiles with lower production rates had more enriched $\Delta^{14}C$ of soil $CO_2$ (10-20 ‰

more enriched) than scenarios with higher production rates. In the profiles with steeper diffusive gradients near the surface, soil $CO_2$ through depth was more representative of the production source, as we would expect from theory.





In Fig. 3 two depth profiles are plotted for each model iteration, one where the model output was used to calculate the conventional representation of radiocarbon, $\Delta^{14}C_{old}$ (Eq. (7); solid line) and the second where the output was used to calculate $\Delta^{14}C_{new}$, our proposed convention for radiocarbon soil $CO_2$ (dashed line). All of the $\Delta^{14}C_{old}$ depth profiles in Fig. 3

had slightly (~ 10 ‰) more depleted $\Delta^{14}C$ soil $CO_2$ values than the $\Delta^{14}C_{new}$ depth profiles. When the radiocarbon composition of $CO_2$ from biological production, $\Delta_J^{14}$, was calculated using $\Delta^{14}C_{new}$ (Eq. (16); Method 2), the result was equal to the radiocarbon value of production input into the model (-200 ‰) through the entire soil $CO_2$ depth profile under all soil scenarios. In contrast, when Method 1 was applied, we would (incorrectly) interpret $\Delta^{14}C_{old}$ soil $CO_2$ to be the radiocarbon composition of $CO_2$ from biological production, the $\Delta^{14}C_{old}$ values through depth never equalled the $\Delta^{14}C$ value of soil

production input into the model (-200 ‰).

### 4.2 Field Experiment

The radiocarbon values for two of the sampling dates where soil $CO_2$ samples were collected from a field site in Weyburn, Saskatchewan, Canada, are presented in Fig. 4. The traditional $\Delta^{14}C$ convention, $\Delta^{14}C_{old}$, reported to us by the laboratory

where the samples were analysed are plotted with solid lines. Using Method 1, we would (incorrectly) interpret these values as the values of biological production, but instead they represent soil $CO_2$ values. The dotted lines are the radiocarbon compositions of production ($\Delta_J^{14}$), where the reported radiocarbon values were back-corrected using our new approach, $\Delta^{14}C_{new}$, and then input into Eq. (16) (Method 2). When considering age, the radiocarbon compositions of production ($\Delta_J^{14}$), calculated using Method 2, were older in the springtime than those calculated using Method 1, despite the fact that Method 2

values lie on either sides of Method 1 values (more deplete at the surface and more enriched at depth). Although the values are more enriched at depth, in terms of age, they are still older C. This is because both Method 1 and 2 values fall within the post 1950s $^{14}C$ bomb spike period (Trumbore, 2000), but the more enriched values are still slightly older in terms of age.

### 5 Discussion

### 5.1 Traditional Convention Error

Diffusion- and production-controlled soil gas profiles, similar to those presented in Fig. 3, have been shown in other papers for $\delta^{13}C$ (e.g., Bowling et al., 2015; Breecker et al., 2012a; Cerling, 1984; Cerling et al., 1991; Davidson, 1995; Nickerson and Risk, 2009b). Given that we based our gas transport corrected $\Delta^{14}C$ calculation on the same equations, this is what we expected, where values of $\Delta^{14}C$ of soil $CO_2$ differed in the soil profile (shown in Fig. 3) were intermediate between the two mixing sources ($CO_2$ in the air just above the soil and soil biological production). These effects of diffusion and mixing

demonstrate that Method 1 (which assumes $\Delta^{14}C_{old}$ of soil $CO_2$ = $\Delta^{14}C$ of soil production) is inappropriate for the soil gas application because it does not describe the radiocarbon composition of $CO_2$ as produced, and before alteration by gas transport processes. In contrast, Method 2 (where the $\Delta^{14}C$ isotopic signature of production, $\Delta_J^{14}$, is calculated from $\Delta^{14}C_{new}$)





does not have any error because this method is able to calculate the true isotopic composition of soil production along the entire soil profile diffusive mixing gradient, no matter how steep or shallow the gradient.

The degree of error for Method 1 will depend on a given soil environment, where soil diffusivity and production rates will either amplify or decrease the error. When we "sampled" model-produced depth profiles using Method 1, we were unable to correctly extract the specified isotopic compositions of biological production that were input in the model, under any scenario. Method 1 error can be quantified as the absolute difference between the model's prescribed isotopic value of production, and $\Delta^{14}C_{old}$. In the specific scenarios shown in Fig. 3, the smallest Method 1 error (10 ‰) was in model scenarios with high production rates or low diffusivities, and the largest error (100 ‰) was in scenarios with higher soil diffusivities. Based on the rate of decline of atmospheric bomb $^{14}C$ of 4 to 5 ‰ $yr^{-1}$ (Graven et al., 2012), a 100 ‰ error would equate to a 18-25 year age error. This age error may not be impactful for ancient C scenarios (such as permafrost thaw), but may make a difference when distinguishing different young C sources.

The fraction of atmosphere-sourced $CO_2$ ($f_a$) present in the soil profile caused by diffusive mixing scenarios typically associated with different types of soils can be a predictor for Method 1 error, as illustrated conceptually in Fig. 5. Soil environments that typically have higher soil diffusivities and lower production rates will have smaller soil-diffusive gradients, and there is a larger amount of atmospheric $CO_2$ in the soil compared to the total $CO_2$. These scenarios have a much higher $f_a$ value. In these scenarios, diffusive mixing dominates, which amplifies Method 1 error. In contrast, in soils with larger soil-diffusive gradients, there is a much smaller amount of atmospheric $CO_2$ compared to the total amount of $CO_2$, and therefore the isotopic depth profile more closely resembles a mass-mixing profile. These soil $CO_2$ depth profiles have lower $f_a$ values and thus Method 1 will be less erroneous in these scenarios, because diffusive mixing is not as prominent.

The clay-soil at our field site (profiles shown in Fig. 4) likely has relatively low soil diffusivity, and is most like the lower soil diffusivity scenarios in Fig. 3. In Fig. 5, this soil is like the intermediate scenario, with an intermediate $f_a$ value, and diffusive mixing gradient. The use of Method 1 in this type of soil environment would be less erroneous than, for example, a dry desert soil environment with a high soil diffusion rate. Based on our model simulations, we can assume that Method 2 is producing the true $\Delta^{14}C$ value of production, so Method 1 error can be calculated as the difference between Method 1 and 2 for this real data. Method 1 error was minor in February (10 ‰; 2 to 2.5 year age error), but it was very large at depth in May (up to 100 ‰; 18-25 year age error).

This analysis has a few limitations. The synthetic soil environment is simplified in our approach, and in reality soil diffusion and production rates are not constant through depth. The model is necessarily simplified so that it can be easily solved analytically, but in reality soils are typically not in steady state (van Asperen et al., 2017; Bowling et al., 2009; Bowling and



Massman, 2011; Goffin et al., 2014; Maier et al., 2010; Moyes et al., 2010; Nickerson and Risk, 2009b; Risk and Kellman, 2008), Our calculated error values are therefore conservative, as these non-equilibrium processes will only add error and uncertainty.

**5.2 Transferability Across Sampling Methodologies**

Some sampling methods disturb the soil's natural diffusive regime, and for those our proposed correction may not be appropriate. One such example is the Hirsch et al. (2002) study, which uses a variant of Method 1. In their study soil $CO_2$ was pulled from a soil gas well over 1-2 weeks with a pump. It would be difficult to correct for transport fractionations in this case, because the soil diffusive gradient would have been disturbed by direct soil air withdrawal. Additionally, over 1-2 week intervals, the soil diffusive gradient would have changed due to changes in temperature and moisture inputs.

Surface flux chambers are commonly used for measuring the radiocarbon composition of production. Method 1 is actually acceptable for use in the case of surface flux chambers, because unlike soil $CO_2$ which will always differ from soil production soil-respired $CO_2$, conservation of mass dictates that isotopic values of flux must represent soil production so long as the soil is in steady-state (Cerling et al., 1991). While radiocarbon surface flux data need no correction for transport fractionation, researchers should be cautious when using surface flux chambers because they can cause isotopic dis-equilibrium (Albanito et al., 2012; Egan et al., 2014; Midwood and Millard, 2011; Nickerson and Risk, 2009a). As shown in the Egan et al. (2014) study, static chamber methods (i.e. Hahn et al., 2006) and the proposed forced-diffusion chamber technique were the least erroneous for radiocarbon measurements, whereas dynamic chamber sampling techniques (i.e. Gaudinski et al., 2000; Schuur and Trumbore, 2006) could cause up to 200 ‰ bias under certain soil diffusion and production scenarios. The Garnet et al. (2009) study used a passive sampling variation to the dynamic surface flux chambers technique, where a pump pulled a sample from the chamber and pushed it through a molecular sieve trap over a few days. This study suggested that the conventional Stuiver and Polach (1977) radiocarbon correction accommodated isotopic fractionation by the sieve. However, under non-steady state conditions, the conventional correction may not actually apply because the $^{14}CO_2/^{12}CO_2$ fractionation factor will not always be a constant multiple of the $^{13}CO_2/^{12}CO_2$ fractionation factor as the system moves from one state to another (Egan et al., 2014).

The final commonly-used technique to determine the radiocarbon composition of particular $CO_2$ sources is soil and root incubations (Hicks Pries et al., 2013; Schuur and Trumbore, 2006). Incubation jars are effectively 1-D chambers so there is no opportunity for bias caused by lateral diffusion, as there could be for 3-D chambers. The main issue with incubation jars is that the oxygen concentration within the jar could change throughout the length of the incubation period. Changes in oxygen availability will also affect fractionations related to the rate of production.





Aside from our proposed correction, a few other analytical alternatives might be appropriate for correcting $^{14}$C in soil CO$_2$. The Davidson (1995) $\delta_J$ method was the only gradient approach we tested for Method 2 in our study, but alternative approaches such as those presented for $\delta^{13}$C by Goffin et al. (2014) and Nickerson et al. (2014), and for $\Delta^{14}$C by Phillips et al. (2013) would likely be similarly successful in producing depth-dependent compositions of production. They are, however, not quite as straightforward as the $\delta_J$ method. However, if a researcher chooses to use an alternate gradient approach, they should still use $\Delta^{14}C_{new}$ rather than $\Delta^{14}C_{old}$ to calculate soil CO$_2$ first, as $\Delta^{14}C_{old}$ does not account for transport fractionations. To compare the degree of error associated with using $\Delta^{14}C_{old}$ with another gradient approach (i.e. Phillips et al., 2013; called Phillips2013 Method here) to Method 1 and Method 2, we calculated error values for a soil environment with given transport parameters in Table 2. As expected, Method 1 error was the largest, and the error ranged from 150 ‰ at the soil-atmosphere boundary, where there was a large f$_a$, to 17 ‰ further down the soil profile where f$_a$ was much lower. The Phillips2013 Method error was lowest at the soil-atmosphere boundary and became higher at depth, with 6.4 ‰ being the largest error, which was still larger than the 3.2 ‰ uncertainty associated with AMS precision and sampling error (from the use of gas canisters) reported in the Phillips et al. (2013) study. As expected, Method 2 had no error through depth.

In at least one other specialized instance, researchers have recognized that the normal Stuiver and Polach (1977) reporting convention was not applicable under the circumstances of an experiment and chose to reformulate it for their application. The Torn and Southon (2001) study evaluated the use of $\Delta^{14}C_{old}$ when radiocarbon is used as a tracer in C cycling field experiments with elevated atmospheric CO$_2$ concentrations. Torn and Southon (2001) proposed that in these experiments, the $\delta^{13}$C correction used in $\Delta^{14}C_{old}$ was invalid because differences in $^{13}$C abundance associated with elevated atmospheric CO$_2$ was associated with mixing of different atmospheric masses, and not isotopic fractionation. They instead used a $\delta^{13}$C value from an adjacent control plot (non-elevated CO$_2$ concentrations) to accurately estimate $\Delta^{14}$C, because the control plot followed the same fractionation pathways, but without the elevated atmosphere.

Both the Torn and Southon (2001) study and ours highlight the importance of reassessing old isotopic approaches for new application environments. To date, only three known studies (Egan et al., 2014; Phillips et al., 2013; Wang et al., 1994) have accounted for $^{14}$C diffusion-transport, though ours is the first to propose a straightforward and theoretically-robust correction that replaces the Stuiver and Polach (1977) solution for the soil gas environment (Method 2).

### 5.4 Workarounds and Establishing New Best Practice

More research groups are starting to use soil gas wells/soil CO$_2$ in conjunction with gradient techniques because of the known isotopic effects caused by many chamber techniques (Albanito et al., 2012; Egan et al., 2014; Midwood and Millard, 2011; Nickerson and Risk, 2009a). Gradient approaches also allow researchers to determine depth-dependent values of production. Previously reported data using the traditional Stuiver and Polach (1977) reporting convention, can be back-corrected using our solution. The back correction solution was tested with the synthetic model and is a fix that works, though we recommend against using this back correction method for new data. For researchers who have soil CO$_2$ data previously



interpreted using the $\Delta^{14}C_{old}$ calculation, the following steps will help correct for transport fractionations: 1) use $\delta_s^{13}$ and $\Delta^{14}C_{old}$ to back out the activity of the sample ($A_s$); 2) calculate the isotopic composition of production for $\delta^{13}C$ using Eq. (2), $\delta_J^{13}$; 3) use $\delta_J^{13}$ and $A_s$ in Eq. (7) to calculate $\Delta^{14}C_{new}$, and finally 4) determine the radiocarbon isotopic composition of production, using Eq. (16), $\Delta_J^{14}$.

Going forward, several changes to best practice are recommended. On a lab level, for new soil $CO_2$ data, we propose that AMS laboratories report radiocarbon using Eq. (3), $\delta^{14}C$, the uncorrected radiocarbon variant, so that the first step above, i.e. use $\delta_s^{13}$ and $\Delta^{14}C_{old}$ to back out the activity of the sample ($A_s$), can be avoided, and researchers can proceed with steps 2-4.

The Stuiver and Polach solution is, however, appropriate for solid sample analysis in the soil environment, and for determining the radiocarbon composition of atmospheric $CO_2$ samples.

**6 Conclusions**

As our fieldwork and analysis has shown, there could be error of 100 ‰ for researchers using the traditional $\Delta^{14}C$ reporting
15 convention, where soil $CO_2$ is used to interpret sources and ages of production. In cases where we are trying to predict the turnover rate and ages of sources of $CO_2$ in future climate scenarios, an error this large is unacceptable. This traditional $\Delta^{14}C$ solution, which uses a $\delta^{13}C$ correction, is not appropriate for the soil gas environment. We propose a new best practice for $\Delta^{14}C$ work in the soil gas environment that accounts for gas transport fractionations and produces true estimates of $\Delta^{14}C$ of production.

**Author Contributions**

JE, DB and DR conceptualized the theory and method for proving the new solution for radiocarbon applications in the soil gas environment. JE carried out the modeling, validation, visualization and writing of the original draft. DB, DR and JE reviewed and edited the draft.

**Acknowledgements**

Thanks to Thure Cerling for helpful discussions on the manuscript. JE is grateful for support from a Research-in-Residence Award from the Interuniversity Training in Continental-scale Ecology Project, US National Science Foundation
30 Macrosystems Biology Program under award EF-1137336. JE was also funded by the Natural Sciences and Engineering Research Council of Canada (NSERC). Additional support was provided by the US Department of Energy, Office of Science, Office of Biological and Environmental Research, Terrestrial Ecosystem Science (TES) Program under award number DE-SC0010625.




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




**Table 1. List of symbols used.** Note the isotope composition ratios are also unitless but traditionally expressed using permil (‰) notation.

| Symbol | Description | Unit |
|---|---|---|
| $A_s$ | sample activity | unitless |
| $A_{SN}$ | normalized sample activity, relative to $\delta^{13}C$ of terrestrial wood | unitless |
| $A_{abs}$ | age-corrected absolute international standard for activity | unitless |
| $Conc$ | $CO_2$ concentration | µmol m$^{-3}$ |
| $Conc_{atm}$ | $CO_2$ concentration in air just above the soil | µmol m$^{-3}$ |
| $C_a$ | $CO_2$ mole fraction in air just above the soil | µmol mol$^{-1}$ |
| $CO_2$ | $CO_2$ mole fraction relative to dry air | µmol mol$^{-1}$ |
| $C_s$ | $CO_2$ mole fraction in soil pore space | µmol mol$^{-1}$ |
| $D$ | soil gas diffusivity | m$^2$ s$^{-1}$ |
| $D(z,t)$ | soil gas diffusivity at depth z and time t | m$^3$ s$^{-1}$ |
| $\delta^{13}C$ | stable ($^{13}C/^{12}C$) isotope composition (relative to VPDB) | ‰ |
| $\delta^{14}C$ | radiocarbon ($^{14}C/^{12}C$) isotope composition (relative to A$_{abs}$) | ‰ |
| $\Delta^{14}C_{old}$ | radiocarbon ($^{14}C/^{12}C$) isotope composition with $\delta^{13}C$ correction | ‰ |
| $\Delta^{14}C_{new}$ | radiocarbon ($^{14}C/^{12}C$) isotope composition with $\delta_J^{13}$ correction | ‰ |
| $\delta_a^{13}$ | $\delta^{13}C$ of $CO_2$ in air above the soil | ‰ |
| $\delta_a^{14}$ | $\delta^{14}C$ of $CO_2$ in air above the soil | ‰ |
| $\Delta_a$ | $\Delta^{14}C$ of $CO_2$ in air above the soil | ‰ |
| $\delta_J^{13}$ | $\delta^{13}C$ of $CO_2$ from soil production, calculated using Eq. (2) | ‰ |
| $\delta_J^{14}$ | $\delta^{14}C$ of $CO_2$ from soil production, calculated using Eq. (4) | ‰ |
| $\Delta_{Jnew}^{14}$ | $\Delta^{14}C$ of $CO_2$ from soil production, calculated using Eq. (17) | ‰ |
| $\delta_s^{13}$ | $\delta^{13}C$ of $CO_2$ in soil pore space | ‰ |
| $\delta_s^{14}$ | $\delta^{14}C$ of $CO_2$ in soil pore space | ‰ |
| $\Delta_s$ | $\Delta^{14}C$ of $CO_2$ in soil pore space | ‰ |
| $f_a$ | fraction of $C_a$ in soil relative to total $CO_2$ in soil pore space | unitless |
| $L$ | lower model depth boundary | m |
| $P(z,t)$ | biological production rate at depth z and time t | µmol $CO_2$ m$^{-3}$ s$^{-1}$ |
| $P$ | biological production rate | µmol $CO_2$ m$^{-3}$ s$^{-1}$ |
| $R_s$ | isotopic ratio (heavy/light) of $CO_2$ sample | unitless |
| $R_{VPDB}$ | isotopic ratio (heavy/light) of Vienna Pee Dee Belemnite standard | unitless |
| $t$ | time | s |
| $\theta$ | air-filled porosity of soil | unitless |
| $z$ | depth | m |





**Table 2. Error calculations for Method 1, Method 2 and Phillips2013 Method. The depth profile used to calculate the method error was generated from a model with a $\Delta^{14}C$ of production of -200 ‰, a soil production rate of 2 μmol m$^{-3}$ s$^{-1}$, and a diffusion rate of 3x10e-6 m$^{-2}$ s$^{-1}$. All error values are absolute. Method 1 Error = -200 ‰ - $\Delta^{14}C_{old}$. Method 2 and the Phillips2013 Method Error = -**

5   **200 ‰ - $\Delta_j^{14}$. The $\Delta_j^{14}$ values are calculated using $\Delta^{14}C_{new}$ for soil $CO_2$ values, whereas the Phillips2013 Method uses $\Delta^{14}C_{old}$.**

| Depth (cm) | $\Delta^{14}C_{old}$ (‰) | Method 1 Error (‰) | $\Delta^{14}C_{new}$ (‰) | $\Delta_j^{14}$ (‰) | Method 2 Error (‰) | $\Delta_j^{14}$ from $\Delta^{14}C_{old}$ (‰) | Phillips2013 Method Error (‰) |
|---|---|---|---|---|---|---|---|
| 0.025 | -49.9 | 150.1 | -24.3 | - 200 | 0 | -201.6 | 1.6 |
| 0.05 | -98.8 | 101.2 | -80 | -200 | 0 | -203.4 | 3.4 |
| 0.1 | -137.6 | 62.4 | -123.5 | -200 | 0 | -204.8 | 4.8 |
| 0.2 | -163 | 37.0 | -151.9 | -200 | 0 | -205.7 | 5.7 |
| 0.3 | -172.5 | 27.5 | -162.5 | -200 | 0 | -206. | 6 |
| 0.4 | -177.2 | 22.8 | -167.8 | -200 | 0 | -206.2 | 6.2 |
| 0.5 | -180 | 20.1 | -170.8 | -200 | 0 | -206.3 | 6.3 |
| 0.6 | -181.5 | 18.5 | -172.5 | -200 | 0 | -206.3 | 6.3 |
| 0.7 | -182.3 | 17.7 | -173.4 | -200 | 0 | -206.4 | 6.4 |
| 0.8 | -182.6 | 17.4 | -173.7 | -200 | 0 | -206.4 | 6.4 |





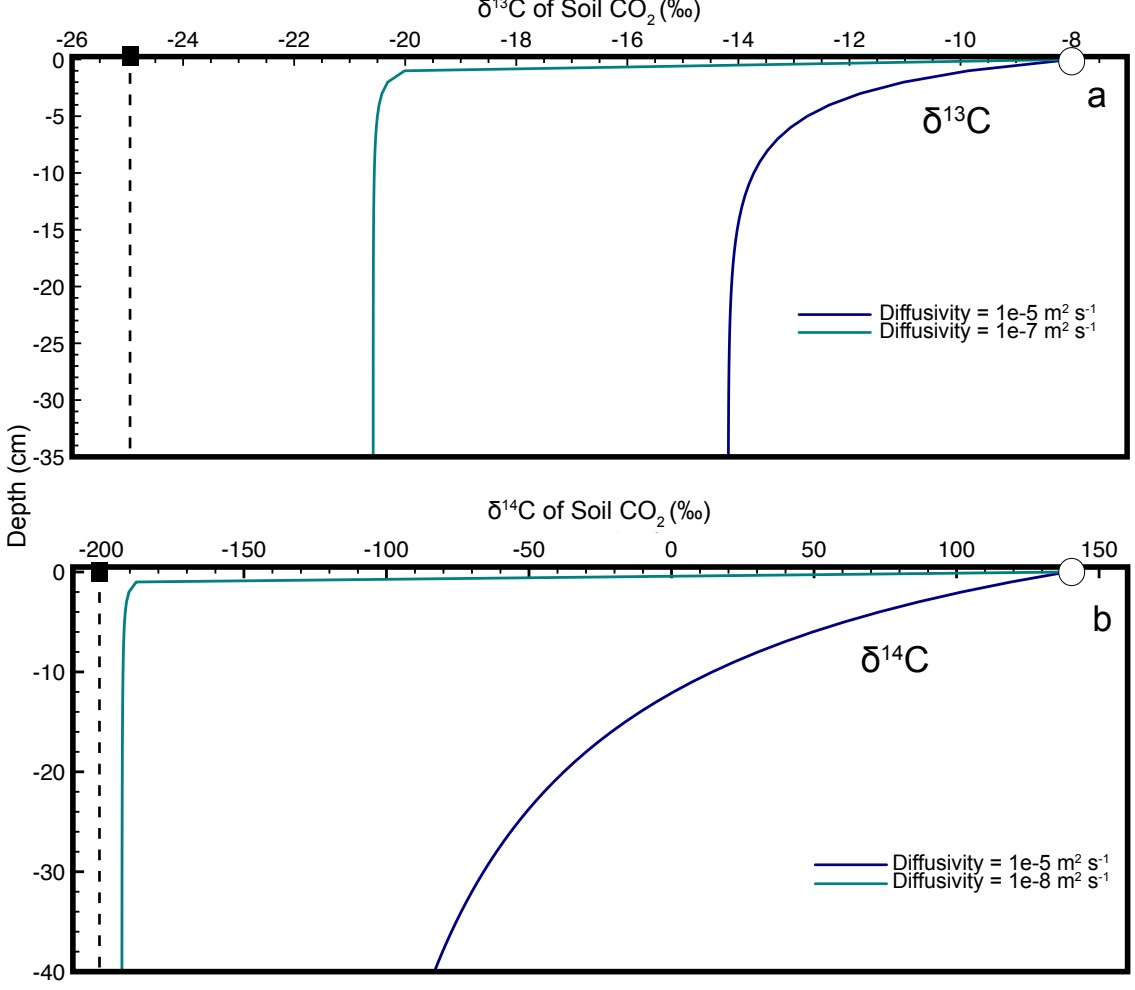

**Figure 1: Modelled steady-state diffusive vertical depth profiles for $\delta^{13}C$ and $\delta^{14}C$ of soil CO$_2$. In the top panel the $\delta^{13}C$ of atmospheric CO$_2$ (circle) is -8 ‰ and CO$_2$ from biological production (square with dashed line; $\delta_J$) is -25 ‰. In the bottom panel the $\delta^{14}C$ of atmospheric CO$_2$ (circle) is 138 ‰ and CO$_2$ from biological production (square with dashed line) is -200 ‰. Both profiles in each of the panels have the same biological production rates and isotopic composition of biological production, but each profile has a different soil diffusivity.**









**Figure 2: Modelled steady-state diffusive vertical depth profiles for $\delta^{13}C$ (top panel; a), $\delta^{14}C$ (middle panel; b), and $\Delta^{14}C_{old}$ (bottom panel; c) of soil $CO_2$. The three soil profiles were generated using the same soil production and diffusivity rates (1e$^{-6}$ m$^2$ sec$^{-1}$ and 2 μmol m$^{-3}$ sec$^{-1}$, respectively). Panels 2(a) and 2(b) were prepared using $\delta^{13}C$ and $\delta^{14}C$ as noted. Panel 2(c) shows an approach consistent with present day, where the $\Delta^{14}C$ profile generated by the model incorporates the traditional Stuiver and Polach (1974)**
5     **correction for biochemical fractionation. Inset "Calculated" panels show how, using input data read directly from each depth profile, a user would arrive at either the correct, or incorrect isotopic value of production using a Davidson approach to adjust for in-soil gas transport. The atmospheric source (Ca) composition is presented as a white circle, the soil $CO_2$ composition (C$_s$) is a black circle, and the isotopic composition of production is a black square. Note that values for the isotopic composition of soil in the three panels are rounded for ease of reading, but are actually -20.1745 ‰, -178.9879 ‰, and -187.0547 ‰ respectively for**
10     **panels (a), (b), and (c). These values are drawn from the curve at a depth of 20 cm.**





**Figure 3: Modelled steady-state vertical depth profiles for $\Delta^{14}C$ of soil $CO_2$. In panel A the model scenarios have the same rates of production (P), but differing diffusivities (D) (solid lines are Method 1, $\Delta^{14}C_{old}$; dashed lined are Method 2, $\Delta^{14}C_{new}$). In panel B the model scenarios have the same diffusivities but differing production rates. The model input for $\Delta^{14}C$ of production was -200 ‰ in**

5 **all cases ($\Delta_J^{14}$; black dashed line) and $\Delta^{14}C$ of the atmospheric source was 100 ‰ (white circle).**




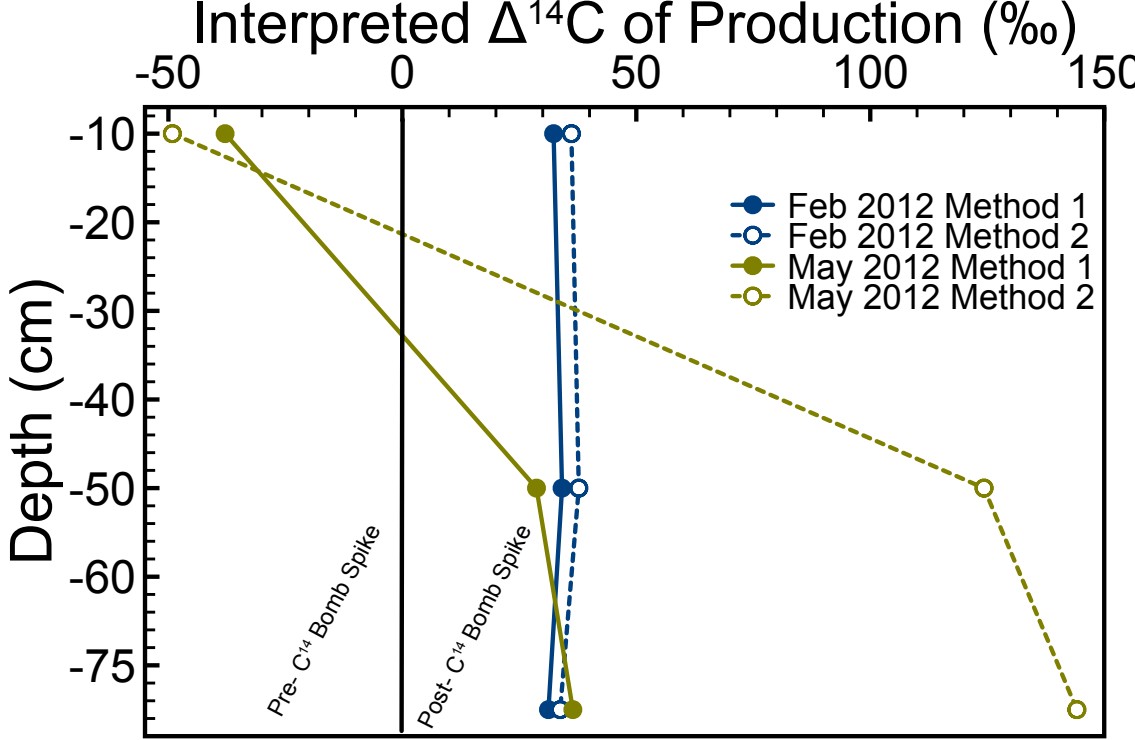

**Figure 4: Field results for interpreted $\Delta^{14}C$ of production calculated from soil $CO_2$ gas samples collected at 10 cm, 50 cm, and 75 cm depths from soil gas wells in February and May 2012 from a site in Saskatchewan, Canada. Solid lines are $\Delta^{14}C$ calculated using Method 1 and the dashed lines are calculated using Method 2.**





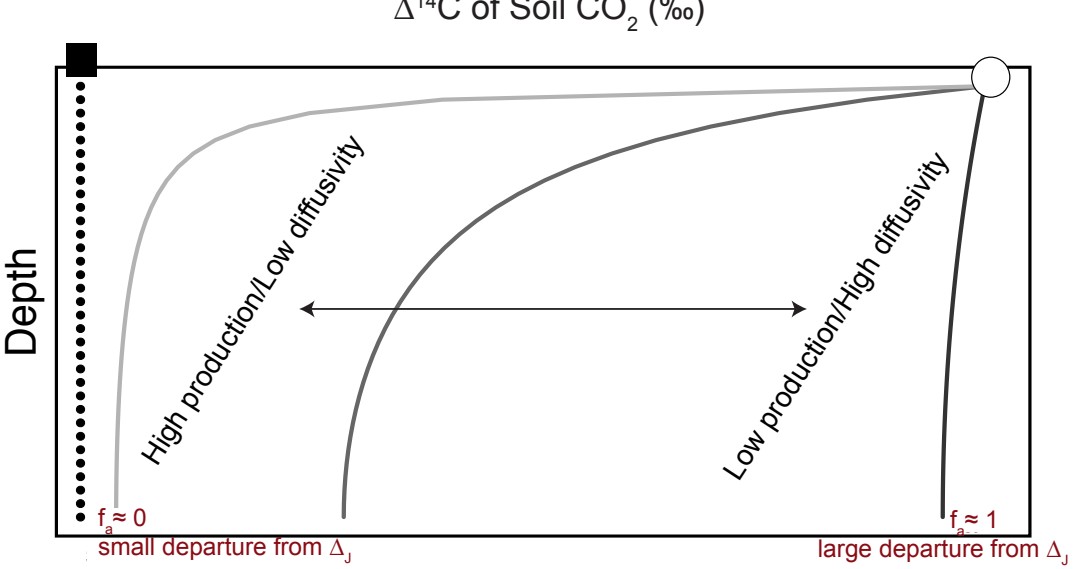

**Figure 5: Method 1 error can be estimated using $f_a$, the fraction of soil $CO_2$ that originated from the atmosphere that has mixed downward into the soil gas profile. The radiocarbon composition of biological production is presented with a black square, and the atmospheric source of $CO_2$ is presented as a white circle.**

