# Peer review of "Technical Note: Isotopic corrections for the radiocarbon composition of $CO_2$ in the soil gas environment must account for diffusion and diffusive mixing"

_Biogeosciences, 2018_

## Referee Comment (RC1) · Anonymous Referee #1 · 4 Nov 2018

The reviewed manuscript proposed new corrections for 14C measurements in soil air, to account for diffusion and mixing. Experimental study, and modeling are used to illustrate the point, and recommendation for best practice are given. The logic of the authors is clear, and usually the manuscript is well written. My main concerns are as follows:

1) The authors writes that the problem with the traditional method of 13C based correction, is that it works only for biological process like photosynthesis and do not apply for example to fractionation in diffusion. This is not correct. The correction based on 13C should work for any mass-dependent fractionation (i.e most fractionation processes).

[Figure]

However, they are correct the traditional correction will not work for mixing. Because this is not a fractionating processes and no mass-dependent can be assumed. This error should be corrected in the text, abstract, and in the title (i.e. correction for mixing is not correction for fractionation).

2) Another point that can be mentioned in the text: In soils that contain carbonates, isotopic exchange of $CO_2$ with the carbonates can introduce additional error which is not mass-dependent.

3) It is stated that: "Graham's law of effusion (rate of diffusion $\sim$ 1/sqrt(mass)) provides us with an expected difference in diffusion rate of isotopologues." In fact, as it is written few lines before, this is a binary diffusion of $CO_2$ in air and hence binary diffusion equation (a one that invokes reduce-mass of $CO_2$ and air) should be used.

Minor comment: Please give a better description of the soil than just "clay".

---

## Referee Comment (RC2) · Anonymous Referee #2 · 25 Nov 2018

The paper proposes new corrections for the radiocarbon composition of CO2 in the soil gas environment. The paper describes in detail how the authors have derived these new corrections to include diffusion and mixing in their approach. The paper should revive and encourage further discussions on this important but somewhat under explored topic. The topic therefore up to now has remained only partly resolved. Even the new corrections proposed by in the current papers have certain limitations, which the authors rightly highlight in the discussion of the paper. Whereas, the isotope corrections are described in great detail, the field profile study is somewhat lacking in such details, especially the description of the site and soil used is rather scant. Please enhance this section. Furthermore, as the topic of the paper is rather specific, e.g. how and

if the way the various formula's relate to each other is correct, some specific detailed isotope/radiocarbon expertise is needed to verify this. In a more general sense sense the paper can be published with minor revision also in light in the perception that it creates a focal point for further discussion around this paper on such isotope fractionation correction for radiocarbon (14C) measurements.

―――――――――――――――――――――

---

## Referee Comment (RC3) · Anonymous Referee #3 · 3 Dec 2018

Egan et al. correctly state that radiocarbon corrections based on $^{13}$C cannot be used to interpret radiocarbon data if there are processes involved that involve process that are not mass-dependent, like mixing. They use simple 1D models to show the potential biases in estimating the radiocarbon signature of source gases if gases in soil air space are interpreted without understanding that soil air both mixes and diffuses. The main advance here is that the authors use information on $^{13}CO_2$ in pore space to estimate the mixing, which in turn allows a better way to estimate the $^{14}$C of $CO_2$ sources.

In some sense, the authors have set up Stuiver and Polach (1977) as a target to shoot down in a way that is not entirely fair. Stuiver and Polach is a paper that sets out the

conventions for reporting radiocarbon data – and only that. It is up to the investigator to interpret the radiocarbon data appropriately – including in the use of models such as those proposed in this paper. I would therefore not say that "their approach is wrong" as it was never their intention to tackle the interpretation of isotope profiles in soils. The authors need to be clear that the data should be correctly reported – i.e. according to Stuiver and Polach – but that to interpret soil $CO_2$ isotopic data requires the use of a model that includes transport and mixing processes. Does anyone in the literature actually assume Method 1?

The abstract states that diffusion and mixing are both problematic. For pure molecular diffusion, the fractionation should be mass dependent (i.e. not problematic). Mixing, however, will not be mass-dependent and therefore cause problems. However, by lumping both into "diffusion" in their model, the authors are confounding things. I think what they are calling "D" in their model is really an "effective" diffusivity. This would effectively mean adding 4.4 per mille to the $^{13}$C of the source in equation 15 (which then might explain differences with Phillips2013?)

In very deep soils, one needs to worry about additional issues that the authors have not included, such as gravitational and thermal effects associated with non-steady state conditions. These issues are treated in Severinghaus et al. 1996 (Fractionation of soil gases by diffusion of water vapor, gravitational settling, and thermal diffusion, Geochim. Cosmochim. Acta, 60: 1005-1018). Any model that hopes to infer the isotopic signatures of sources from field data of $CO_2$ in pore space needs to include the possibility of non-steady state conditions, or at least demonstrate that steady state is a reasonable assumption.

The authors are correct that a model such as the one they propose is required to deconvolve what is affecting observed gradients of $14CO_2$ and $13CO_2$ in soil pore space. However, it is worth pointing out that the authors have picked a rather extreme

condition where the $^{14}$C signature of decomposition derived $CO_2$ differs greatly (300 per mille) from the atmospheric value – in many soil profiles this is not the case as the sources of $CO_2$ are dominated by root respiration and decomposition of more labile soil organic matter rather than the decomposition of very old peat material, and the gradients in 14$CO_2$ (and thus the biases) are much smaller. (This of course depends on fA, based on Figure 4 – though we do not know what $^{13}$C or $[CO_2]$ were for these sites/dates).

The authors have assumed a case where both the production rate and the radiocarbon content of $CO_2$ produced by decomposition are constant with soil depth, whereas that is also not normally the case (production declines exponentially with depth, and radiocarbon tends to decline linearly). Why not test with a more realistic model? I do not accept the statement at the end of section 5.1 ("Our calculated error values are therefore conservative, as these non-equilibrium processes will only add error and uncertainty") without some demonstration of how sensitive the biases are to the assumptions. Also, I think the statements about the size of the potential error ("there could be error of 100 ‰") are a bit too specific to the case investigated. Certainly, there are biases, but perhaps these could be expressed as something a percentage of the difference between the $^{14}$C of sources and that of overlying air (in the most extreme case here, 30 per cent, or 1/3 of (-200-100). As noted above, more realistic simulations would likely estimate smaller absolute values, but not necessarily percentages, as these are based on the $^{13}$C differences.

The "field experiment" and the use of models to interpret these data are not well enough described for the reader to understand. Were the same assumptions (constant production and 'effective' diffusion with depth) applied for interpreting these results? Either this needs to be more thoroughly described (for example by including the $CO_2$ and 13$CO_2$ profiles) or removed from the paper, as it does not really fit with the rest.

Overal/, the authors need to do more sensitivity tests with their 'thought experiment" models; if they use the field data it needs to be much better described.

Other points:

Lines 10-15 in the Introduction are somewhat misleading. The 'traditional' way to report $^{14}$C values from $CO_2$ in soil air would use the $^{13}$C signature of the $CO_2$ that was sampled. This value is important as it by definition would include both the biochemical and physical (diffusion) effects in a system where mixing is not important. As noted above, the reporting of the data should not be confounded with the interpretation of the data.

Page 4, line 16 – the estimation of the isotopic effect is based on comparing the reduced masses of the $^{12}$C-air, $^{13}$C-air, or $^{14}$C-air system, not just the square root of the masses of the isotopic species of $CO_2$ alone (air has a 'mass' of 28). This is unclear from what is written.

Most radiocarbon is measured directly these days as isotope ratios, rather than activities as presented in equation (5). Perhaps this nomenclature could be updated.

While $\Delta^{14}$C is the $\delta^{13}$C-corrected version of $\delta^{14}$C as stated in line 24 on page 5, both of these include an additional correction for the decay of the standard since 1950 . (This is actually correct in the formula stated by the authors, which uses $A_abs$ , but could be really confusing for the reader who does not know what $A_{abs}$ is and how it differs from the equations (5) that refer only to the activity of oxalic acid. This correction is not negligible, the oxalic acid standard has decreased by about 10 ‰since 1950. (The same is not true for Fraction Modern or D$^{14}$C, which are not expressed relative to $A_{abs}$ , because the standard and sample are presumed to decay at the same rate). Because

of this, the authors should really state the assumed year of sampling/measurement in their simulations (as noted above, $\Delta^{14}$C and $\delta^{14}$C will differ slightly depending on the year of measurement/sampling).

The current $\Delta^{14}$C of atmospheric $CO_2$ is closer to 10 ‰ than 100 ‰.

One problem with the "new" approach is that one must assume that they know the $\delta^{13}$C signature of the $CO_2$ being produced – what is the sensitivity to getting that value incorrect by 1-2 ‰?

Table 2 compares with "Phillips et al 2013" but that is first mentioned in the Discussion – it should be explained in the methods or the introduction if used.

Page 12 , line 20. Presumably the authors are pointing out that the movement of air in dynamic chambers could cause some kind of enhanced exchange with soil pore space air – where does the 200 ‰ number come from? – is there a reference for this? Wouldn't mixing also mean that in this case the 'standard' $^{13}$C correction would not be appropriate for interpreting the $^{14}$C of the measured flux?

Page 12, line 32. "Changes in oxygen availability will also affect fractionations related to the rate of production. " This statement needs a reference. What the authors mean here is that the source of C might change as might any fractionation associated with respiration under low O2. However, the isotopic signatures of $CO_2$ that accumulate in the chamber will still represent that source without need for additional corrections.

Page 14, lines 5-10. AMS laboratories measure all three isotopes, but the $^{13}$C they measure includes fractionation processes in the sample preparation and measurement

(all of which are correctable using mass-dependent assumptions). However, the AMS-measured $\delta^{13}$C is not a good measure compared to the $^{13}$C of the $CO_2$ that is measured with an IRMS. Therefore, the labs cannot easily report the $\Delta^{14}$C as the authors propose unless the user has independently supplied the $\delta^{13}$C values for their samples. Instead I would advise people to 'uncorrect' the reported data themselves (and give them the formula). The current guidelines are to use Fraction Modern and report the year of collection and the year of measurement so that other ways of expressing $^{14}$C can be calculated. Also, AMS labs do not measure activity, they measure isotope ratios.

---

## Author Comment (AC1) · 15 Jan 2019

Manuscript bg-2018-451 Title: Isotopic fractionation corrections for the radiocarbon composition of CO2 in the soil gas environment must include diffusion and mixing Authors: Jocelyn Egan et al.

Thanks to the referees and the editor for helpful comments that have led to a much-improved manuscript. Referee comments are listed and our responses follow each - line numbers refer to those in the revised version.

Sincerely, Jocelyn Egan (for all authors)

Referee #1

The reviewed manuscript proposed new corrections for 14C measurements in soil air, to account for diffusion and mixing. Experimental study, and modeling are used to illustrate the point, and recommendation for best practice are given. The logic of the authors is clear, and usually the manuscript is well written. Thanks for the supportive comments. The authors writes that the problem with the traditional method of 13C based correction, is that it works only for biological process like photosynthesis and do not apply for example to fractionation in diffusion. This is not correct. The correction based on 13C should work for any mass-dependent fractionation (i.e most fractionation processes). However, they are correct the traditional correction will not work for mixing. Because this is not a fractionating processes and no mass-dependent can be assumed. This error should be corrected in the text, abstract, and in the title (i.e. correction for mixing is not correction for fractionation).

Response: We agree, and this is merely an issue of terminology. Although mixing might not be considered a traditional fractionating process, there are important isotopic effects associated with mixing. An isotopic fractionation is defined as a difference in the isotopic composition between a reactant and a product. For the application here with radiocarbon, the radiocarbon composition of biological respiration is the "reactant" and the soil gas is the "product". We must take into account all isotopic effects that influence the product in order to understand the true isotopic composition of the reactant. Therefore both diffusion and mixing need to be included and accounted for and that is the central message of our paper. We have changed wording in the text to address this:

Title: removed the word fractionation.

Page 3 lines 6-7.

Another point that can be mentioned in the text: In soils that contain carbonates, isotopic exchange of $CO_2$ with the carbonates can introduce additional error which is not

mass-dependent.

Response: Agreed, but carbonates are not the focus of the paper.

It is stated that: "Graham's law of effusion (rate of diffusion âĹij 1/sqrt(mass)) provides us with an expected difference in diffusion rate of isotopologues." In fact, as it is written few lines before, this is a binary diffusion of $CO_2$ in air and hence binary diffusion equation (a one that invokes reduce-mass of $CO_2$ and air) should be used.

Response: Agreed. We have modified the text to remove the mention of Graham's law (see line 19 on page 5).

Minor comment: Please give a better description of the soil than just "clay".

Response: More detail has been added (lines 21-23 on page 11).  

Please also note the supplement to this comment:
https://www.biogeosciences-discuss.net/bg-2018-451/bg-2018-451-AC1-supplement.pdf

---

## Author Comment (AC2) · 15 Jan 2019

Manuscript bg-2018-451 Title: Isotopic fractionation corrections for the radiocarbon composition of CO2 in the soil gas environment must include diffusion and mixing Authors: Jocelyn Egan et al.

Thanks to the referees and the editor for helpful comments that have led to a much-improved manuscript. Referee comments are listed below, and our responses follow each - line numbers refer to those in the revised version.

Sincerely, Jocelyn Egan (for all authors)

The paper proposes new corrections for the radiocarbon composition of CO2 in the soil gas environment. The paper describes in detail how the authors have derived these new corrections to include diffusion and mixing in their approach. The paper should revive and encourage further discussions on this important but somewhat under explored topic. The topic therefore up to now has remained only partly resolved. Even the new corrections proposed by in the current papers have certain limitations, which the authors rightly highlight in the discussion of the paper.

Response: Thanks for the positive comments.

Whereas, the isotope corrections are described in great detail, the field profile study is somewhat lacking in such details, especially the description of the site and soil used is rather scant. Please enhance this section.

Response: In this paper we did not intend to use the field data as validation for the method, but as an example of how the new theory could be applied. We prefer to avoid a greater emphasis on the field study than we have included. However, we have added a few more details about the soils (page 11 lines 21-23).

Furthermore, as the topic of the paper is rather specific, e.g. how and if the way the various formula's relate to each other is correct, some specific detailed isotope/radiocarbon expertise is needed to verify this.

Response: This comment is too vague for us to craft a response. The authors collectively have decades of experience with isotopes, and hopefully our perspective will be taken seriously.

In a more general sense the paper can be published with minor revision also in light in the perception that it creates a focal point for further discussion around this paper on such isotope fractionation correction for radiocarbon (14C) measurements.

Response: Thanks, we would also like to see it published. 

[Figure]

Please also note the supplement to this comment:
https://www.biogeosciences-discuss.net/bg-2018-451/bg-2018-451-AC2-
supplement.pdf

—————————————————————

**BGD**

[Figure]

**Supplement:**

[revised manuscript text omitted]

$P(z) = P.$ $\hfill$ (11)

The following boundary conditions were used:

20 $\qquad C(z = 0) = Conc_{atm},$ $\hfill$ (12)

$\qquad \frac{\partial C}{\partial z}\big|_{z=L} = 0,$ $\hfill$ (13)

where $Conc_{atm}$ is the concentration of $CO_2$ in air just above the soil and $L$ is the model lower spatial boundary, the point below which no production or diffusion occurs. Eq. (8) is solved analytically to yield the following equation:

$$Conc(z) = \frac{P/L}{D}\left(L \times z - \frac{z^2}{z}\right) + Conc_{atm}.$$

5              (14)

In the model, isotopologues of $CO_2$ are treated as independent gases, with their own specific concentration gradients and diffusion rates (Cerling et al., 1991; Nickerson and Risk, 2009b; Risk and Kellman, 2008). We assume total $CO_2$ to be $^{12}CO_2$ because of its high abundance. The error associated with this assumption is less than 0.01% (Amundson et al., 1998). Eq. (14) is thus applied for $^{13}CO_2$ and
10    $^{14}CO_2$. For the full derivation see Nickerson et al. (2014) Section 2.3.

The analytical gas transport model was applied across a range of soil diffusivity ($1 \times 10^{-7}$, $1 \times 10^{-6}$ and $1 \times 10^{-5}$ $m^2$ $s^{-1}$) and soil production rates (0.5, 1, 2, and 4 $\mu$mol $CO_2$ $m^{-3}$ $s^{-1}$), with a depth resolution of 0.025 m. In addition it used $\Delta^{14}C$ and $\delta^{13}C$ compositions of soil biogeochemical production and
15    atmospheric $CO_2$, all representing realistic conditions found in nature. The other model boundary conditions were as follows: $L = 0.8$ m, $z = 0.025$ m, $Conc_{atm} = 15833$ $\mu$mol $m^{-3}$ (~380 ppm), $\delta^{13}C$ of atmospheric $CO_2$ ($\delta_a$) = -8 ‰, $\delta^{13}C$ of production = - 30 ‰, $\Delta^{14}C$ of atmospheric $CO_2$ ($\Delta_a$) = 100 ‰, and $\Delta^{14}C$ of production = -200 ‰. The output of the model under these applied conditions were profiles of $^{12}CO_2$, $^{13}CO_2$, and $^{14}CO_2$ for each depth (z) down to the bottom boundary (L).

20  **3.2 Testing the Traditional and Alternative Conventions**

The model-generated soil pore space values of $^{12}CO_2$, $^{13}CO_2$, and $^{14}CO_2$ at each depth were then used to produce soil depth profiles of $\delta^{13}C$, $\delta^{14}C$, and $\Delta^{14}C$ of $CO_2$ as demonstrated earlier in Figs. 1 and 2. To simulate the traditional convention for radiocarbon (referred to as $\Delta^{14}C_{old}$ here), we input the soil pore space values into Eq. (7) to calculate the $\Delta^{14}C$ depth profiles.

Our proposed new approach is based on Davidson's (1995) theory. Rather than using the $\delta^{13}C$ soil pore space as a mass-dependent correction in Fig. 2, we suggest instead using the value $\delta_J^{13}$ (Eq. (2)), the biological production of $\delta^{13}C$, in its place in the denominator of Eq. (7) as follows:

$$\Delta^{14}C_{new} = \left[\left(\frac{A_S}{A_{abs}}\right)\frac{\left(1-\frac{25}{1000}\right)^2}{\left(1+\frac{\delta_J^{13}}{1000}\right)^2} - 1\right]1000.$$

5          (15)

The model-generated soil pore space isotope values were input into Eq. (15) to produce depth profiles of $\Delta^{14}C_{new}$. Values of $\Delta^{14}C_{new}$ through depth represent transport-fractionation-corrected soil $CO_2$ values of radiocarbon. For our simulated data, we assume the measurement year is present day, as we do not use the oxalic acid ($A_{abs}$) in Eq. 15. Then to calculate the radiocarbon composition of production, $\Delta_J^{14}$,

10   the $\Delta^{14}C$ composition of the soil $CO_2$, $\Delta^{14}C_{new}$, can be placed into our adaption of Davidson (1995) for $^{14}C$ (Eq. (4)) as follows:

$$\Delta_J^{14} = \frac{C_s(\Delta^{14}C_{new}-8.8)-C_a(\Delta_a^{14}-8.8)}{1.0088(C_s-C_a)},$$

        (16)

where $\Delta_J^{14}$ is the $\Delta^{14}C$ composition of soil production, $C_s$ and $\Delta^{14}C_{new}$ are the mole fraction and $\Delta^{14}C$

15   composition of the soil $CO_2$, and $C_a$ and $\Delta_a^{14}$ are the mole fraction and $\Delta^{14}C$ composition of $CO_2$ in the air just above the soil.

**3.3 Field Soil Profiles**

To compare the model results to field results, we also sampled soil depth profiles in the field. We collected samples of soil $CO_2$ at 3 depths from the soil profile and from the air just above the soil, from

20   3 sites in Weyburn, Saskatchewan, for both stable and radio-isotope analysis. These sites were located in agricultural fields, with soils characterized as a combination of Estevan and Roughbark soils. These soils have sandy loam surfaces, with thin organic layers and soil layers below that are poorly drained and have high clay content (Edwards et al., 1987).

To collect samples, we installed horizontal polyvinyl chloride (PVC) soil gas wells, 60 cm long, with 3-ply Gore-tex$^{TM}$ membranes covering 2 cm holes drilled every 10 cm along the length of PVC, at soil depths of 5 cm, 10 cm, 20 cm, 50 cm and 75 cm. Details on the soil gas well design and sampling disturbance prevention are described in detail in Risk et al. (2013). We withdrew a minimum of two samples (one shallow and one deep) and one air sample at each of the three sites 6 times (approximately every 2 months) between August 2011 and May 2012 as described in Risk et al. (2013), by connecting a $N_2$-purged and evacuated 1 L stainless steel canister (Lab Commerce) to the well and allowing it to equilibrate for 15 mins. The samples were sent to the University of Florida for cryogenic purification and then to the University of California Irvine Keck Accelerator Mass Spectrometer (AMS) facility to be graphitized and analyzed for $\Delta^{14}C$.

As described in Risk et al. (2013), on each visit triplicate samples were also collected in 10 ml $N_2$-purged and evacuated Exetainers vials (Labco, UK) at each depth as well as from the air above the soil. The triplicate samples were analyzed for bulk $CO_2$ and $\delta^{13}C$ within 2 weeks of sampling using a GV Isoprime CF-IRMS and Multiflow gas bench (Isoprime, UK).

**3.4 Interpreting Soil Profiles**

Our primary goal was to correctly extract the isotopic composition of $CO_2$, produced by biological production. We applied two methods to extract this signal from the model-generated soil $CO_2$ profiles as well as soil $CO_2$ profiles from the field:

Method 1 follows the traditional correction, where Eq. (7) was used to calculate $\Delta^{14}C_{old}$, and, following convention, we interpret this value as the radiocarbon composition of $CO_2$ from biological production.

Method 2, our proposed solution, instead determines the radiocarbon composition of $CO_2$ from biological production in two steps, as described in Section 3.2. First, the transport fractionation corrected radiocarbon composition of soil $CO_2$, is calculated using Eq. (15) ($\Delta^{14}C_{new}$). Then this

properly interpreted soil $CO_2$ value can be used in Eq. (16) to calculate the radiocarbon composition of production, $\Delta_J^{14}$.

In Section 5.2 we compare these two Methods to that used in Phillips et al. (2013), which we will call the Phillips2013 Method. Phillips et al. (2013) use a different gradient approach (not Eq. 16; $\Delta_J^{14}$) along with $\Delta^{14}C_{old}$ to calculate the radiocarbon composition of production.

[revised manuscript text omitted]

This analysis has a few limitations. The synthetic soil environment is simplified in our approach, and in reality soil diffusion and production rates are not constant through depth. The model is necessarily simplified so that it can be easily solved analytically, but in reality soils are typically not in steady state (van Asperen et al., 2017; Bowling et al., 2009; Bowling and Massman, 2011; Goffin et al., 2014; Maier et al., 2010; Moyes et al., 2010; Nickerson and Risk, 2009b; Risk and Kellman, 2008). For example, gravitational settling and thermal diffusion can impact the diffusive non-steady state in deep soils (Severinghause et al., 1996) and diurnal temperature cycles (Nickerson and Risk, 2009b; Phillips et al., 2010; Risk and Kellman, 2008) and wind advection (Bowling and Massman, 2011) can be of impact non-steady state in near surface soils. The error values presented here are therefore likely conservative, since previous $\delta^{13}C$ studies have shown that these non-equilibrium processes generally add fractionation uncertainty (Nickerson and Risk, 2009b; Phillips et al., 2010; Risk and Kellman, 2008). Additionally, sampling methods may impose an additional layer of non-equilibrium fractionation uncertainty (Egan et al., 2014) that may need to be evaluated. The real magnitude of error will be situationally specific, and sensitive to the soil conditions and sampling methodology.

[revised manuscript text omitted]
^{14}$. This solution assumes that the researcher has measured a value of $\delta_s^{13}$ independent from $\Delta^{14}C_{old}$, as the AMS measured $\delta^{13}$ is not comparable to Isotope-Ratio Mass Spectrometry (IRMS)

25  measured $\delta^{13}$.

Going forward, several changes to best practice are recommended. On a lab level, for new soil $CO_2$ data, we propose that AMS laboratories report radiocarbon using Eq. (3), $\delta^{14}C$, the uncorrected

radiocarbon variant, so that the first step above, i.e. use $\delta_s^{13}$ and $\Delta^{14}C_{old}$ to back out the activity of the sample ($A_s$), can be avoided, and researchers can proceed with steps 2-4. We also suggest that researchers measure $\delta^{13}$ alongside $\Delta^{14}C$, so that there are no issues with depending on the AMS measured $\delta^{13}$ for potential back-corrections, as $\delta_J^{13}$ is sensitive in calculating the proper $\Delta^{14}C_{new}$ (as shown in Fig. 2 – 4 decimal places needed for $\delta_J^{13}$).

[revised manuscript text omitted]

---

## Author Comment (AC3) · 15 Jan 2019

Manuscript bg-2018-451 Title: Isotopic fractionation corrections for the radiocarbon composition of CO2 in the soil gas environment must include diffusion and mixing Authors: Jocelyn Egan et al.

Thanks to the referees and the editor for helpful comments that have led to a muchimproved manuscript. Referee comments are listed below, and our responses follow each - line numbers refer to those in the revised version.

Sincerely, Jocelyn Egan (for all authors)

Egan et al. correctly state that radiocarbon corrections based on 13C cannot be used to interpret radiocarbon data if there are processes involved that involve process that are not mass-dependent, like mixing. They use simple 1D models to show the potential biases in estimating the radiocarbon signature of source gases if gases in soil air space are interpreted without understanding that soil air both mixes and diffuses. The main advance here is that the authors use information on 13CO2 in pore space to estimate the mixing, which in turn allows a better way to estimate the 14C of CO2 sources.

Response: Agreed, we feel this is an important advance.

In some sense, the authors have set up Stuiver and Polach (1977) as a target to shoot down in a way that is not entirely fair. Stuiver and Polach is a paper that sets out the conventions for reporting radiocarbon data – and only that. It is up to the investigator to interpret the radiocarbon data appropriately – including in the use of models such as those proposed in this paper. I would therefore not say that "their approach is wrong" as it was never their intention to tackle the interpretation of isotope profiles in soils. The authors need to be clear that the data should be correctly reported – i.e. according to Stuiver and Polach – but that to interpret soil CO2 isotopic data requires the use of a model that includes transport and mixing processes.

Response: Agreed. However, we do not say that their approach is "wrong". In the abstract we state "We found that the conventional correction is inappropriate for interpreting the radio-isotopic composition of CO2 from biological production, because it does not account for diffusion and diffusive mixing." and we stand by this statement. Our intention was not to target Stuiver and Polach (1977), but to highlight problems that have cropped up in the literature with regard to misinterpretation. We have modified the text in the following places to highlight the important of interpretation as the referee suggests: line 16 on page 1, and lines 6-7 and 23 on page 2.

We also want it to be clear that in our proposed solution, a model is not required to interpret soil CO2 isotopic data. Here, we have used a model to illustrate the problem
and solution, but with real data we suggest that people use  $\triangle$ 14Cnew and  $\triangle$ J14 to report radiocarbon results measured in the soil gas environment (as shown in section 5.4).

Does anyone in the literature actually assume Method 1?

Response: We don't know the answer to this question. It's very hard to determine all the behind-the-scenes corrections that occur in complicated isotope papers. We have not found any treatment of diffusion and mixing in the literature similar to ours. We feel that it's important to get this issue out in the open so that people can consider it, and we have tried to present this as a way forward rather than targeting particular studies.

The abstract states that diffusion and mixing are both problematic. For pure molecular diffusion, the fractionation should be mass dependent (i.e. not problematic). Mixing, however, will not be mass-dependent and therefore cause problems. However, by lumping both into "diffusion" in their model, the authors are confounding things. I think what they are calling "D" in their model is really an "effective" diffusivity. This would effectively mean adding 4.4 per mille to the 13C of the source in equation 15 (which then might explain differences with Phillips2013?)

Response: The model used is a steady-state diffusion model, and it does not include advection or dispersion. However, mixing does of course occur via diffusion, and given boundary conditions of atmospheric CO2 on one end and biologically-produced CO2 on the other, with isotopic differences between them, the soil gas will be a diffusively-mixed environment. As pointed out by the referee 3 in the earlier comments above, it's the interpretation that is important. Our main point is to highlight that interpreting the isotopic composition of soil CO2 without appropriately accounting for the isotopic processes that affect it will lead to error.

In very deep soils, one needs to worry about additional issues that the authors have not included, such as gravitational and thermal effects associated with non-steady state conditions. These issues are treated in Severinghaus et al. 1996 (Fractionation of soil Interactive comment

gases by diffusion of water vapor, gravitational settling, and thermal diffusion, Geochim. Cosmochim. Acta, 60: 1005-1018). Any model that hopes to infer the isotopic signatures of sources from field data of CO2 in pore space needs to include the possibility of non-steady state conditions, or at least demonstrate that steady state is a reasonable assumption.

Response: We agree with the reviewer that this a direction for future research (Section 5.1). For this paper, we decided to adhere to a readily understandable steady state model, so that it could be easily understood, solved analytically, and so that the soil 14CO2 user community can discuss, and push this work farther where sensible.

Added on page 16 lines 19-23: "For example, gravitational settling and thermal diffusion can impact the diffusive non-steady state in deep soils (Severinghause et al., 1996) and diurnal temperature cycles (Nickerson and Risk, 2009b; Phillips et al., 2010; Risk and Kellman, 2008) and wind advection (Bowling and Massman, 2011) can be of impact non-steady state in near surface soils. "

The authors are correct that a model such as the one they propose is required to deconvolve what is affecting observed gradients of 14CO2 and 13CO2 in soil pore space. However, it is worth pointing out that the authors have picked a rather extreme condition where the 14C signature of decomposition derived CO2 differs greatly (300 per mille) from the atmospheric value – in many soil profiles this is not the case as the sources of CO2 are dominated by root respiration and decomposition of more labile soil organic matter rather than the decomposition of very old peat material, and the gradients in 14CO2 (and thus the biases) are much smaller. (This of course depends on fA, based on Figure 4 – though we do not know what 13C or [CO2] were for these sites/dates).

Response: If the math and physics are correct when extreme boundary conditions are involved, then they are correct and should be dealt with in all cases.

The authors have assumed a case where both the production rate and the radiocar-
bon content of CO2 produced by decomposition are constant with soil depth, whereas that is also not normally the case (production declines exponentially with depth, and radiocarbon tends to decline linearly). Why not test with a more realistic model?

Response: Our focus is not to represent every condition possible, but to highlight the problem. Hopefully others will investigate their own situations appropriately in the future.

I do not accept the statement at the end of section 5.1 ("Our calculated error values are therefore conservative, as these non-equilibrium processes will only add error and uncertainty") without some demonstration of how sensitive the biases are to the assumptions. Also, I think the statements about the size of the potential error ("there could be error of 100 %) are a bit too specific to the case investigated. Certainly, there are biases, but perhaps these could be expressed as something a percentage of the difference between the 14C of sources and that of overlying air (in the most extreme case here, 30 per cent, or 1/3 of (-200-100). As noted above, more realistic simulations would likely estimate smaller absolute values, but not necessarily percentages, as these are based on the 13C differences.

Response: Thank you for this comment. Our text could be improved here to add specificity. We were not intending to be overly definitive, but to communicate general learnings from previous studies, and useful examples so as to illustrate scale of effects. To address this we have added the following on lines 19-28 of page 16: "For example, gravitational settling and thermal diffusion can impact the diffusive non-steady state in deep soils (Severinghause et al., 1996) and diurnal temperature cycles (Nickerson and Risk, 2009b; Phillips et al., 2010; Risk and Kellman, 2008) and wind advection (Bowling and Massman, 2011) can be of impact non-steady state in near surface soils. The error values presented here are therefore likely conservative, since previous  $\delta$ 13C studies have shown that these non-equilibrium processes generally add fractionation uncertainty (Nickerson and Risk, 2009b; Phillips et al., 2000; Phillips et al., 2
tionation uncertainty (Egan et al., 2014) that may need to be evaluated. The real magnitude of error will be situationally specific, and sensitive to the soil conditions and sampling methodology."

Yes, the potential error is specific to our case ("there could be error of 100 permil") because we did need to use an example. To be clearer on this point, we have added to the text on lines 10-13 on page 16: "In our case here, we observed Method 1 error to be as low as 10 % (2 to 2.5 year age error) in February and up to 100 % (18-25 year age error) at depth in May, although the magnitude of expected error is variable and situationally specific as shown in Fig. 3 and 5.

The "field experiment" and the use of models to interpret these data are not well enough described for the reader to understand. Were the same assumptions (constant production and 'effective' diffusion with depth) applied for interpreting these results? Either this needs to be more thoroughly described (for example by including the CO2 and 13CO2 profiles) or removed from the paper, as it does not really fit with the rest.

Response: The field data was used in this manuscript as an example of the difference in using old and new reporting style. It was not meant to be an in depth field case.

Overall, the authors need to do more sensitivity tests with their 'thought experiment" models; if they use the field data it needs to be much better described.

Response: The thought experiment model was also tested across the ranges described in section 3.1. The field data was not used in the model work. Figures 1,2 and 3 show the sensitivity of the model across a wide range of diffusivity and production rates. Figures 2 and 3 also demonstrate that our proposed solution, Method 2, is insensitive to soil parameters, as it worked across the ranges of parameters tested. We have added text on lines 1-2 of page 14 to make this clear in more than just a graphical sense as shown in the figures.

Lines 10-15 in the Introduction are somewhat misleading. The 'traditional' way to report

**BGD**
14C values from CO2 in soil air would use the 13C signature of the CO2 that was sampled. This value is important as it by definition would include both the biochemical and physical (diffusion) effects in a system where mixing is not important. As noted above, the reporting of the data should not be confounded with the interpretation of the data.

Response: As we noted above, mixing occurs by diffusion too.

Page 4, line 16 – the estimation of the isotopic effect is based on comparing the reduced masses of the 12C-air, 13C-air, or 14C-air system, not just the square root of the masses of the isotopic species of CO2 alone (air has a 'mass' of 28). This is unclear from what is written. Response: We agree. We have added a citation on Page 5 lines 21-22 for Cerling et al. 1991 who describes in detail with appropriate references.

Most radiocarbon is measured directly these days as isotope ratios, rather than activities as presented in equation (5). Perhaps this nomenclature could be updated. While  $\Delta$ 14C is the  $\delta$ 13C-corrected version of  $\delta$ 14C as stated in line 24 on page 5, both of these include an additional correction for the decay of the standard since 1950. (This is actually correct in the formula stated by the authors, which uses Aabs , but could be really confusing for the reader who does not know what Aabs is and how it differs from the equations (5) that refer only to the activity of oxalic acid. This correction is not negligible, the oxalic acid standard has decreased by about 10 ‰ since 1950. (The same is not true for Fraction Modern or D14C, which are not expressed relative to Aabs , because the standard and sample are presumed to decay at the same rate). Because of this, the authors should really state the assumed year of sampling/measurement in their simulations (as noted above,  $\Delta$ 14C and  $\delta$ 14C will differ slightly depending on the year of measurement/sampling).

Response: We thank the reviewer for this comment. We found that while reading the foundational papers on radiocarbon reporting, the description of terms and equations lacked sufficient detail, and we were at times unclear on all the different terms. We
believe that as we build a new foundation for interpreting radiocarbon of the soil gas environment, it is important that we're very clear on nomenclature. When we use our proposed solution we assume that year is present day, because there is no oxalic acid correction in our form of  $\Delta$ 14C (added this on line 8-9 page 11).

The current  $\triangle$ 14C of atmospheric CO2 is closer to 10 ‰ than 100 ‰.

Response: Thank you for pointing this out. The model works with any atmospheric value for  $\Delta$ 14C.

One problem with the "new" approach is that one must assume that they know the  $\delta$ 13C signature of the CO2 being produced – what is the sensitivity to getting that value incorrect by 1-2 ‰

Response: We agree that this could be problematic for back-correcting older data, but moving forward if researchers are going to measure the radiocarbon signature of soil CO2 at different depths to infer production, they should also measure  $\delta$ 13C (added text on lines 2-5 of page 20 to address this.) There is a high sensitivity to the  $\delta$ 13C signature of production. As shown in Fig. 2 and Section 2, where we had to include all decimal places for  $\delta$ 13Cs and  $\delta$ 14Cs from the model to input into the Davidson (1995)  $\delta$ J, in order to get the true value of production. We've added text on line 5 and page 20 to address this. However, if  $\delta$ J is used to calculate the  $\delta$ 13C signature of the CO2 being produced, using  $\delta$ 13C of soil CO2 from any depth, it should still yield the true radiocarbon composition of production, as in shown in Fig. 3.

Table 2 compares with "Phillips et al 2013" but that is first mentioned in the Discussion – it should be explained in the methods or the introduction if used.

Response: On Page 13 line 4-6 we added the following: In Section 5.2 we compare these two Methods to that used in Phillips et al. (2013), which we will call the Phillips2013 Method. Phillips et al. (2013) use a different gradient approach (not Eq. 16;  $\Delta_J^{14}$ ) along with  $\Delta$ 14Cold to calculate the radiocarbon composition of producBGD
tion.

Page 12, line 20. Presumably the authors are pointing out that the movement of air in dynamic chambers could cause some kind of enhanced exchange with soil pore space air – where does the 200 ‰ number come from? – is there a reference for this? Wouldn't mixing also mean that in this case the 'standard' 13C correction would not be appropriate for interpreting the 14C of the measured flux?

Response: The value for 200 ‰ is referenced as coming from the Egan et al. 2014 study. As mentioned on page 17 lines 9-14 the 'standard' 13C correction actually works in the case of chamber-based measurements for flux ("...Method 1 is actually acceptable for use in the case of surface flux chambers, because unlike soil CO2 which will always differ from soil production soil-respired CO2, conservation of mass dictates that isotopic values of flux must represent soil production so long as the soil is in steady-state (Cerling et al., 1991).").

Page 12, line 32. "Changes in oxygen availability will also affect fractionations related to the rate of production. " This statement needs a reference. What the authors mean here is that the source of C might change as might any fractionation associated with respiration under low O2. However, the isotopic signatures of CO2 that accumulate in the chamber will still represent that source without need for additional corrections.

Response: We agree with the reviewer, so we have removed the following statement from the top of page 18: "Changes in oxygen availability will also affect fractionations related to the rate of production."

Page 14, lines 5-10. AMS laboratories measure all three isotopes, but the 13C they measure includes fractionation processes in the sample preparation and measurement (all of which are correctable using mass-dependent assumptions). However, the AMS-measured  $\delta$ 13C is not a good measure compared to the 13C of the CO2 that is measured with an IRMS. Therefore, the labs cannot easily report the  $\Delta$ 14C as the authors propose unless the user has independently supplied the  $\delta$ 13C values for their sam-
ples. Instead I would advise people to 'uncorrect' the reported data themselves (and give them the formula). The current guidelines are to use Fraction Modern and report the year of collection and the year of measurement so that other ways of expressing 14C can be calculated. Also, AMS labs do not measure activity, they measure isotope ratios.

Response: We agree with this. On lines 20-23 of page 19 we give the formula for how people can "uncorrect" their already reported data: "...1) use  $\delta$ 13s and  $\Delta$ 14Cold to back out the activity of the sample (As); 2) calculate the isotopic composition of production for  $\delta$ 13C using Eq. (2),  $\delta$ J13; 3) use  $\delta$ J13 and As in Eq. (7) to calculate  $\Delta$ 14Cnew, and finally 4) determine the radiocarbon isotopic composition of production, using Eq. (16),  $\Delta$ J14." This is what we did with our already reported data in Fig. 4 and Section 4.2. We also added the following text on lines 23-25 of page 19: This solution assumes that the researcher has measured a value of  $\delta$ 13s independent from  $\Delta$ 14Cold, as the AMS measured  $\delta$ 13 is not comparable to Isotope-Ratio Mass Spectrometry (IRMS) measured  $\delta$ 13.

Please also note the supplement to this comment: https://www.biogeosciences-discuss.net/bg-2018-451/bg-2018-451-AC3supplement.pdf

**BGD**

---

## Author Response (AR1)

Manuscript bg-2018-451 Title: Isotopic fractionation corrections for the radiocarbon composition of CO2 in the soil gas environment must include diffusion and mixing Authors: Jocelyn Egan et al.

Thanks to the referees and the editor for their helpful comments. We apologize for previous insufficient responses. We appreciate all the comments and believe that they have now led to a much-improved manuscript. Referee comments are listed below in bold, and our responses follow each.

Sincerely, Jocelyn Egan (for all authors)

**Referee #1**

The reviewed manuscript proposed new corrections for 14C measurements in soil air, to account for diffusion and mixing. Experimental study, and modeling are used to illustrate the point, and recommendation for best practice are given. The logic of the authors is clear, and usually the manuscript is well written.

Thanks for the supportive comments.

The authors writes that the problem with the traditional method of 13C based correction, is that it works only for biological process like photosynthesis and do not apply for example to fractionation in diffusion. This is not correct. The correction based on 13C should work for any mass-dependent fractionation (i.e most fractionation processes). However, they are correct the traditional correction will not work for mixing. Because this is not a fractionating processes and no mass-dependent can be assumed. This error should be corrected in the text, abstract, and in the title (i.e. correction for mixing is not correction for fractionation).

Thank you to the reviewer for pointing this out. We agree, and think that we have not been clear with our terminology throughout the text. Although mixing might not be considered a traditional fractionating process, there are important isotopic effects associated with mixing. An isotopic fractionation is defined as a difference in the isotopic composition between a reactant and a product. For the application here with radiocarbon, the radiocarbon composition of biological respiration is the "reactant" and the soil gas is the "product". We must take into account all isotopic effects that influence the product in order to understand the true isotopic composition of the reactant. Therefore both diffusion and diffusive mixing need to be included and accounted for and that is the central message of our paper.

- In 22 instances in the text we have changed "mixing" to "diffusive mixing".
- We have also removed the word "fractionation" throughout the text when it was used in association with mixing.
- On Page 6, line 14 we also added the following highlighted text: "This traditional approach uses  $\delta^{13}$ C as an input parameter to make a mass-dependent correction to obtain  $\Delta^{14}$ C, but the profiles of  $\delta^{13}$ C and  $\delta^{14}$ C of soil CO2 (Fig. 1) highlight that both vary within the soil because of diffusion and diffusive mixing. This makes it unclear what form of  $\delta^{13}$ C should actually be used in the correction in the soil gas environment ( $\delta^{13}$ C of the soil CO2 is measured, but  $\delta^{13}$ C of biological production is not) as diffusive mixing is not a mass-dependent process."

Another point that can be mentioned in the text: In soils that contain carbonates,

**isotopic exchange of CO2 with the carbonates can introduce additional error which is not mass-dependent.**

We thank the reviewer for this comment and we have added the following statement in the Discussion Section on page 12 to address this: "This analysis does, however, have limitations. The synthetic soil environment is simplified in our approach, and in reality soil diffusion and production rates are not constant through depth. The model is necessarily simplified so that it can be easily solved analytically, but in reality soils are typically not in steady state (van Asperen et al., 2017; Bowling et al., 2009; Bowling and Massman, 2011; Goffin et al., 2014; Maier et al., 2010; Moyes et al., 2010; Nickerson and Risk, 2009b; Risk and Kellman, 2008). For example, gravitational settling and thermal diffusion can impact the diffusive non-steady state in deep soils (Severinghause et al., 1996) and diurnal temperature cycles (Nickerson and Risk, 2009b; Phillips et al., 2010; Risk and Kellman, 2008) and wind advection (Bowling and Massman, 2011) can be of impact non-steady state in near surface soils. The error values presented here are therefore likely conservative, since previous  $\delta^{13}$ C studies have shown that these nonequilibrium processes generally add fractionation uncertainty (Nickerson and Risk, 2009b; Phillips et al., 2010; Risk and Kellman, 2008). Additionally, sampling methods may impose an additional layer of non-equilibrium fractionation uncertainty (Egan et al., 2014) that may need to be evaluated. The real magnitude of error will depend on the given soil environment and will be sensitive to the soil conditions and sampling methodology. Additionally, carbonate soils could introduce more error, as the isotopic exchange between soil gas and carbonates is not mass-dependent (Breecker et al., 2009)."

It is stated that: "Graham's law of effusion (rate of diffusion ~ 1/sqrt(mass)) provides us with an expected difference in diffusion rate of isotopologues." In fact, as it is written few lines before, this is a binary diffusion of CO2 in air and hence binary diffusion equation (a one that invokes reduce-mass of CO2 and air) should be used.

Thank you to the reviewer for pointing this out. We agree and have modified the text to say "binary diffusion equation" rather than "Graham's law of effusion" (see line 16 on page 4).

**Minor comment: Please give a better description of the soil than just "clay".**

Thank you for pointing out the vagueness of "clay" as a soil descriptor. The following detail has been added on page 8: "These sites were located in agricultural fields, with soils characterized as a combination of Estevan and Roughbark souls. There soils have sandy loam surfaces, with thin organic layers and soil layers that are poorly drained and have high clay content (SCSR, 1997)." In Table 2 and 3 we also included modeled examples of 3 potential soil types, including clay, based on general soil production and diffusion rates associated with those soils.

**Referee #2**

The paper proposes new corrections for the radiocarbon composition of  $CO_2$  in the soil gas environment. The paper describes in detail how the authors have derived these new corrections to include diffusion and mixing in their approach. The paper should revive and encourage further discussions on this important but somewhat under explored topic. The topic therefore up to now has remained only partly resolved. Even the new corrections proposed by in the current papers have certain limitations, which the authors rightly highlight in the discussion of the paper.

Thanks for the positive comments.

Whereas, the isotope corrections are described in great detail, the field profile study is somewhat lacking in such details, especially the description of the site and soil used is rather scant. Please enhance this section.

We thank the reviewer for this comment and agree that we have not been detailed enough explaining both the point of including the field data, and we have not been thorough enough in our description of the fieldsite. In this paper we did not intend to use the field data as validation for the method, but as an example of how the new theory could be applied.

We added the following details about the soil and field methods (page 8): "To demonstrate our new approach with real data, we used  $\Delta^{14}$ C values collected in the field. We collected samples of soil CO2 at 3 depths from the soil profile and from the air just above the soil, from 3 sites in Weyburn, Saskatchewan, for both stable and radio-isotope analysis. These sites were located in agricultural fields, with soils characterized as a combination of Estevan and Roughbark souls. There soils have sandy loam surfaces, with thin organic layers and soil layers that are poorly drained and have high clay content (SCSR, 1997)."

We also added more detail about how we used the field data to apply our new proposed correction on page 9: "We used the  $\Delta^{14}$ C values reported to us from the AMS laboratory along with  $\delta^{13}$ C (presented in Risk. et al. 2013) to back-correct the values so that we could present the results using  $\Delta^{14}$ Cnew (Eq. 15) and then also calculate the radiocarbon composition of biological production using  $\Delta_{I}^{14}$  (Eq.16)."

Furthermore, as the topic of the paper is rather specific, e.g. how and if the way the various formula's relate to each other is correct, some specific detailed isotope/radiocarbon expertise is needed to verify this.

We thank the reviewer for this comment and hope that the edits we have made based on the 3 referees' comments as well as the editor's comments will reinforce that the work we did is correct.

In a more general sense the paper can be published with minor revision also in light in the perception that it creates a focal point for further discussion around this paper on such isotope fractionation correction for radiocarbon (14C) measurements.

Thanks, we would also like to see it published.

**Referee #3**

Egan et al. correctly state that radiocarbon corrections based on 13C cannot be used to interpret radiocarbon data if there are processes involved that involve process that are not mass-dependent, like mixing. They use simple 1D models to show the potential biases in estimating the radiocarbon signature of source gases if gases in soil air space are interpreted without understanding that soil air both mixes and diffuses. The main advance here is that the authors use information on 13CO2 in pore space to estimate the mixing, which in turn allows a better way to estimate the 14C of CO2 sources.

Thank you for this supportive comment. We agree and feel this is an important advance.

In some sense, the authors have set up Stuiver and Polach (1977) as a target to shoot down in a way that is not entirely fair. Stuiver and Polach is a paper that sets out the conventions for reporting radiocarbon data – and only that. It is up to the investigator to interpret the radiocarbon data appropriately – including in the use of models such as those proposed in this paper. I would therefore not say that "their approach is wrong" as it was never their intention to tackle the interpretation of isotope profiles in soils. The authors need to be clear that the data should be correctly reported – i.e. according to Stuiver and Polach – but that to interpret soil  $CO_2$  isotopic data requires the use of a model that includes transport and mixing processes.

We thank the reviewer for this comment. We agree that we were not clear enough. Our intention was not to target Stuiver and Polach (1977), but to highlight problems that have cropped up in the literature with regard to misinterpretation and misuse. We removed the word "convention" throughout the text, when what we really meant was that the mass-dependent correction did not account for diffusion and diffusive mixing in the soil gas environment (usually replaced with the word "correction"). We have also modified the text in the following places to highlight the importance of interpretation of the data as the referee suggests:

- One page 2 we removed the sentence that has a strike through it and added the following highlighted words: "Overall, soil researchers have two implicit expectations, which is that physical fractionations are present, and that biochemical fractionations are absent. These expectations seem clearly out of step with the classical reporting conventions for 14C that compensates only for biochemical fractionation. To date, no study has examined the conventional radiocarbon corrections with these implicit expectations in mind. We believe the traditional correction should be re-evaluated for soil gas studies."
- On page 10 we added the highlighted word in the following: "In Fig. 3 two depth profiles are plotted for each model iteration, one where the model output was used to calculate the conventional representation of radiocarbon,  $\Delta^{14}C_{old}$  (Eq. (7); solid line) and the second where the output was used to calculate  $\Delta^{14}C_{new}$ , our proposed convention for presenting radiocarbon soil CO2 (dashed line)."

- On page 13 we added the following text, where we wanted to emphasize interpreting the data incorrectly: "This isotopic difference is not large, but it still does not follow theory, so we also compared the difference between using  $\Delta^{14}C_{new}$ and  $\Delta^{14}C_{old}$  to interpret the isotopic composition of soil CO2 in these same soil environments in Table 2. In the three soil types modelling, the bias in using  $\Delta^{14}C_{old}$  to interpret soil CO2 was always larger near the surface, and was largest in sandy type soils with mid-range production rates and high soil diffusion rates."

**Does anyone in the literature actually assume Method 1?**

We do not think it is commonplace for researchers to assume Method 1, however we feel that it's important to get this issue out in the open so that people can consider it, and we have tried to present this as a way forward rather than targeting particular studies. We have added the following highlight text on page 9 to address this:

"Method 1 follows the traditional correction, where Eq. (7) was used to calculate  $\Delta^{14}C_{old}$ , and we interpret this value as the radiocarbon composition of CO2 from biological production. Although this interpretation may not seem commonly used, as most people now understand the soil CO2 and soil-respired CO2 differ, we want to use it here as an example. If a researcher were to interpret a soil CO2 radiocarbon measurement as old and representative of the end-member source it came from, as demonstrated in Section 2, this soil CO2 sample will not necessarily be representative of the end-member production source because of gas transport mechanisms."

The abstract states that diffusion and mixing are both problematic. For pure molecular diffusion, the fractionation should be mass dependent (i.e. not problematic). Mixing, however, will not be mass-dependent and therefore cause problems. However, by lumping both into "diffusion" in their model, the authors are confounding things. I think what they are calling "D" in their model is really an "effective" diffusivity. This would effectively mean adding 4.4 per mille to the 13C of the source in equation 15 (which then might explain differences with Phillips2013?)

We thank the reviewer for this comment as it has made it clear to us that we have not been clear enough with terminology throughout the paper. The model used is a steadystate diffusion model and it does not include advection or dispersion (or use "effective" diffusivity like that used in the following paper: D.R. Bowling and W.J. Massman, 2011, Pesistent wind-induced enhancement of diffusive  $CO_2$  transport in a mountain forest snowpack, Journal of Geophysical Research, Vol. 116, G04006). However, mixing does of course occur via diffusion, and given boundary conditions of atmospheric  $CO_2$  on one end and biologically-produced  $CO_2$  on the other, with isotopic differences between them, the soil gas will be a diffusively-mixed environment. As pointed out by the referee 3 in the earlier comments above, it's the interpretation that is important. Our main point is to highlight that interpreting the isotopic composition of soil CO2 without appropriately accounting for the isotopic processes that affect it will lead to error.

As mentioned in our response to Referee #1, although mixing might not be considered a traditional fractionating process, there are important isotopic effects associated with mixing. An isotopic fractionation is defined as a difference in the isotopic composition between a reactant and a product. For the application here with radiocarbon, the radiocarbon composition of biological respiration is the "reactant" and the soil gas is the "product". We must take into account all isotopic effects that influence the product in order to understand the true isotopic composition of the reactant. Therefore both diffusion and diffusive mixing need to be included and accounted for and that is the central message of our paper. We have edited the text in 22 instances to say "diffusive mixing" rather than "mixing".

In very deep soils, one needs to worry about additional issues that the authors have not included, such as gravitational and thermal effects associated with non-steady state conditions. These issues are treated in Severinghaus et al. 1996 (Fractionation of soil gases by diffusion of water vapor, gravitational settling, and thermal diffusion, Geochim. Cosmochim. Acta, 60: 1005-1018). Any model that hopes to infer the isotopic signatures of sources from field data of CO2 in pore space needs to include the possibility of non-steady state conditions, or at least demonstrate that steady state is a reasonable assumption.

We agree with the reviewer that this a direction for future research. For this paper, we decided to adhere to a readily understandable steady state model, so that it could be easily understood, solved analytically, and so that the soil 14CO2 user community can discuss, and push this work farther where sensible.

We have added following text on page 12 to clearly address potential bias in ignoring non-steady state conditions: "This analysis does, however, have limitations. The synthetic soil environment is simplified in our approach, and in reality soil diffusion and production rates are not constant through depth. The model is necessarily simplified so that it can be easily solved analytically, but in reality soils are typically not in steady state (van Asperen et al., 2017; Bowling et al., 2009; Bowling and Massman, 2011; Goffin et al., 2014; Maier et al., 2010; Moyes et al., 2010; Nickerson and Risk, 2009b; Risk and Kellman, 2008). For example, gravitational settling and thermal diffusion can impact the diffusive non-steady state in deep soils (Severinghause et al., 1996) and diurnal temperature cycles (Nickerson and Risk, 2009b; Phillips et al., 2010; Risk and Kellman, 2008) and wind advection (Bowling and Massman, 2011) can be of impact non-steady state in near surface soils. The error values presented here are therefore likely conservative, since previous  $\delta^{13}$ C studies have shown that these non-equilibrium processes generally add fractionation uncertainty (Nickerson and Risk, 2009b; Phillips et al., 2010; Risk and Kellman, 2008). Additionally, sampling methods may impose an additional layer of non-equilibrium fractionation uncertainty (Egan et al., 2014) that may

need to be evaluated. The real magnitude of error will depend on the given soil environment and will be sensitive to the soil conditions and sampling methodology."

The authors are correct that a model such as the one they propose is required to deconvolve what is affecting observed gradients of 14CO2 and 13CO2 in soil pore space. However, it is worth pointing out that the authors have picked a rather extreme condition where the 14C signature of decomposition derived CO2 differs

greatly (300 per mille) from the atmospheric value – in many soil profiles this is not the case as the sources of  $CO_2$  are dominated by root respiration and decomposition

of more labile soil organic matter rather than the decomposition of very old peat material, and the gradients in 14CO2 (and thus the biases) are much smaller. (This

of course depends on fA, based on Figure 4 – though we do not know what  ${}^{13}C$  or  $[CO_2]$  were for these sites/dates).

Thank you to the reviewer for pointing this out. We agree that the value of atmospheric  $\Delta^{14}$ C is not common. We have changed the figures and text to address this.

The following text was added on pages 7 and 8 to reiterate that our proposed solution was tested under a range of values for  $\Delta^{14}$ C of production:

"The analytical gas transport model was applied across a range of soil diffusivity  $(1 \times 10^{-7}, 1 \times 10^{-6} \text{ and } 1 \times 10^{-5} \text{ m}^2 \text{ s}^{-1})$ , soil production rates (0.5, 1, 2, and 4 µmol CO2 m-3 s-1),  $\delta^{13}$ C of biological production (- 30 ‰, -20 ‰, -15 ‰), and  $\Delta^{14}$ C of biological production (-500 ‰, -200 ‰, -1 ‰, 1 ‰, 200 ‰, 500 ‰). In addition it used  $\Delta^{14}$ C of atmospheric CO2 ( $\Delta_a$ ; 10 ‰) and  $\delta^{13}$ C of atmospheric CO2 ( $\delta_a$ ; -8 ‰), representing realistic conditions found in nature. The other model boundary conditions were as follows: L = 0.8 m, z = 0.025 m, and  $Conc_{atm} = 15833$  µmol m-3 (~380 ppm). The output of the model under these applied conditions were profiles of 12CO2, 13CO2, and 14CO2 for each depth (z) down to the bottom boundary (L)."

Based on your comments, for the updated Figures and Tables, we used a value of  $\Delta^{14}$ C of atmospheric CO2 of 10 ‰ and a  $\Delta^{14}$ C of biological production of -50 ‰, so the range between the values is not as extreme. We agree that these values are more representative of soils primarily dominated by the decomposition of relatively young C sources.

**The authors have assumed a case where both the production rate and the radiocarbon content of CO2 produced by decomposition are constant with soil**

depth, whereas that is also not normally the case (production declines exponentially with depth, and radiocarbon tends to decline linearly). Why not test with a more realistic model?

We thank the reviewer for this comment. Our focus here was not to represent every condition possible, but to highlight the problem. Hopefully others will investigate their

own situations appropriately in the future. We added the following text on page 12 to highlight the limitation of our model: "This analysis does, however, have limitations. The synthetic soil environment is simplified in our approach, and in reality soil diffusion and production rates are not constant through depth. The model is necessarily simplified so that it can be easily solved analytically, but in reality soils are typically not in steady state (van Asperen et al., 2017; Bowling et al., 2009; Bowling and Massman, 2011; Goffin et al., 2014; Maier et al., 2010; Moyes et al., 2010; Nickerson and Risk, 2009b; Risk and Kellman, 2008). For example, gravitational settling and thermal diffusion can impact the diffusive non-steady state in deep soils (Severinghause et al., 1996) and diurnal temperature cycles (Nickerson and Risk, 2009b; Phillips et al., 2010; Risk and Kellman, 2008) and wind advection (Bowling and Massman, 2011) can be of impact non-steady state in near surface soils. The error values presented here are therefore likely conservative, since previous  $\delta^{13}$ C studies have shown that these non-equilibrium processes generally add fractionation uncertainty (Nickerson and Risk, 2009b; Phillips et al., 2010; Risk and Kellman, 2008). Additionally, sampling methods may impose an additional layer of non-equilibrium fractionation uncertainty (Egan et al., 2014) that may need to be evaluated. The real magnitude of error will depend on the given soil environment and will be sensitive to the soil conditions and sampling methodology."

I do not accept the statement at the end of section 5.1 ("Our calculated error values are therefore conservative, as these non-equilibrium processes will only add error and uncertainty") without some demonstration of how sensitive the biases are to the assumptions. Also, I think the statements about the size of the potential error ("there could be error of 100 ‰") are a bit too specific to the case investigated. Certainly, there are biases, but perhaps these could be expressed as something a percentage of the difference between the 14C of sources and that of overlying air (in the most extreme case here, 30 per cent, or 1/3 of (-200-100). As noted above, more realistic simulations would likely estimate smaller absolute values, but not necessarily percentages, as these are based on the 13C differences.

Thank you for this comment. Our text could be improved here to add specificity. We were not intending to be overly definitive, but to communicate general learnings from previous studies, and useful examples so as to illustrate scale of effects.

To aid in demonstrating the sensitivity of our proposed correction method in Figure 3, we have remade Table 2. The Table demonstrates the range in bias in misinterpreting  $\Delta^{14}C_{old}$  as both the value for soil CO2 and soil production at two depths and over three different potential soil environments.

Table 2. Bias from interpreting the radiocarbon isotopic composition of soil CO2 and biological production using  $\Delta^{14}$ Cold. The depth profiles used to calculate the bias was generated from a model with a  $\Delta^{14}$ C of production of -50 ‰ and combinations of diffusion and production rates to represent three different general soil types: clay (D = 1e-7 m2 s-1: P = 0.5 µmol m-3 s-1), loam (D = 1e-6 m2 s-1: P = 4 µmol m-3 s-1), and sand (D = 1e-5 m2 s-1: P = 2 µmol m-3 s-1). All bias values are absolute.

| Soil Type                                                                               | Depth
(cm) | Δ 14 Cold
(‰) | $\Delta_J^{14} \text{ from} \\ \Delta_J^{14} C_{old} \\ (\%_0)$ | Δ 14 C new (%)) | $\Delta_J^{14}$ from
$\Delta^{14}C_{new}$ (%0) | Δ 14 C old - Δ 14 C new
(‰) | $-50 - \Delta_J^{14} \text{ from } \Delta^{14} C_{old}$ (‰) |
|-----------------------------------------------------------------------------------------|---------------|-----------------------------|-----------------------------------------------------------------|---------------------------------------|---------------------------------------------------|----------------------------------------------------------------------------|-------------------------------------------------------------|
| Clay                                                                                    | 2.5           | -43.0                       | -58.1                                                           | -31.7                                 | -50.0                                             | 11.3                                                                       | 8.1                                                         |
| D = 1e-7 m 2 s .1
P = 0.5 μmol m .3 s .1 | 50            | -49.5                       | -58.3                                                           | -40.8                                 | -50.0                                             | 8.6                                                                        | 8.3                                                         |
| Loam                                                                                    | 2.5           | -41.5                       | -58.1                                                           | -29.6                                 | -50.0                                             | 11.9                                                                       | 8.1                                                         |
| D = 1e-6 m 2 s -1
P = 4 µmol m -3 s -1   | 50            | -49.3                       | -58.3                                                           | -40.6                                 | -50.0                                             | 8.7                                                                        | 8.3                                                         |
| Sand                                                                                    | 2.5           | -3.9                        | -57.0                                                           | 24.9                                  | -50.0                                             | 28.8                                                                       | 7.0                                                         |
| $D = 1e-5 m^2 s^{-1}$                                                                   | 50            | -38 5                       | -58.0                                                           | -25 3                                 | -50.0                                             | 13.2                                                                       | 8.0                                                         |
| $P = 2 \mu mol m^{-3} s^{-1}$                                                           | 50            | -30.3                       | -50.0                                                           | -23.3                                 | -30.0                                             | 13.2                                                                       | 0.0                                                         |

The "field experiment" and the use of models to interpret these data are not well enough described for the reader to understand. Were the same assumptions (constant production and 'effective' diffusion with depth) applied for interpreting these results? Either this needs to be more thoroughly described (for example by including the CO2 and 13CO2 profiles) or removed from the paper, as it does not recelly fit with the rest

**really fit with the rest.**

We are sorry that we were not more specific. We have added the following text on page 9 to explain how we used the field results to give an example of how to apply our solution: "We used the  $\Delta^{14}$ C values reported to us from the AMS laboratory along with  $\delta^{13}$ C (presented in Risk. et al. 2013) to back-correct the values so that we could present the results using  $\Delta^{14}$ Cnew (Eq. 15) and then also calculate the radiocarbon composition of biological production using  $\Delta_I^{14}$  (Eq.16)."

**Overall, the authors need to do more sensitivity tests with their 'thought experiment' models; if they use the field data it needs to be much better described.**

We apologize for not explaining our sensitivity analysis better.

We have added text to the following paragraph on pages 7 and 8 to show the ranges of parameters that our solution was tested across:

"The analytical gas transport model was applied across a range of soil diffusivity (1x10-7, 1x10-6 and 1x10-5 m2 s-1), soil production rates (0.5, 1, 2, and 4 µmol CO2 m-3 s-1),  $\delta^{13}$ C of biological production (- 30 ‰, -20 ‰, -15 ‰), and  $\Delta^{14}$ C of biological production (-500 ‰, -200 ‰, -1 ‰, 1 ‰, 200 ‰, 500 ‰). In addition it used  $\Delta^{14}$ C of atmospheric CO2 ( $\Delta_a$ ; 10 ‰) and  $\delta^{13}$ C of atmospheric CO2 ( $\delta_a$ ; -8 ‰), representing realistic conditions found in nature. The other model boundary conditions were as follows: L = 0.8 m, z = 0.025 m, and  $Conc_{atm} = 15833$  µmol m-3 (~380 ppm). The output of the model under these applied conditions were profiles of 12CO2, 13CO2, and 14CO2 for each depth (z) down to the bottom boundary (L). "

We have also remade Table 2, as mentioned in a response above, and created Table 3 to also demonstrate the sensitivity of the  $\delta_I^{13}$  value used in  $\Delta^{14}C_{new}$  and  $\Delta_I^{14}$  calculations.

Table 3. Sensitivity of the  $\delta_J^{13}$  value in  $\Delta^{14}C_{new}$  and  $\Delta_J^{14}$  calculations. The depth profiles used to calculate the bias from using the wrong  $\delta_J^{13}$  value was generated from a model with a  $\Delta^{14}C$  of production of -50 ‰ and combinations of diffusion and production rates to represent three different general soil types: clay (D = 1e-7 m2 s-1: P = 0.5 µmol m-3 s-1), loam (D = 1e-6 m2 s-1: P = 4 µmol m-3 s-1), and sand (D = 1e-5 m2 s-1: P = 2 µmol m-3 s-1). All bias values are absolute.

| Soil Type                                       | Soil Type Depth
(cm) |      | Bias using wrong $\delta_J^{13}$ in $\Delta^{14}C_{new}$ (%) | Bias using $\Delta^{14}C_{new}$ with wrong $\delta_J^{13}$ to calculate $\Delta_J^{14}$ (‰) |
|-------------------------------------------------|-------------------------|------|--------------------------------------------------------------|---------------------------------------------------------------------------------------------|
| Clay                                            | 2.5 cm                  | -0.5 | 1.0                                                          | 1.0                                                                                         |
| $D = 1e-7 m^2 s^{-1}$                           |                         | +0.5 | 1.0                                                          | 1.0                                                                                         |
| P = 0.5 µmol m -3 s -1    |                         | -1   | 2.0                                                          | 2.0                                                                                         |
|                                                 |                         | +1   | 1.9                                                          | 1.9                                                                                         |
|                                                 |                         | -5   | 10.0                                                         | 9.8                                                                                         |
|                                                 |                         | +5   | 9.9                                                          | 9.7                                                                                         |
|                                                 | 50 cm                   | -0.5 | 1.0                                                          | 1.0                                                                                         |
|                                                 |                         | +0.5 | 1.0                                                          | 1.0                                                                                         |
|                                                 |                         | -1   | 2.0                                                          | 2.0                                                                                         |
|                                                 |                         | +1   | 2.0                                                          | 1.9                                                                                         |
|                                                 |                         | -5   | 9.9                                                          | 9.8                                                                                         |
|                                                 |                         | +5   | 9.8                                                          | 9.7                                                                                         |
| Loam                                            | 2.5 cm                  | -0.5 | 1.0                                                          | 1.0                                                                                         |
| $D = 1e-6 m^2 s^{-1}$                           |                         | +0.5 | 1.0                                                          | 1.0                                                                                         |
| $P = 4 \mu mol m^{-3} s^{-1}$                   |                         | -1   | 2.0                                                          | 2.0                                                                                         |
|                                                 |                         | +1   | 2.0                                                          | 1.9                                                                                         |
|                                                 |                         | -5   | 10.0                                                         | 9.8                                                                                         |
|                                                 |                         | +5   | 9.9                                                          | 9.7                                                                                         |
|                                                 | 50 cm                   | -0.5 | 1.0                                                          | 1.0                                                                                         |
|                                                 |                         | +0.5 | 1.0                                                          | 1.0                                                                                         |
|                                                 |                         | -1   | 2.0                                                          | 2.0                                                                                         |
|                                                 |                         | +1   | 2.0                                                          | 1.9                                                                                         |
|                                                 |                         | -5   | 9.9                                                          | 9.8                                                                                         |
|                                                 |                         | +5   | 9.8                                                          | 9.7                                                                                         |
| Sand                                            | 2.5 cm                  | -0.5 | 1.1                                                          | 1.0                                                                                         |
| $D = 1e-5 m^2 s^{-1}$                           |                         | +0.5 | 1.1                                                          | 1.0                                                                                         |
| P = 2 $\mu$ mol m -3 s -1 |                         | -1   | 2.1                                                          | 2.0                                                                                         |
|                                                 |                         | +1   | 2.1                                                          | 1.9                                                                                         |
|                                                 |                         | -5   | 10.6                                                         | 9.8                                                                                         |
|                                                 |                         | +5   | 10.4                                                         | 9.7                                                                                         |
|                                                 | 50 cm                   | -0.5 | 1.0                                                          | 1.0                                                                                         |
|                                                 |                         | +0.5 | 1.0                                                          | 1.0                                                                                         |

| -1 | 2.0  | 2.0 |
|----|------|-----|
| +1 | 2.0  | 1.9 |
| -5 | 10.1 | 9.8 |
| +5 | 9.9  | 9.7 |

We have also added the following text on page 14 to address the sensitivity of the  $\delta^{13}$  value: "The back-correcting solution does, however, assume that the researcher has measured a value of  $\delta_s^{13}$  independent from  $\Delta^{14}C_{old}$ , as the AMS measured  $\delta^{13}$  is not comparable to Isotope-Ratio Mass Spectrometry (IRMS) measured  $\delta^{13}$ . To assess the sensitivity of the  $\delta_J^{13}$  value in calculating  $\Delta^{14}C_{new}$  and  $\Delta_J^{14}$  for back corrections, in Table 3 we present the error in using a  $\delta_J^{13}$  value that is wrong by  $\pm 0.5$ , 1, and 5 ‰. If rounded to the nearest ‰ value, the bias associated with using a  $\delta_J^{13} \pm 0.5$ , 1 and 5 ‰ across the three soil types and depths modelled is always 1, 2, and 10 ‰, respectively. The back correction solution can therefore work if a researcher collected  $\delta^{13}C$  measurements independently, although we recommend against using this back correction method for new data ."

Lines 10-15 in the Introduction are somewhat misleading. The 'traditional' way to report  ${}^{14}$ C values from CO2 in soil air would use the  ${}^{13}$ C signature of the CO2 that was sampled. This value is important as it by definition would include both the biochemical and physical (diffusion) effects in a system where mixing is not important. As noted above, the reporting of the data should not be confounded with the interpretation of the data.

Thank you to the reviewer for pointing this out. We agree, and think that we have not been clear with our terminology throughout the text. Although mixing might not be considered a traditional fractionating process, there are important isotopic effects associated with mixing. An isotopic fractionation is defined as a difference in the isotopic composition between a reactant and a product. For the application here with radiocarbon, the radiocarbon composition of biological respiration is the "reactant" and the soil gas is the "product". We must take into account all isotopic effects that influence the product in order to understand the true isotopic composition of the reactant. Therefore both diffusion and diffusive mixing need to be included and accounted for and that is the central message of our paper.

- In 22 instances in the text we have changed "mixing" to "diffusive mixing".
- We have also removed the word "fractionation" throughout the text when it was used in association with mixing.
- On Page 6, line 14 we also added the following highlighted text: "This traditional approach uses  $\delta^{13}C$  as an input parameter to make a mass-dependent correction to obtain  $\Delta^{14}C$ , but the profiles of  $\delta^{13}C$  and  $\delta^{14}C$  of soil CO2 (Fig. 1) highlight that both vary within the soil because of diffusion and diffusive mixing. This makes it

unclear what form of  $\delta^{13}$ C should actually be used in the correction in the soil gas environment ( $\delta^{13}$ C of the soil CO2 is measured, but  $\delta^{13}$ C of biological production is not) as diffusive mixing is not a mass-dependent process."

Page 4, line 16 – the estimation of the isotopic effect is based on comparing the reduced masses of the 12C-air, 13C-air, or 14C-air system, not just the square root of the masses of the isotopic species of CO2 alone (air has a 'mass' of 28). This is unclear from what is written.

We agree. We have added a citation on page 3 for Cerling et al. 1991 who describes in detail with appropriate references.

**Most radiocarbon is measured directly these days as isotope ratios, rather than activities as presented in equation (5). Perhaps this nomenclature could be updated.**

We agree, and we do use isotope ratio further on, but we wanted to present the math as it was originally, to show how we get to the isotope ratios from there.

While  $\Delta^{14}$ C is the  $\delta^{13}$ C-corrected version of  $\delta^{14}$ C as stated in line 24 on page 5, both of these include an additional correction for the decay of the standard since 1950. (This is actually correct in the formula stated by the authors, which uses  $A_abs$ , but could be really confusing for the reader who does not know what  $A_{abs}$  is and how it differs from the equations (5) that refer only to the activity of oxalic acid. This correction is not negligible, the oxalic acid standard has decreased by about 10 ‰ since 1950. (The same is not true for Fraction Modern or  $D^{14}$ C, which are not expressed relative to  $A_{abs}$ , because the standard and sample are presumed to decay at the same rate). Because of this, the authors should really state the assumed year of sampling/measurement in their simulations (as noted above,  $\Delta^{14}$ C and  $\delta^{14}$ C will differ slightly depending on the year of measurement/sampling).

We thank the reviewer for this comment. We found that while reading the foundational papers on radiocarbon reporting, the description of terms and equations lacked detail, and we were at times unclear on all the different terms. We believe that as we build a new foundation for interpreting radiocarbon of the soil gas environment, it is important that we're very clear on nomenclature. When we use our proposed solution we assume that year is present day, because there is no oxalic acid correction in our form of  $\Delta^{14}C$  (added this on page 8).

**The current $\triangle^{14}$ C of atmospheric CO2 is closer to 10 ‰ than 100 ‰.**

Thank you for pointing this out. As mentioned above, we have updated the text as well as all the figures and tables to include modeled profiles using a value of 10 ‰.

**One problem with the "new" approach is that one must assume that they know the $\delta^{13}$ C signature of the CO2 being produced – what is the sensitivity to getting that value incorrect by 1-2 ‰?**

We agree that this could be problematic for back-correcting older data, but moving forward if researchers are going to measure the radiocarbon signature of soil CO2 at different depths to infer production, they should also measure  $\delta^{13}C$ . We added the following text on page 14 to address this: "We also suggest that researchers measure  $\delta^{13}$ alongside  $\Delta^{14}C$ , so that they are not dependent on the AMS measured  $\delta^{13}$  for potential back-corrections, to prevent potential error ranging from 1-10 ‰ (Table 3)."

We also performed some sensitivity analysis of the  $\delta_J^{13}$  value in calculating  $\Delta^{14}C_{\text{new}}$  and  $\Delta_J^{14}$  as mentioned above, which can be found in Table 3.

**Table 2 compares with "Phillips et al 2013" but that is first mentioned in the Discussion – it should be explained in the methods or the introduction if used.**

We updated Table 2 and removed this reference, and to instead just use a general example of the bias associated with using  $\Delta^{14}C_{old}$  to calculate the radiocarbon composition of production using any gradient method.

Page 12, line 20. Presumably the authors are pointing out that the movement of air in dynamic chambers could cause some kind of enhanced exchange with soil pore space air – where does the 200 ‰ number come from? – is there a reference for this? Wouldn't mixing also mean that in this case the 'standard' 13C correction would not be appropriate for interpreting the 14C of the measured flux?

The value for 200 ‰ is referenced as coming from the Egan et al. 2014 study. We have removed the Discussion Section addressing all the different potential additional bias associated with all different methods, thanks to this comment as well as a comment from the Editor. We did however, leave in the section about the flux chambers in the following text on page 12: "Surface flux chambers are commonly used sampling methodology used for measuring the radiocarbon composition of production, and were not addressed in our analysis. However, Method 1 is actually acceptable for use in the case of surface flux chambers, because unlike soil CO2 which will always differ from soil production soil-respired CO2, conservation of mass dictates that isotopic values of flux must represent

soil production so long as the soil is in steady-state (Cerling et al., 1991). While radiocarbon surface flux data need no correction for transport fractionation, researchers should be cautious when using surface flux chambers because they can cause isotopic dis-equilibrium (Albanito et al., 2012; Egan et al., 2014; Midwood and Millard, 2011; Nickerson and Risk, 2009a). As shown in the Egan et al. (2014) study, static chamber methods (i.e. Hahn et al., 2006) and the proposed forced-diffusion chamber technique were the least erroneous for radiocarbon measurements, whereas dynamic chamber sampling techniques (i.e. Gaudinski et al., 2000; Schuur and Trumbore, 2006) could cause up to 200 ‰ bias under certain soil diffusion and production scenarios."

We left this section as we think it is important that the readers understand why the old interpretation correction will still work with this method.

Page 12, line 32. "Changes in oxygen availability will also affect fractionations related to the rate of production." This statement needs a reference. What the authors mean here is that the source of C might change as might any fractionation associated with respiration under low O2. However, the isotopic signatures of CO2

that accumulate in the chamber will still represent that source without need for additional corrections.

As mentioned above, this whole section is now removed from the paper as it was disjointed and did not help.

Page 14, lines 5-10. AMS laboratories measure all three isotopes, but the 13C they measure includes fractionation processes in the sample preparation and measurement (all of which are correctable using mass-dependent assumptions). However, the AMS- measured  $\delta^{13}$ C is not a good measure compared to the 13C of the CO2 that is measured with an IRMS. Therefore, the labs cannot easily report the  $\Delta^{14}$ C as the authors propose unless the user has independently supplied the  $\delta^{13}$ C values for their samples. Instead I would advise people to 'uncorrect' the reported data themselves (and give them the formula). The current guidelines are to use Fraction Modern and report the year of collection and the year of measurement so that other ways of expressing 14C can be calculated. Also, AMS labs do not measure activity, they measure isotope ratios.

We agree with this. In Section 5.1 we give the formula for how people can "uncorrect" their already reported data: "...1) use  $\delta_s^{13}$  and  $\Delta^{14}C_{old}$  to back out the activity of the sample (As); 2) calculate the isotopic composition of production for  $\delta^{13}C$  using Eq. (2),  $\delta_J^{13}$ ; 3) use  $\delta_J^{13}$  and As in Eq. (7) to calculate  $\Delta^{14}C_{new}$ , and finally 4) determine the radiocarbon isotopic composition of production, using Eq. (16),  $\Delta_J^{14}$ ." This is what we did with our already reported data in Fig. 4 and Section 4.2.

**Isotopic fractionation corrections for the radiocarbon composition of $CO_2$ in the soil gas environment must account for diffusion and diffusive mixing**

Jocelyn E. Egan1, David R. Bowling2, David A. Risk3

1Department of Earth Sciences, Dalhousie University, Halifax, Nova Scotia, B3H 4R2, Canada
 2School of Biological Sciences, University of Utah, Salt Lake City, Utah, 84112, USA
 3Department of Earth Sciences, St. Francis Xavier University, Antigonish, Nova Scotia, B2G 2W5, Canada
 *Correspondence to*: Jocelyn E. Egan (jocelyn.egan@dal.ca)

Abstract. Earth system scientists working with radiocarbon in organic samples use a stable carbon isotope ( $\delta^{13}$ C) correction

- to account for mass-dependent fractionation, Although researchers apply this correction routinely, it has not been evaluated for the soil gas environment, where both diffusive gas transport and diffusive mixing are important. Towards this end we applied an analytical soil gas transport model across a range of soil diffusivities and biological CO2 production rates, allowing us to control the radiocarbon ( $\Delta^{14}$ C) and stable isotope ( $\delta^{13}$ C) compositions of modeled soil CO2 production and atmospheric CO2. This approach allowed us to assess the bias that results from using the conventional correction method for
- 15 estimating  $\Delta^{14}$ C of soil production. We found that the conventional correction is inappropriate for interpreting the radioisotopic composition of CO2 from biological production, because it does not account for both diffusive transport and diffusive mixing. The resultant  $\Delta^{14}$ C bias associated with the traditional correction is highest in soils with low biological production and/or high soil diffusion rates. We propose a new correction solution for radiocarbon applications in the soil gas environment that fully accounts for both diffusion and diffusive mixing.

**20 1 Introduction**

Understanding the age of soil-respired carbon is important for process-based carbon cycle studies in the face of climate change. By knowing the age of respired carbon, we can make predictions about the rate at which carbon stocks will be transformed under current and future climates. In the case of Arctic environments underlain by permafrost, this is especially important as "ancient" carbon sources previously stored may become available to decomposers as the permafrost thaws.

25 Radiocarbon allows us to measure the age of *soil-respired CO*2 (CO2 diffusing from the soil surface to the atmosphere, aka soil flux as in Cerling et al., 1991), but the traditional calculation and reporting convention for radiocarbon was not established for soil gas-phase sampling, but rather for solid (organic matter) sample analysis. The validity of this convention has never been explicitly tested for soil-respired CO2.

1

Jocelyn Egan 2019-2-28 3:18 PM Deleted: include

Jocelyn Egan 2019-2-28 3:19 PM Deleted: caused primarily by photosynthesis

Jocelyn Egan 2019-2-20 3:00 PM Deleted: on Jocelyn Egan 2019-4-8 1:25 AM Deleted: (up to 150 ‰) The traditional radiocarbon reporting convention (Stuiver and Polach, 1977), uses a mass-dependent correction based on the isotopic composition of wood. Its purpose is to correct for biochemical fractionation against the radiocarbon isotopologue  $({}^{14}CO_2)$  abundance during photosynthesis, which is assumed to be twice as strong as for  ${}^{13}CO_2$  based on their respective departures in molecular mass from  ${}^{12}CO_2$ . The classical reference describing these conventional calculations is Stuiver and

5 Polach (1977).

In the soil gas environment, researchers have different implicit expectations for fractionation processes. They generally assume that 14C of CO2 is *not* biochemically fractionated in the gas phase, between the points of CO2 *production* (biological production of CO2 by soil organisms and roots) and measurement (subsurface or flux chamber samples). This assumption is reasonable based on the short residence time of CO2 (minutes to days) in the soil profile before emission to the atmosphere.

- However, soil gas isotopic signatures depart in predictable ways from the signature of production because of physical fractionation. It has been recognized for decades that δ13C of CO2 at any point in the soil profile will never equal the isotopic signature of production, because of transport fractionations that alter produced CO2 before it is measured (Cerling et al., 1991). This theory extends readily to 14C. Overall, soil researchers have two implicit expectations, which is that physical fractionations are present, and that biochemical fractionations are absent. These expectations seem clearly out of step with
- 15 the classical reporting conventions for 14C that compensates only for biochemical fractionation. To date, no study has examined the conventional radiocarbon corrections with these implicit expectations in mind. We believe the traditional correction should be re-evaluated for soil gas studies.

Jocelyn Egan 2019-2-28 7:22 PM Formatted: Strikethrough

Jocelyn Egan 2019-2-28 3:22 PM Deleted: reporting conventions Jocelyn Egan 2019-2-28 3:22 PM Deleted: radiocarbon convention

The principal objective of this study is to determine a way to properly interpret measurements of radiocarbon from the soil gas environment. Using theory and physical modelling of soil gas transport we assess traditional techniques, propose and

20 validate an alternative approach for specific use cases, and consider uncertainties that would result from the misapplication of traditional approaches.

**2 Theory**

25

To understand why the mass-dependent correction presented in Stuiver and Polach (1977) may be a poor fit for soil gas studies, we can look at our current understanding of the stable isotopic composition,  $\delta^{13}C_{\bullet}$  of *soil CO*2 (pore space CO2, mole fraction with respect to dry air). We use delta notation to present the stable isotopic composition of CO2:

$$\delta^{13}\mathsf{C} = \left(\frac{\mathsf{R}_{\mathsf{s}}}{\mathsf{R}_{\mathsf{VPDB}}} - 1\right) 1000,$$

where  $\delta^{13}C$  is the isotopic composition in ‰ (see Table 1 for a full list of abbreviations),  $R_s$  is the  ${}^{13}C/{}^{12}C$  ratio of the sample, and  $R_{VPDB}$  is the  ${}^{13}C/{}^{12}C$  ratio of the international standard, Vienna Pee Dee Belemnite.

2

Jocelyn Egan 2019-2-28 10:39 AM **Deleted:** convention

(1)

From foundational work done by Cerling (1991) we know that the isotopic composition of soil CO2 is different from that of soil-respired CO2 Any change in  $\delta^{13}$ C of soil CO2 with depth is influenced by 1) mixing of atmospheric and biological (or biogeochemical) sources of isotopically-distinct CO2, which may occur via diffusion (no bulk gas flow; referred to as diffusive mixing for the remainder of the paper) or advection (bulk gas flow) and 2) kinetic fractionation by diffusion. The

- 5 effect of these is illustrated in Fig. 1. In panel (a) two depth profiles of δ13C of CO2 were modelled in a steady-state environment are shown (the model will be described later). The profiles differ only in soil diffusivity; all other characteristics were held constant, including rates of production, and δ13C of CO2 in the atmosphere (-8 ‰; circle) and biological production (-25 ‰; square with dashed line). In the resultant depth profile with higher soil diffusivity in panel (a), the δ13C of soil CO2 ranges from -8 to -15.1 ‰. In the depth profile representing a soil with lower diffusivity, the δ13C of soil
- 10  $CO_2$  ranges from -8 to -20.6 %. We stress again these two isotopic depth profiles differ only due to differences in transport
- as a result of their varying soil diffusivities. In the depth profile with lower soil diffusivity, atmospheric CO2 does not penetrate downwards as readily, so the profile shape is much steeper near the soil-atmosphere boundary, and is more reflective of the production source composition, -25 ‰, at depth. In the depth profile with higher soil diffusivity, atmospheric air of -8 ‰ more readily mixes from the surface downward by diffusion, so the near surface isotopic 15] composition will be more reflective of the atmosphere due to diffusive mixing of these end-members near the soil surface.

Importantly, the soil CO2 never equals the  $\delta^{13}$ C of production (-25 ‰) at any depth, in either profile in Fig. 1(a). It is not possible to directly measure  $\delta^{13}$ C of production *in situ*, because diffusion and diffusive mixing alter the character of CO2 immediately after its production. From the site of production in the soil, 12CO2 diffuses somewhat faster through the soil than 13CO2, because the former has lower mass. This diffusive difference leads to isotopic fractionation, and results in depth

- 20 profiles of  $\delta^{13}$ C of soil CO2 that are isotopically enriched (less negative) as compared to the source of production. Work by Cerling (1984) and later by Cerling et al. (1991), demonstrated that the mass differences between the two isotopologues led to a difference in diffusion rate of each in air, amounting to a fractionation of 4.4 ‰ (note that this applies only to binary diffusion of CO2 in air and will differ if CO2 diffuses in other gases). As a result, the  $\delta^{13}$ C of soil CO2 measured at any depth will be enriched by a minimum of 4.4 ‰ relative to the biological production CO2 source. However, the  $\delta^{13}$ C of soil-respired
- 25 CO2 can be considerably more enriched than 4.4 ‰ relative to production due to diffusive mixing with the atmosphere as shown in Fig. 1(a).

A convenient theoretical formulation for correcting δ13C for both diffusion fractionation and diffusive mixing was introduced by Davidson (1995), following on the work of Cerling (1984) and Cerling et al. (1991). This approach allows one to combine measurements of CO2 and its isotopic composition within the soil and the air above it, to infer the isotopic
composition of CO2 produced in the soil. This only applies when transport within the soil is purely by diffusion (no bulk air movement). The Davidson (Davidson, 1995) solution uses the difference between the diffusion coefficients for 12C and 13C as follows:

3

 Jocelyn Egan 2019-4-7 9:49 PM

 Jocelyn Egan 2019-4-7 9:50 PM

 $\delta_J^{13} = \frac{C_{\rm s}(\delta_s^{13} - 4.4) - C_{\rm a}(\delta_a^{13} - 4.4)}{1.0044(C_{\rm s} - C_{\rm a})}$

where  $\delta_j^{13}$  is the  $\delta^{13}$ C composition of CO2 from soil production (biological respiration within the soil), Cs and  $\delta_s^{13}$  are the mole fraction and isotopic composition of soil CO2, and Ca and  $\delta_a^{13}$  are the mole fraction and isotopic composition of CO2 in the air just above the soil. In Fig. 2(a) the mole fraction and isotopic composition of soil CO2 at a 40 cm depth and of the air

- 5 just above the soil was "sampled" from model-generated soil depth profiles and the (unrounded) values were used to calculate the isotopic composition of production using Davidson's equation ( $C_s = 14780_e ppm, \delta_s^{13} = -20.3832_e\%, C_a = 380$ ppm and  $\delta_a^{13} = -8$  ‰). The resulting  $\delta_J^{13}$  (e.g. Eq. (2)) at this depth equals the true isotopic composition of production (see inset box, 2(a)). However, because the Davidson approach accounts for diffusion and diffusive mixing, at any given soil depth, not just 40 cm, the modelled values of  $C_s$  and  $\delta_s^{13}$  in Fig. 1(a) and 2(a) will always yield (via Eq. (2)) the true isotopic
- 10 composition of production,  $\delta_I^{13} = -25 \ \%$  (dashed line). If  $\delta^{13}$ C of soil CO2 were (erroneously) interpreted to represent the  $\delta^{13}$ C of soil-respired CO2, the error could be as large as the absolute value of  $(\delta_a \delta_f^{13}) 4.4 \ \%$ . In Fig. 2(a), the error in interpreting  $\delta^{13}$ C of soil CO2 at 40 cm depth ( $\varsigma 20 \ \%$ ) as a value of production would be ~5 \ \%, and the error increases in magnitude at shallower depths. This Davidson (1995)  $\delta_I^{13}$  approach has been shown to be robust when applied to field data from natural soils (Bowling et al., 2015; Breecker et al., 2012b; Liang et al., 2016).
- 15 While 14C is a radioactive isotope and thus decays with time, the half-life is sufficiently long so that 14CO2 behaves similarly
  15 to stable isotopes on the timescales at which diffusion occurs in a soil gas system. The binary diffusion equation (rate of diffusion ~ 1/√mass) provides us with an expected difference in diffusion rate of isotopologues. In this way, δ13C diffusive fractionation theory can be applied to the radiocarbon isotopic composition, δ14C, so long as we account for the mass difference. The larger mass of 14C means that the diffusion fractionation factor is calculated to be 8.8 ‰ based on the atomic
  20 masses of 14CO2, 12CO2 and of bulk air (Southon, 2011).

We can show that  ${}^{14}CO_2$  distribution in soils will be like that of  ${}^{13}CO_2$ , if we model its distribution through depth in the same synthetic soil gas environment. In Fig. 1(b) we present a modelled soil environment with defined atmospheric and production source CO2 isotopic composition boundary conditions for  $\delta^{14}C$ , the  ${}^{14}C$  equivalent to  $\delta^{13}C$ :

$$\delta^{14}C = \left(\frac{A_s}{A_{aba}} - 1\right) 1000$$

25 where δ14C is the isotopic composition in ‰, As is the measured activity of the sample, and Aabs is the activity of the oxalic acid standard (both unitless). As in Fig. 1(a), in panel (b) the profile with lower soil diffusivity, the downward penetration of atmospheric CO2 into the soil profile is reduced, and as a consequence the isotopic depth profile more closely reflects (but does not equal) the composition of production (-50 ‰; dashed line). When the diffusion rate is high and transport is rapid, the atmospheric source is more readily able to penetrate the profile and mix with the production source. In both profiles, the

4

**Jocelyn Egan 2019-4-7 9:53 PM Deleted: 20 Jocelyn Egan 2019-4-7 9:52 PM Deleted: 9020 Jocelyn Egan 2019-4-7 9:53 PM Deleted: 1745 Jocelyn Egan 2019-4-7 9:54 PM Deleted: 20**

Jocelyn Egan 2019-2-20 3:22 PM **Deleted:** depth (~ 2 Jocelyn Egan 2019-2-20 3:22 PM **Deleted:** and

Jocelyn Egan 2019-2-28 4:17 PM Deleted: Graham's law of effusion

Jocelyn Egan 2019-4-7 9:54 PM Deleted: 200

(3)

measured value of soil CO2 at a given depth will not equal the isotopic production value of  $\frac{50}{50}$ , because of diffusion and diffusive mixing. Similar profiles of  $\delta^{14}$ C of soil CO2 with depth, highlighting the diffusive effects, have been presented by Wang et al. (1994).

Since δ14C transport of soil CO2 is like that of δ13C, it follows that we should apply corrections for δ14C like those in Eq. (2)
in order to calculate the isotopic composition of production. The δ14C reformulation of Davidson's δ13j equation is as follows:

$$\delta_J^{14} = \frac{C_s(\delta_s^{14} - 8.8) - C_a(\delta_a^{14} - 8.8)}{1.0088(C_s - C_a)},$$

where  $\delta_J^{14}$  is the  $\delta^{14}$ C composition of soil production, Cs and  $\delta_s^{14}$  are the mole fraction and  $\delta^{14}$ C composition of the soil CO2, and Ca and  $\delta_a^{14}$  are the mole fraction and  $\delta^{14}$ C composition of CO2 in the air just above the soil. This Davidson reformulation for  $\delta^{14}$ C,  $\delta_J^{14}$ , was applied to a model-generated profile of soil  $\delta^{14}$ C at a 40 cm depth in Fig. 2(b), like in panel (a) for  $\delta^{13}$ C

- 10 for  $\delta^{14}C$ ,  $\delta^{14}_J$ ,  $\delta^{14}_J$ , was applied to a model-generated profile of soil  $\delta^{14}C$  at a 40 cm depth in Fig. 2(b), like in panel (a) for  $\delta^{13}C$ ( $C_s = \underline{14780}$  ppm,  $\delta^{14}_s = \underline{-39.3989}$  %,  $C_a = 380$  ppm and  $\delta^{14}_a = \underline{45.5276}$  %; see inset box, Fig. 2(b)). As was the case for  $\delta^{13}C$ in Fig. 2(a), the modelled values of  $C_s$  and  $\delta^{14}_s$  at any depth will yield the true isotopic composition of production,  $\underline{-50}$  % (dashed line), because this approach accounts for diffusion and diffusive mixing.
- 15 The traditional approach for interpreting the 14C composition of soil CO2 and soil-respired CO2 (e.g., Trumbore, 2000) differs from the  $\delta^{14}$ C example above, because a  $\delta^{13}$ C correction is applied to account for mass-dependent isotopic fractionation of biochemical origin (Stuiver and Polach, 1977), ultimately converting  $\delta^{14}$ C to a variant called  $\Delta^{14}$ C. The derivation of the mass-dependent correction is provided in Stuiver and Robinson (1974), where observations are normalized to an arbitrary baseline value of -25 ‰ for  $\delta^{13}$ C (a value for terrestrial wood), and the 13C fractionation factors are squared to account for the 14C/12C fractionation factor as follows:

$$\begin{split} \mathbf{A}_{SN} &= A_{S} \left[ \frac{R_{S}(-25)}{R_{S}} \right]^{2}, \\ &= A_{S} \frac{\left[ \left( 1 - \frac{25}{1000} \right)^{*} R_{VPDB} \right]^{2}}{\left[ \left( 1 + \frac{8^{13}C}{1000} \right)^{*} R_{VPDB} \right]^{2}}, \\ &= A_{S} \frac{\left[ \left( 1 - \frac{25}{1000} \right) \right]^{2}}{\left[ \left( 1 + \frac{4^{33}C}{1000} \right)^{2}}, \end{split}$$

where  $A_{SN}$  is the normalized sample activity,  $A_s$  is the sample activity, and  $\delta^{13}C$  is the isotopic composition of the sample 25 (soil CO2 in our case). As explained in Stuiver and Robinson (1974), the 0.975 term sometimes used in forms of  $A_{SN}$  is equivalent to  $\left(1 - \frac{25}{1000}\right)$ , which we will retain for clarity. The equation for  $\Delta^{14}C$ , the  $\delta^{13}C$  corrected variant of  $\delta^{14}C$ , can then

5

(4)

(5)

| Jocelyn Egan 2019-4-7 9:55 PM |
|-------------------------------|
| Deleted: 20                   |
| Jocelyn Egan 2019-4-7 9:56 PM |
| Deleted: 9020                 |
| Jocelyn Egan 2019-4-7 9:56 PM |
| Deleted: 178.9879             |
| Jocelyn Egan 2019-4-7 9:56 PM |
| Deleted: 138                  |
| Jocelyn Egan 2019-4-7 9:57 PM |
| Deleted: 200                  |

be created from Eq. (5) by substituting in delta notation for  $\Delta^{14}C$  of  $\Delta^{14}C = (A_{SN}/A_{abs} - 1)*1000$  as follows Stuiver and Robinson (1974):

[revised manuscript text omitted]

$$Conc(z) = \frac{P/L}{p} \left( L \times z - \frac{z^2}{z} \right) + Conc_{atm}.$$
(14)

In the model, isotopologues of CO2 are treated as independent gases, with their own specific concentration gradients and diffusion rates (Cerling et al., 1991; Nickerson and Risk, 2009b; Risk and Kellman, 2008). We assume total CO2 to be 12CO2 20 because of its high abundance. The error associated with this assumption is less than 0.01% (Amundson et al., 1998). Eq. (14) is thus applied for 13CO2 and 14CO2. For the full derivation see Nickerson et al. (2014) Section 2.3.

25

The analytical gas transport model was applied across a range of soil diffusivity (1x10-7, 1x10-6 and 1x10-5 m2 s-1)\_soil production rates (0.5, 1, 2, and 4  $\mu$ mol CO2 m-3 s-1),  $\delta^{13}$ C of biological production (- 30 ‰, -20 ‰, -15 ‰), and  $\Delta^{14}$ C of biological production (-500  $\infty_{\circ}$  -200  $\infty_{\circ}$  -1  $\infty_{\circ}$  1  $\infty_{\circ}$  200  $\infty_{\circ}$  500  $\infty_{\circ}$ ). In addition it used  $\Delta^{14}$ C of atmospheric CO2 ( $\Delta_{n2}$  10  $\frac{(6)}{2}$  and  $\delta^{13}C$  of atmospheric CO2 ( $\delta_{a_{3}}$  · 8  $\frac{(6)}{2}$ ), representing realistic conditions found in nature. The other model boundary

conditions were as follows: L = 0.8 m, z = 0.025 m, and Concatm = 15833 µmol m-3 (~380 ppm). The output of the model under these applied conditions were profiles of 12CO2, 13CO2, and 14CO2 for each depth (z) down to the bottom boundary (L).

**3.2 Testing the Traditional and Alternative Corrections**

**The model-generated soil pore space values of ${}^{12}CO_2$ , ${}^{13}CO_2$ , and ${}^{14}CO_2$ at each depth were then used to produce soil depth 5 profiles of $\delta^{13}C$ , $\delta^{14}C$ , and $\Delta^{14}C$ of CO2 as demonstrated earlier in Figs. 1 and 2. To simulate the traditional correction for radiocarbon (referred to as $\Delta^{14}C_{old}$ here), we input the soil pore space values into Eq. (7) to calculate the $\Delta^{14}C$ depth profiles.**

Our proposed new approach is based on Davidson's (1995) theory. Rather than using the  $\delta^{13}C$  soil pore space as a massdependent correction in Fig. 2, we suggest instead using the value  $\delta_J^{13}$  (Eq. (2)), the biological production of  $\delta^{13}C$ , in its place in the denominator of Eq. (7) as follows:

$$\Delta^{14}C_{new} = \left[ \left(\frac{A_S}{A_{abs}}\right) \frac{\left(1 - \frac{25}{1000}\right)^2}{\left(1 + \frac{\delta_J^3}{1000}\right)^2} - 1 \right] 1000.$$
(15)

The model-generated soil pore space isotope values were input into Eq. (15) to produce depth profiles of  $\Delta^{14}C_{new}$ . Values of  $\Delta^{14}C_{new}$  through depth represent transport-fractionation-corrected soil CO2 values of radiocarbon. For our simulated data, we assume the measurement year is present day, as we do not use the oxalic acid (Aabs) in Eq. 15. Then to calculate the

15 radiocarbon composition of production,  $\Delta_{j}^{14}$ , the  $\Delta^{14}$ C composition of the soil CO2,  $\Delta^{14}$ Cnew, can be placed into our adaption of Davidson (1995) for 14C (Eq. (4)) as follows:

$$\Delta_J^{14} = \frac{c_{\rm s}(\Delta^{14}c_{new} - 8.8) - c_{\rm a}(\Delta_a^{14} - 8.8)}{1.0088(c_{\rm s} - c_{\rm a})},\tag{16}$$

where  $\Delta_j^{14}$  is the  $\Delta^{14}$ C composition of soil production,  $C_s$  and  $\Delta^{14}C_{new}$  are the mole fraction and  $\Delta^{14}C$  composition of the soil CO2, and  $C_a$  and  $\Delta_a^{14}$  are the mole fraction and  $\Delta^{14}C$  composition of CO2 in the air just above the soil.

**20 3.3 Field Soil Profiles**

10

To demonstrate our new approach with real data, we used  $\Delta^{14}$ C values collected in the field. We collected samples of soil CO2 at 3 depths from the soil profile and from the air just above the soil, from 3 sites in Weyburn, Saskatchewan, for both stable and radio-isotope analysis. These sites were located in agricultural fields, with soils characterized as a combination of Estevan and Roughbark souls. There soils have sandy loam surfaces, with thin organic layers and soil layers that are poorly.

25 drained and have high clay content (SCSR, 1997),

Jocelyn Egan 2019-2-28 10:47 AM Deleted: Conventions

Jocelyn Egan 2019-2-28 10:07 PM Deleted: convention

Jocelyn Egan 2019-2-20 3:36 PM **Deleted:** composed of mostly clay, with shallow organic layers.

To collect samples, we installed horizontal polyvinyl chloride (PVC) soil gas wells, 60 cm long, with 3-ply Gore-tex™ membranes covering 2 cm holes drilled every 10 cm along the length of PVC, at soil depths of 5 cm, 10 cm, 20 cm, 50 cm and 75 cm. Details on the soil gas well design and sampling disturbance prevention are described in detail in Risk et al. (2013). We withdrew a minimum of two samples (one shallow and one deep) and one air sample at each of the three sites 6

- times (approximately every 2 months) between August 2011 and May 2012 as described in Risk et al. (2013), by connecting a N2-purged and evacuated 1 L stainless steel canister (Lab Commerce) to the well and allowing it to equilibrate for 15 mins. The samples were sent to the University of Florida for cryogenic purification and then to the University of California Irvine Keck Accelerator Mass Spectrometer (AMS) facility to be graphitized and analyzed for Δ14C.
- As described in Risk et al. (2013), on each visit triplicate samples were also collected in 10 ml N2-purged and evacuated Exetainers vials (Labco, UK) at each depth as well as from the air above the soil. The triplicate samples were analyzed for bulk CO2 and δ13C within 2 weeks of sampling using a GV Isoprime CF-IRMS and Multiflow gas bench (Isoprime, UK).

We used the  $\Delta^{14}$ C values reported to us from the AMS laboratory along with  $\delta^{13}$ C (presented in Risk. et al. 2013) to backcorrect the values so that we could present the results using  $\Delta^{14}C_{new}$  (Eq. 15) and then also calculate the radiocarbon composition of biological production using  $\Delta_1^{14}$  (Eq.16).

**3.4 Interpreting Soil Profiles**

Our primary goal was to correctly extract the isotopic composition of  $CO_2$ , produced by biological production. We applied two methods to extract this signal from the model-generated soil  $CO_2$  profiles as well as soil  $CO_2$  profiles from the field:

Method 1 follows the traditional correction, where Eq. (7) was used to calculate Δ14Cold, and\_we interpret this value as the radiocarbon composition of CO2 from biological production. Although this interpretation may not seem commonly used, as most people now understand the soil CO2 and soil-respired CO2 differ, we want to use it here as an example. If a researcher were to interpret a soil CO2 radiocarbon measurement as old and representative of the end-member source it came from, as demonstrated in Section 2, this soil CO2 sample will not necessarily be representative of the end-member production source because of gas transport mechanisms.

Method 2, our proposed solution, instead determines the radiocarbon composition of CO2 from biological production in two

- steps, as described in Section 3.2. First, the transport corrected radiocarbon composition of soil CO2, is calculated using Eq.
- 30 (15) ( $\Delta^{14}C_{new}$ ). Then this properly interpreted soil CO2 value can be used in Eq. (16) to calculate the radiocarbon composition of production,  $\Delta_j^{14}$ .

Jocelyn Egan 2019-2-28 10:20 PM Deleted: fractionation

**4 Results**

5

**4.1 Testing the Traditional Approach**

In Fig. 3 we present the results from model scenarios that simulated natural soil diffusive\_mixing profiles between sources that are intended to represent a realistic range of values as a function of depth and soil properties. All the scenarios presented had the same prescribed  $\Delta^{14}$ C source values (CO2 in the air just above the soil and soil biological production), and only soil diffusivities and production rates differed. Like the  $\delta^{13}$ C and  $\delta^{14}$ C soil CO2 depth profiles in Fig. 1, the  $\Delta^{14}$ C soil CO2 profiles varied as a function of soil diffusivities and production rates. In the model scenarios in Fig. 3(a), the depth profiles with higher soil diffusivities had more enriched  $\Delta^{14}$ C soil CO2 (20-60 ‰ more enriched, depending on depth) than the scenarios

with lower soil diffusion rates. In Fig. 3(b), the depth profiles with lower production rates had more enriched  $\Delta^{14}$ C of soil 10 CO2 (5-10 ‰ more enriched) than scenarios with higher production rates. In the profiles with steeper diffusive gradients near the surface, soil CO2 through depth was more representative of the production source, as we would expect from theory.

In Fig. 3 two depth profiles are plotted for each model iteration, one where the model output was used to calculate the conventional representation of radiocarbon,  $\Delta^{14}C_{old}$  (Eq. (7); solid line) and the second where the output was used to 15 calculate  $\Delta^{14}C_{new}$ , our proposed convention for presenting\_radiocarbon soil CO2 (dashed line). All of the  $\Delta^{14}C_{old}$  depth

- profiles in Fig. 3 had slightly (~ 10 ‰) more depleted  $\Delta^{14}$ C soil CO2 values than the  $\Delta^{14}$ Cnew depth profiles. When the radiocarbon composition of CO2 from biological production,  $\Delta_J^{14}$ , was calculated using  $\Delta^{14}$ Cnew (Eq. (16); Method 2), the result was equal to the radiocarbon value of production input into the model (-50 ‰) through the entire soil CO2 depth profile under all soil scenarios. In contrast, when Method 1 was applied, we would (incorrectly) interpret  $\Delta^{14}$ Cold soil CO2 to
- 20 be the radiocarbon composition of  $CO_2$  from biological production, the  $\Delta^{14}C_{old}$  values through depth never equalled the  $\Delta^{14}C$ value of soil production input into the model ( $\frac{50}{30}$ %).

**4.2 Field Experiment**

25

The radiocarbon values for two of the sampling dates where soil CO2 samples were collected from a field site in Weyburn, Saskatchewan, Canada, are presented in Fig. 4. The traditional  $\Delta^{14}$ C convention,  $\Delta^{14}$ Cold, reported to us by the laboratory

- where the samples were analysed are plotted with solid lines. If Method 1 was used, we would (incorrectly) interpret these values as the values of biological production, but instead they represent soil CO2 values. The dotted lines are the radiocarbon compositions of production ( $\Delta_{j}^{14}$ ), where the reported radiocarbon values were back-corrected using our new approach,  $\Delta^{14}C_{new}$ , and then input into Eq. (16) (Method 2). When considering age, the radiocarbon compositions of production ( $\Delta_{j}^{14}$ ),
- 30 calculated using Method 2, were older in the springtime than those calculated using Method 1, despite the fact that Method 2 values lie on either sides of Method 1 values (more deplete at the surface and more enriched at depth). Although the values are more enriched at depth, in terms of age, they are still older C. This is because both Method 1 and 2 values fall within the post 1950s 14C bomb spike period (Trumbore, 2000), but the more enriched values are still slightly older in terms of age.

10

| Jocelyn Egan 2019-4-7 10:06 PM |
|--------------------------------|
| Deleted: 50                    |
| Jocelyn Egan 2019-4-7 10:06 PM |
| Deleted: 100                   |
| Jocelyn Egan 2019-4-7 10:07 PM |
| Deleted: 10                    |
| Jocelyn Egan 2019-4-7 10:07 PM |
| Deleted: 2                     |

Jocelyn Egan 2019-4-7 10:08 PM Deleted: 200

Jocelyn Egan 2019-4-7 10:08 PM Deleted: 200

Jocelyn Egan 2019-2-28 10:22 PM Deleted: Using

**5** Discussion**

**5.1 Correction Bias and Limitations**

Diffusion- and production-controlled soil gas profiles, similar to those presented in Fig. 3, have been shown in other papers for  $\delta^{13}$ C (e.g., Bowling et al., 2015; Breecker et al., 2012a; Cerling, 1984; Cerling et al., 1991; Davidson, 1995; Nickerson

- 5 and Risk, 2009b). Given that we based our gas transport corrected  $\Delta^{14}$ C calculation on the same equations, this is what we expected, where values of  $\Delta^{14}$ C of soil CO2 differed in the soil profile (shown in Fig. 3) were intermediate between the two mixing sources (CO2 in the air just above the soil and soil biological production). These effects of diffusion and diffusive mixing demonstrate that Method 1 (which assumes  $\Delta^{14}$ Cold of soil CO2 =  $\Delta^{14}$ C of soil production) is inappropriate for the soil gas application because it does not describe the radiocarbon composition of CO2 as produced, and before alteration by gas
- 10 transport processes. In contrast, Method 2 (where the  $\Delta^{14}$ C isotopic signature of production,  $\Delta_J^{14}$ , is calculated from  $\Delta^{14}$ Cnew) does not have any error because this method is able to calculate the true isotopic composition of soil production along the entire soil profile diffusive mixing gradient, no matter how steep or shallow the gradient.
- The degree of error for Method 1 will depend on a given soil environment, where soil diffusivity and production rates will
  either amplify or decrease the error. When we "sampled" model-produced depth profiles using Method 1, we were unable to correctly extract the specified isotopic compositions of biological production that were input in the model, under any scenario. Method 1 error can be quantified as the absolute difference between the model's prescribed isotopic value of production, and Δ14Cold. In the specific scenarios shown in Fig. 3, the smallest Method 1 error (0.3 ‰) was in model scenarios with high production rates or low diffusivities, and the largest error (18 ‰) was in scenarios with higher soil diffusivities. Based on the rate of decline of atmospheric bomb 14C of 4 to 5.5 ‰ yr-1 (Graven et al., 2012), a 18 ‰ error would equate to a 3.3 to 5 year age error.

The fraction of atmosphere-sourced  $CO_2$  ( $f_a$ ) present in the soil profile caused by diffusive mixing scenarios typically associated with different types of soils can be a predictor for Method 1 error, as illustrated conceptually in Fig. 5. Soil

- environments that typically have higher soil diffusivities and lower production rates will have smaller soil-diffusive gradients, and there is a larger amount of atmospheric CO2 in the soil compared to the total CO2. These scenarios have a much higher fa value. In these scenarios, diffusive mixing dominates, which amplifies Method 1 error. In contrast, in soils with larger soil-diffusive gradients, there is a much smaller amount of atmospheric CO2 present compared to the total amount of CO2, and therefore the isotopic depth profile more closely resembles a mass-mixing profile. These soil CO2 depth profiles
- 30 have lower fa values and thus Method 1 will be less erroneous in these scenarios, because diffusive mixing is not as prominent.

Using our field study as an example (profiles shown in Fig. 4), the high clay content of its soil (SCSR, 1997) likely has relatively low soil diffusivity, and is most like the lower soil diffusivity scenarios in Fig. 3. In Fig. 5, this soil is like the intermediate scenario, with an intermediate  $f_a$  value, and diffusive mixing gradient. The use of Method 1 in this type of soil environment would be less erroneous than, for example, a dry desert soil environment with a high soil diffusion rate. Based

- 5 on our model simulations, we can assume that Method 2 is producing the true Δ14C value of production, so Method 1 error can be calculated as the difference between Method 1 and 2 for this real data. In our case here, we observed Method 1 error to be as low as 10 ‰ (2 to 2.5 year age error) in February and up to 100 ‰ (18-25 year age error) at depth in May, although the magnitude of expected error is variable and is dependent on diffusivity and production rates as shown in Fig. 3 and 5.
- 10 This analysis does, however, have limitations. The synthetic soil environment is simplified in our approach, and in reality soil diffusion and production rates are not constant through depth. The model is necessarily simplified so that it can be easily solved analytically, but in reality soils are typically not in steady state (van Asperen et al., 2017; Bowling et al., 2009; Bowling and Massman, 2011; Goffin et al., 2014; Maier et al., 2010; Moyes et al., 2010; Nickerson and Risk, 2009b; Risk and Kellman, 2008). For example, gravitational settling and thermal diffusion can impact the diffusive non-steady state in
- 15 deep soils (Severinghause et al., 1996) and diurnal temperature cycles (Nickerson and Risk, 2009b; Phillips et al., 2010; Risk and Kellman, 2008) and wind advection (Bowling and Massman, 2011) can be of impact non-steady state in near surface soils. The error values presented here are therefore likely conservative, since previous  $\delta^{13}$ C studies have shown that these non-equilibrium processes generally add fractionation uncertainty (Nickerson and Risk, 2009b; Phillips et al., 2010; Risk and Kellman, 2008). Additionally, sampling methods may impose an additional layer of non-equilibrium fractionation
- 20 uncertainty (Egan et al., 2014) that may need to be evaluated. The real magnitude of error will depend on the given soil environment and will be sensitive to the soil conditions and sampling methodology, Additionally, carbonate soils could introduce more error, as the isotopic exchange between soil gas and carbonates is not mass-dependent (Breecker et al., 2009).
- 25 Surface flux chambers are commonly used sampling methodology used for measuring the radiocarbon composition of production, and were not addressed in our analysis. However, Method 1 is actually acceptable for use in the case of surface flux chambers, because unlike soil CO2 which will always differ from soil production soil-respired CO2, conservation of mass dictates that isotopic values of flux must represent soil production so long as the soil is in steady-state (Cerling et al., 1991). While radiocarbon surface flux data need no correction for transport fractionation, researchers should be cautious
- 30 when using surface flux chambers because they can cause isotopic dis-equilibrium (Albanito et al., 2012; Egan et al., 2014; Midwood and Millard, 2011; Nickerson and Risk, 2009a). As shown in the Egan et al. (2014) study, static chamber methods (i.e. Hahn et al., 2006) and the proposed forced-diffusion chamber technique were the least erroneous for radiocarbon measurements, whereas dynamic chamber sampling techniques (i.e. Gaudinski et al., 2000; Schuur and Trumbore, 2006) could cause up to 200 ‰ bias under certain soil diffusion and production scenarios.

**12**

**Jocelyn Egan 2019-2-28 10:33 PM**

**Deleted:** Based on our model simulations, we can assume that Method 2 is producing the true  $\Delta^{14}$ C value of production, so Method 1 error can be calculated as the difference between Method 1 and 2 for this real data. Method 1 error was minor in February (10 %; 2 to 2.5 year age error), but it was very large at depth in May (up to 100 %; 18-25 year age error).

Jocelyn Egan 2019-2-28 10:57 AM Deleted: has a few

**Jocelyn Egan 2019-2-28 10:35 PM**

**Jocelyn Egan 2019-2-28 10:30 PM**

**Deleted:** The Garnet et al. (2009) study used a passive sampling variation to the dynamic surface flux chambers technique, where a pump pulled a sample from the chamber and pushed it through a molecular sieve trap over a few days. This study suggested that the conventional Stuiver and Polach (1977) radiocarbon correction accommodated isotopic fractionation by the sieve. However, under non-steady state conditions, the conventional correction amy not actually apply because the  ${}^{14}\text{CO}_{2}/{}^{12}\text{CO}_{2}$  fractionation factor will not always be a constant multiple of the  ${}^{13}\text{CO}_{2}/{}^{12}\text{CO}_{2}$  fractionation factor as the system moves from one state to another (Egan et al., 2014).

Aside from our proposed correction, a few other analytical alternatives might be appropriate for correcting 14C in soil CO2. The Davidson (1995)  $\delta_1$  method was the only gradient approach we tested for Method 2 in our study, but alternative approaches such as those presented for  $\delta^{13}$ C by Goffin et al. (2014) and Nickerson et al. (2014), and for  $\Delta^{14}$ C by Phillips et

- 5 al. (2013) would likely be similarly successful in producing depth-dependent compositions of production. They are, however, not quite as straightforward as the  $\delta_{J}$  method. However, if a researcher chooses to use an alternate gradient approach, they should still use  $\Delta^{14}C_{new}$  rather than  $\Delta^{14}C_{old}$  to calculate soil CO2 first, as  $\Delta^{14}C_{old}$  does not account for transport fractionations. To demonstrate the bias associated with using  $\Delta^{14}C_{old}$  with another gradient approach, we calculated error values for three soil environments with given transport parameters in Table 2. As expected, across all three soil environments
- and depths, Method 2 had no error. If a researcher were to use Δ14Cold with another gradient approach, the bias would be between 7 and 8 ‰ depending on the soil type and depth. This isotopic difference is not large, but it still does not follow theory, so we also compared the difference between using Δ14Cnew and Δ14Cold to interpret the isotopic composition of soil CO2 in these same soil environments in Table 2. In the three soil types modelling, the bias in using Δ14Cold to interpret soil CO2 was always larger near the surface, and was largest in sandy type soils with mid-range production rates and high soil
   diffusion rates

- 15 diffusion rates.

In at least one other specialized instance, researchers have recognized that the normal Stuiver and Polach (1977) reporting convention was not applicable under the circumstances of an experiment and chose to reformulate it for their application. The Torn and Southon (2001) study evaluated the use of  $\Delta^{14}C_{old}$  when radiocarbon is used as a tracer in C cycling field

20 experiments with elevated atmospheric CO2 concentrations. Torn and Southon (2001) proposed that in these experiments, the  $\delta^{13}$ C correction used in  $\Delta^{14}$ Cnld was invalid because differences in  $^{13}$ C abundance associated with elevated atmospheric CO2 was associated with diffusive mixing of different atmospheric masses, and not isotopic fractionation. They instead used a  $\delta^{13}$ C value from an adjacent control plot (non-elevated CO2 concentrations) to accurately estimate  $\Delta^{14}$ C, because the control plot followed the same fractionation pathways, but without the elevated atmosphere.

25

Both the Torn and Southon (2001) study and ours highlight the importance of reassessing old isotopic approaches for new application environments. To date, only three known studies (Egan et al., 2014; Phillips et al., 2013; Wang et al., 1994) have accounted for 14C diffusion-transport, though ours is the first to propose a straightforward and theoretically-robust correction that replaces the Stuiver and Polach (1977) solution for the soil gas environment (Method 2).

**30**

**5.4 Workarounds and Establishing New Best Practice**

More research groups are starting to use soil gas wells/soil  $CO_2$  in conjunction with gradient techniques because of the known isotopic effects caused by many chamber techniques (Albanito et al., 2012; Egan et al., 2014; Midwood and Millard, 2011; Nickerson and Risk, 2009a). Gradient approaches also allow researchers to determine depth-dependent values of

**13**

**Jocelyn Egan 2019-2-28 10:28 PM Moved (insertion) [1]**

Jocelyn Egan 2019-2-28 10:36 PM Deleted: Surface flux chambers are commonly used for measuring the radiocarbon composition of production Method 1 is actually acceptable for use in the case of surface flux chambers, because unlike soil CO2 which will always differ from soil production soil-respired CO2, conservation of mass dictates that isotopic values of flux must represent soil production so long as the soil is in steady-state (Cerling et al., 1991). While radiocarbon surface flux data need no correction for transport fractionation, researchers should be cautious when using surface flux chambers because they can cause isotopic dis-equilibrium (Albanito et al., 2012; Egan et al., 2014; Midwood and Millard, 2011; Nickerson and Risk, 2009a). As shown in the Egan et al. (2014) study, static chamber methods (i.e. Hahn et al., 2006) and the proposed forced-diffusion chamber technique were the least erroneous for radiocarbon measurements, whereas dynamic chamber sampling techniques (i.e. Gaudinski et al., 2000; Schuur and Trumbore, 2006) could cause up to 200 ‰ bias under certain soil diffusion and production scenarios. The Garnet et al. (2009) study used a passive sampling variation to the dynamic surface flux chambers technique, where a pump pulled a sample from the chamber and pushed it through a molecular sieve trap over a few days. This study suggested that the conventional Stuiver and Polach (1977) radiocarbon correction accommodated isotopic fractionation by the sieve. However, under non-steady state conditions the conventional correction may not actually apply because the 14CO2/12CO2 fractionation factor will not always be a constant multiple of the 13CO2/12CO2 fractionation factor as the system moves from one state to another (Egan et al., 2014).

**Jocelyn Egan 2019-2-28 10:28 PM**

Moved up [1]: Surface flux chambers are commonly used for measuring the radiocarbon composition of production. Method 1 is actually acceptable for use in the case of surface flux chambers, because unlike soil CO2 which will always differ from soil production soil-respired CO2, conservation of mass dictates that isotopic values of flux must represent soil production so long as the soil is in steady-state (Cerling et al., 1991). While radiocarbon surface flux data need no correction for transport fractionation, researchers should be cautious when using surface flux chambers because they can cause isotopic dis-equilibrium (Albanito et al., 2012; Egan et al., 2014; Midwood and Millard, 2011; Nickerson and Risk, 2009a). As shown in the Egan et al. (2014) study, static chamber methods (i.e. Hahn et al., 2006) and the proposed forceddiffusion chamber technique were the least erroneous for radiocarbon measurements, wh ... [2]

production. Previously reported data using the traditional Stuiver and Polach (1977) reporting convention, can be backcorrected using our solution. The back-correcting solution does, however, assume that the researcher has measured a value of  $\delta_s^{13}$  independent from  $\Delta^{14}C_{old}$ , as the AMS measured  $\delta^{13}$  is not comparable to Isotope-Ratio Mass Spectrometry (IRMS) measured  $\delta^{13}$ . To assess the sensitivity of the  $\delta_j^{13}$  value in calculating  $\Delta^{14}C_{mew}$  and  $\Delta_j^{14}$  for back corrections, in Table 3 we present the error in using a  $\delta_j^{13}$  value that is wrong by  $\pm$  0.5, 1, and 5 ‰. If rounded to the nearest ‰ value, the bias associated with using a  $\delta_j^{13} \pm 0.5$ , 1 and 5 ‰ across the three soil types and depths modelled is always 1, 2, and 10 ‰, respectively. The back correction solution can therefore work if a researcher collected  $\delta^{13}C$  measurements independently, although we recommend against using this back correction method for new data.

- 10 For researchers who have soil CO2 data previously interpreted using the  $\Delta^{14}C_{old}$  calculation, the following steps will help correct for transport fractionations: 1) use  $\delta_s^{13}$  and  $\Delta^{14}C_{old}$  to back out the activity of the sample (As); 2) calculate the isotopic composition of production for  $\delta^{13}C$  using Eq. (2),  $\delta_j^{13}$ ; 3) use  $\delta_j^{13}$  and As in Eq. (7) to calculate  $\Delta^{14}C_{new}$ , and finally 4) determine the radiocarbon isotopic composition of production, using Eq. (16),  $\Delta_j^{14}$ .
- 15 Going forward, several changes to best practice are recommended. On a lab level, for new soil CO2 data, we propose that AMS laboratories report radiocarbon using Eq. (3), δ14C, the uncorrected radiocarbon variant, so that the first step above, i.e. use δs13 and Δ14Cold to back out the activity of the sample (As), can be avoided, and researchers can proceed with steps 2-4. We also suggest that researchers measure δ13 alongside Δ14C, so that they are not dependent on the AMS measured δ13 for potential back-corrections, to prevent potential error ranging from 1-10 ‰ (Table 3).

[revised manuscript text omitted]

Saskatchewan Centre for Soil Research (SCSR)... The Soils of the Weyburn-Virden Map Areas 62E and 62F Saskatchewan. University of Saskatchewan No. S7, Saskatcon, Saskatchewan 128 pp, 1997.

15 Schuur, E. A. G. and Trumbore, S. E.: Partitioning sources of soil respiration in boreal black spruce forest using radiocarbon, Glob. Change Biol., 12(2), 165–176, doi:10.1111/j.1365-2486.2005.01066.x, 2006.

Severinghause, J. P., Bender, M. L., Keeling, R. F. and Broecker, W. S.: Fractionation of soil gases by diffusion of water vapor, gravitational settling, and thermal diffusion, Geochim. Cosmochim. Acta, 60(6), 1005–1018, 1996.

 $\frac{\text{Southon, J. R.: Are the fractionation corrections correct: Are the isotopic shifts for {}^{14}\text{C}/{}^{12}\text{C} ratios in physical processes and}{\frac{\text{chemical reactions really twice those for }{}^{13}\text{C}/{}^{12}\text{C}?, \text{Radiocarbon, 53(4), 691-704, 2011.}}$

Stuiver, M. and Polach, H. A.: Discussion: Reporting of 14C Data, Radiocarbon, 19(3), 355–363, 1977.

[revised manuscript text omitted]

Table 2. Bias from interpreting the radiocarbon isotopic composition of soil CO2 and biological production using  $A^{14}C_{add}$ . The depth profiles used to calculate the bias was generated from a model with a  $A^{14}C$  of production of -50 & and combinations of diffusion and production rates to represent three different general soil types: clay (D = 1e-7 m2s-1: P = 0.5 µmol m-3s-1), loam (D = 1e-6 m2s-1: P = 4 µmol m-3s-1), and sand (D = 1e-5 m2s-1: P = 2 µmol m-3s-1). All bias values are absolute.

|   | Soil Type                                                                                           | Depth
(cm) | $\frac{\Delta^{14}C_{old}}{(\%)}$ | $\frac{\Delta_J^{14} \text{ from}}{\Delta^{14} C_{\text{old}} (\%)}$ | Δ14Cnew (‰) | $\frac{\Delta_J^{14} \underline{\text{from}}}{\Delta^{14} \underline{C}_{\text{new}} (\%)}$ | $\frac{\Delta^{14}C_{old} - \Delta^{14}C_{new}}{(\%)}$ | $\frac{-50}{(\%)} \Delta_J^{14} \frac{\text{from } \Delta^{14} C_{\text{old}}}{(\%)}$ |
|---|-----------------------------------------------------------------------------------------------------|---------------|-----------------------------------|----------------------------------------------------------------------|------------------------------------------|---------------------------------------------------------------------------------------------|--------------------------------------------------------|---------------------------------------------------------------------------------------|
| 1 | Clay                                                                                                | 2.5           | -43.0                      | -58.1                                                         | -31.7                                    | -50.0                                                                                       | 11.3                                            | 8.1                                                                            |
|   | $\frac{D = 1e-7 \text{ m}^2 \text{ s}^{-1}}{P = 0.5  \mu \text{mol } \text{m}^{-3} \text{ s}^{-1}}$ | 50     | -49.5                      | -58.3                                                         | -40.8                             | -50.0                                                                                | 8.6                                             | 8.3                                                                            |
|   | Loam                                                                                                | 2.5           | -41.5                      | -58.1                                                         | -29.6                                    | -50.0                                                                                       | 11.9                                            | 8.1                                                                                   |
|   | $\frac{D = 1e-6 \text{ m}^2 \text{ s}^{-1}}{P = 4 \mu\text{mol }\text{m}^{-3} \text{ s}^{-1}}$      | 50     | -49.3                      | -58.3                                                         | -40.6                             | -50.0                                                                                | 8.7                                             | 8.3                                                                            |
|   | Sand                                                                                                | 2.5           | -3.9                       | -57.0                                                                | 24.9                                     | -50.0                                                                                       | 28.8                                                   | 7.0                                                                            |
|   | $\frac{D = 1e-5 m^2 s^{-1}}{P = 2 \mu mol m^{-3} s^{-1}}$                                           | 50     | -38.5                      | -58.0                                                         | -25.3                             | -50.0                                                                                | 13.2                                            | 8.0                                                                            |

Table 3. Sensitivity of the  $\delta_J^{13}$  value in  $\Delta^{44}C_{arev}$  and  $\Delta_J^{14}$  calculations. The depth profiles used to calculate the bias from using the wrong  $\delta_J^{13}$  value was generated from a model with a  $\Delta^{14}C$  of production of -50 ‰ and combinations of diffusion and production rates to represent three different general soil types: clay (D = 1e-7 m2 s-1: P = 0.5 µmol m-3 s-1), loam (D = 1e-6 m2 s-1: P = 4 µmol m-3 s-1), and sand (D = 1e-5 m2 s-1: P = 2 µmol m-3 s-1). All bias values are absolute.

| Soil Type                              | Depth (cm) | $rac{	ext{Deviation in}}{\delta_J^{13}(\underline{\%})}$ | $\frac{\text{Bias using wrong}}{\delta_J^{13} \underline{\text{ in } \Delta^{14} C_{\text{new}}}} (\%)$ | $\frac{\text{Bias using } \Delta^{14}\text{C}_{\text{new}} \text{ with wrong}}{\delta_J^{13} \underline{\text{to calculate } } \Delta_J^{14} \underline{(\%)}}$ |
|-----------------------------------------------|-------------------|-----------------------------------------------------------|---------------------------------------------------------------------------------------------------------|-----------------------------------------------------------------------------------------------------------------------------------------------------------------|
| Clay                                          | 2.5 cm     | -0.5                                               | 1.0                                                                                              | 1.0                                                                                                                                                      |
| $D = 1e-7 m^2 s^{-1}$                         |                   | +0.5                                                      | 1.0                                                                                                     | 1.0                                                                                                                                                      |
| $P = 0.5 \ \mu mol \ m^{-3} \ s^{-1}$         |                   | -1                                                 | 2.0                                                                                                     | 2.0                                                                                                                                                             |
| _                                             |                   | +1                                                        | 1.9                                                                                              | 1.9                                                                                                                                                      |
| -                                             |                   | -5                                                        | 10.0                                                                                                    | 9.8                                                                                                                                                      |
| -                                             |                   | +5                                                        | 9.9                                                                                              | 9.7                                                                                                                                                      |
| -                                             | 50 cm      | -0.5                                                      | 1.0                                                                                              | 1.0                                                                                                                                                      |
| -                                             |                   | +0.5                                                      | 1.0                                                                                              | 1.0                                                                                                                                                      |
| _                                             |                   | -1                                                 | 2.0                                                                                              | 2.0                                                                                                                                                      |
| -                                             |                   | +1                                                 | 2.0                                                                                                     | 1.9                                                                                                                                                      |
| -                                             |                   | -5                                                 | 9.9                                                                                              | 9.8                                                                                                                                                      |
|                                               |                   | +5                                                        | 9.8                                                                                              | 9.7                                                                                                                                                      |
| Loam                                          | 2.5 cm     | -0.5                                               | 1.0                                                                                              | 1.0                                                                                                                                                      |
| $D = 1e-6 m^2 s^{-1}$                         |                   | +0.5                                                      | 1.0                                                                                              | 1.0                                                                                                                                                      |
| $P = 4 \ \mu mol \ m^{-3} \ s^{-1}$           |                   | -1                                                 | 2.0                                                                                              | 2.0                                                                                                                                                             |
| -                                             |                   | +1                                                 | 2.0                                                                                                     | 1.9                                                                                                                                                      |
| -                                             |                   | -5                                                 | 10.0                                                                                             | 9.8                                                                                                                                                      |
| -                                             |                   | +5                                                        | 9.9                                                                                              | 9.7                                                                                                                                                      |
| -                                             | 50 cm      | -0.5                                               | 1.0                                                                                              | 1.0                                                                                                                                                      |
| -                                             |                   | +0.5                                                      | 1.0                                                                                              | 1.0                                                                                                                                                             |
| -                                             |                   | -1                                                 | 2.0                                                                                              | 2.0                                                                                                                                                             |
| -                                             |                   | +1                                                        | 2.0                                                                                                     | 1.9                                                                                                                                                      |
| -                                             |                   | -5                                                 | 9.9                                                                                              | 9.8                                                                                                                                                      |
| _                                             |                   | +5                                                        | 9.8                                                                                              | 9.7                                                                                                                                                             |
| Sand                                          | 2.5 cm     | -0.5                                               | 1.1                                                                                              | 1.0                                                                                                                                                      |
| $D = 1e-5 m^2 s^{-1}$                         |                   | +0.5                                                      | 1.1                                                                                              | 1.0                                                                                                                                                      |
| $\underline{P=2 \ \mu mol \ m^{-3} \ s^{-1}}$ |                   | -1                                                 | 2.1                                                                                                     | 2.0                                                                                                                                                             |
| -                                             |                   | +1                                                        | 2.1                                                                                                     | 1.9                                                                                                                                                      |
| -                                             |                   | -5                                                 | 10.6                                                                                             | 9.8                                                                                                                                                      |
| -                                             |                   | +5                                                        | 10.4                                                                                             | 9.7                                                                                                                                                      |
| -                                             | 50 cm      | -0.5                                               | 1.0                                                                                                     | $\frac{1.0}{1.0}$                                                                                                                                               |
| -                                             |                   | +0.5                                                      | 1.0                                                                                                     | 1.0                                                                                                                                                      |
| -                                             |                   | -1                                                 | 2.0                                                                                                     | 2.0                                                                                                                                                             |
| -                                             |                   | +1                                                        | 2.0                                                                                                     | 1.9                                                                                                                                                      |
| -                                             |                   | -5                                                 | 10.1                                                                                             | 9.8                                                                                                                                                      |
|                                               |                   | +5                                                 | 9.9                                                                                              | 9.7                                                                                                                                                      |

---

## Author Response (AR2)

Manuscript bg-2018-451

Title: "Technical Note: Isotopic fractionation corrections for the radiocarbon composition of  $CO_2$  in the soil gas environment must include diffusion and diffusive mixing" Authors: Jocelyn Egan et al.

Thanks to the editor and to the referees for their constructive comments. Based on the comments, we have reformatted the manuscript to be a Technical Note and believe that it has improved both its clarity and application. Referee and editor comments are listed below in bold, and our responses follow each. Our edited manuscript with tracked changes can be found after our responses.

Sincerely, Jocelyn Egan (for all authors)

**Associate Editor Comments to the Author:**

This is a revised version with major revisions. I recommend another cycle of major revisions because the core of the paper, introducing a valid correction for interpretations of 14C in soil CO2 justifies publication. This is in spite of one reviewer being sufficiently frustrated with the revisions to indicate that another cycle is not justified (recommending rejection). And it seems the authors run the risk that our 2nd expert reviewer will respond the same way. I do concur with the main criticisms and the paper must sufficiently address it.

Generally, it seems everyone involved agree the paper raises a valid point regarding the need to introduce additional corrections when dealing with 14C in soil CO2. And as I think I pointed out upon submission, it would probably be most effective as an important short Method note (after considering the confusing parts such as noted by Ref 3). But it seems the author opted for more complicated options. This is apparent already in the Abstract, most of which focuses on "...assess the bias from using the conventional method... etc". And reach what is the real message of the work only at the last sentence: "...we propose a new correction for radiocarbon applications in soil gas that accounts for diffusion and mixing..." As a result, the paper is loaded with data while insisting it is not Experimental paper and just demonstrate the calculation, and so no need for details, and deferring various points to 'future work'. This does not seem to work well, raises more questions, provides unrealistic picture, and at the end fails to convince reviewers of the importance of the otherwise valid correction. If at all, a more effective demonstration would be to apply the corrections to a published work with soil profiles by others. (And in general, why focus on the 'size' of the correction rather than simply rest the case on the underlying 'correctness').

There are also some more minor issues that do not help in getting through these revisions. Just for example, the distinction between mass-dependent diffusion and mass-independent mixing is highlighted (correctly), but much effort is invested in insisting that these are more similar than not, using linguistic changes, such as changing 'mixing' to 'diffusive mixing' as a response, and using an odd definition of fractionation (which traditionally applies to reactant and product that are the same entity rather than a new one composed of the real product and any additional background). Or, the repeated cut and paste of the same response to different reviewers' comment (unpolite, to say it politely). BTW, comments on the relevance of carbonates is addressed with a single line saying CO2 interactions with carbonate is not mass-dependent, which seems odd.

I suggest to revise the paper with more attention to all comments, but consider changing the balance of the paper to further clarified theoretical part and reduced otherwise. Thank you for the feedback. We agree that by focusing on the bias associated with using our proposed convention and the traditional convention that we were taking away from the main message of the paper. We have reformatted the manuscript into a Technical Note, and have removed the field data and the assessment of "how good" the different corrections were. Instead, we focus on the math and theory, and explain how based on this math, we must use our proposed correction in the case of soil gas work.

We also apologize for the confusion in describing the differences between massdependent diffusion and mass-independent mixing in the past round of edits. We have maintained the wording of "diffusive mixing" in the text so that it is understood that in the soil gas environment, this mixing is still occurring as a result of diffusion. We have made sure to go through the text and make this clear. Referee #1

The authors made some efforts to correct the manuscript according to the reviewer's comments. However, it seems that some issues were only dealt with half-way.

For example, the text now refers to the right way to calculate the fractionation by diffusion of CO2 in air: "Binary Diffusion". But the same wrong equation still appears, and not that of binary diffusion.

For the request of soil profile description the authors added:

"There soils have sandy loam surfaces, with thin organic layers and soil layers that are poorly drained and have high clay content"

This leaves more questions than it answers. At what depth are these layers? Is there a clay layer in the middle of this profile? This has great implication for the model. And in the discussion the results are still discussed as if the soil is "clayey".

And where is the organic layer? As it is written, it appears to be not on the surface. The authors also wrote in response for details on the field study:

"We thank the reviewer for this comment and agree that we have not been detailed enough explaining both the point of including the field data, and we have not been thorough enough in our description of the field site. In this paper we did not intend to use the field data as validation for the method, but as an example of how the new theory could be applied."

This is confusing. If the field data is important, than the site and methods should be described in detail. If not, this part should be removed.

Another main issue, that was raised by the reviewers before, is that diffusion is a mass-dependent processes, so the "standard method" will correct for it. It seems that the only addition here is including the effect of mixing. I was not convinced by the authors' response on this point.

We apologize that the referee was not satisfied with our responses and edits on the last round of revisions. We have checked to make sure the proper equation for binary diffusion is now included. Based on your feedback, as well as that from the other referees and editor, we have removed the field data from the manuscript. The manuscript has now been reformatted into a Technical Note, so that the focus is on the theory and math, and not on the field data, which we were trying to use as an example of how to apply our proposed correction with real data. We have included 1 new figure (Figure 3) to illustrate how our proposed correction works (which has an added correction for mixing) in comparison to the traditional one. We hope that by removing the field data and the assessment of how well the different methods worked, that we are now clearly demonstrating how our correction must be used in soil gas environment applications. We also hope that with this new format it is clear that our new correction includes the effect of mixing. We are sorry that you were not satisfied with how this was communicated in the past.

**Referee #3**

I fully agree with the editor's assessment that the authors run the risk of producing something that is mathematically correct but not clearly useful. It is for this reason that in the earlier review I asked for sensitivity studies and assumptions more likely to be found in real soils. As the authors have not seen fit to make these changes, I think they should either (1) be required to do so or (2) be required to remove the language that I still think overstates the importance of their correction for conditions likely to be found in real soils (up to 150 per mille, etc). I leave it up to the editor to decide which of these is more appropriate.

To me there are two main issues that remain.

First, the way the authors introduce the topic still adds unnecessary confusion to the already confusing nomenclature of the 14C.

At the end of the paragraph at the top of page 2, the authors still confuse the reporting conventions of 14C data with the interpretation of those data. As the authors need a soil-specific model to interpret soil gas data – and that model also contains assumptions, I would replace the last two sentences (page 2, lines 14-16) with a statement to the effect that "The reporting conventions using Fraction Modern or D14C for radiocarbon data implicitly assume mass-dependent fractionation based on observed 13C values accurately represent the processes contributing to observed 13C and 14C in the soil atmosphere – i.e. that the 14C is fractionated twice as much as 13C by biochemical and physical processes affecting both isotopes. We argue that this is not always the case in soil pore space, where mixing of air masses mean that this assumption no longer holds." In fact, there are other ways of reporting 14C that explicitly remove the 13C correction – this is the notation that the authors use starting in equation (3) and that was originally used by Wang et al (1994). However, the do not state that this notation is part of the reporting nomenclature suggested by Stuiver and Polach.

The authors could just avoid a lot of extra confusion and state at the outset that the small delta notation is preferred for reporting soil gas data over others like fraction Modern or "Delta" in cases where factors other than mass-dependent ones likely influence radiocarbon results.

We thank the referee for these useful comments. We have gone through the text to make sure it is clear that it is not the convention itself that has been wrong, but how it was applied for this specific environment. We have made edits as you suggested on page 2, lines 14-16 (now page 2 lines 5-7). We have also gone through to make it clear that the small delta 14C reporting convention is part of the nomenclature suggested by Stuiver and Polach.

Second, the assumptions built into the model demonstrably do not hold in real soils, and what the authors give as their main example is a case that is not likely to be observed in the real world. What they have added as 'sensitivity' tests do not really do much to inform the reader in my opinion, as they do not address the sensitivity to

this fundamental problem of how they have formulated their model. The results will clearly be highly sensitive to model formulation, and it is this sensitivity that is ignored in their discussion.

1) CO2 production rates drop exponentially with depth in most soils, because roots and most additions of new plant C are at concentrated at the surface. The authors argue in their response to previous comments that they wanted an equation they could solve analytically and use that to justify continuing to use this assumption. Do they really think this is the best way to interpret their own field data, or to represent production of CO2 with depth in soils? What is the sensitivity to this assumption? Analytical solutions for the case with constant diffusivity and exponentially declining respiration with depth exist (see for example the books by Crank or Carlslaw and Jaeger on diffusion and heat fluxes; or Cook (1995) Ecological Modelling 78: 277-283, though this is for oxygen consumption instead of CO2 production). Letting production change with depth would provide a much more realistic and useful case for the authors, as well as show how sensitive their results are to this important model assumption.

2) The assumption of a constant 14C signature of respired CO2 of -200 ‰ is also something I doubt was ever observed in a real soil. As stated in my previous review, observed values are mostly closer to the atmosphere 14CO2 value because of root respiration mixing with decomposition-derived CO2 – especially at the depths where most of the CO2 in soils is produced (at the surface). At the very least, it is worth seeing how important the correction is when the difference is 40 per mille instead of 300 per mille – again what is the sensitivity to the assumption?

I would therefore ask at a minimum that the authors include a case where they assume depth dependent CO2 productions rate (using published analytical solutions) and more realistic values for the radiocarbon signature of CO2 produced. It is also important to determine how important it is to vary diffusivity as they did in the paper.

Overall, my takeaway from this paper is that the authors are mathematically correct, and this approach should definitely be used. However, they have set up an artificial system to maximize how they state its importance and use this to imply that past attempts that did not use this approach have been subject to very large errors. Without more realistic scenarios, I remain unconvinced that applying these corrections in a more realistic scenario will lead to very important changes in how existing or future soil pore space data will be interpreted.

Based on these comments, as well as those from the other referee and editor, we have reformatted the manuscript into a Technical Note, so that the focus is no longer based on the bias resulting from either of the corrections (new or old), but rather on the math/theory and how based on it, we need to also correct for mixing. In this case, our steady-state model is only used to illustrate the theory using soil gas profiles, and nothing else. We do believe that in the future work in this field should include non-steady state models and field data, especially considering the lack of information known about the radiocarbon composition of soil-respired CO2 through different depths. We hope that the

way we have reformatted the paper is a starting point and illustrates based on theory, why we must account for mixing. Based on your feedback from the last rounds of edits, the illustrative soil depth profiles were derived using a more realistic radiocarbon signature of production (- 50 permil).

**Technical Note:** Isotopic corrections for the radiocarbon composition of CO2 in the soil gas environment must account for diffusion and diffusive mixing**

Jocelyn E. Egan1, David R. Bowling2, David A. Risk3

1Department of Earth Sciences, Dalhousie University, Halifax, Nova Scotia, B3H 4R2, Canada 5 2School of Biological Sciences, University of Utah, Salt Lake City, Utah, 84112, USA 3Department of Earth Sciences, St. Francis Xavier University, Antigonish, Nova Scotia, B2G 2W5, Canada Correspondence to: Jocelyn E. Egan (jocelyn.egan@dal.ca)

**Abstract.** Earth system scientists working with radiocarbon in organic samples use a stable carbon isotope ( $\delta^{13}$ C) correction

10 to account for mass-dependent fractionation, but it has not been evaluated for the soil gas environment, where both diffusive gas transport and diffusive mixing are important. Using theory and an analytical soil gas transport model, we demonstrate that the conventional correction is inappropriate for interpreting the radio-isotopic composition of CO2 from biological production, because it does not account for important gas transport mechanisms. Based on theory used to interpret  $\delta^{13}C$  of soil production from soil CO2, see propose a new solution for radiocarbon applications in the soil gas environment that fully

accounts for both mass-dependent diffusion and mass-independent diffusive mixing. 15

**1** Introduction**

25

Radiocarbon allows us to measure the age of soil-respired CO2 (CO2 diffusing from the soil surface to the atmosphere, aka soil flux as in Cerling et al., 1991), but the traditional reporting convention for radiocarbon was not established for soil gasphase sampling, but rather for solid (organic matter) sample analysis. The validity of this convention has never been explicitly tested for soil-respired CO2.

20

The traditional radiocarbon reporting convention,  $\Delta^{14}C$  (Stuiver and Polach, 1977), uses a mass-dependent correction based on the isotopic composition of wood. Its purpose is to correct for biochemical fractionation against the radiocarbon isotopologue (14CO2) abundance during photosynthesis, which is assumed to be twice as strong as for 13CO2 based on their respective departures in molecular mass from 12CO2. The classical reference describing these conventional calculations is Stuiver and Polach (1977).

In the soil gas environment, researchers have different implicit expectations for fractionation processes. They generally assume that  $^{14}$ C of CO2 is not biochemically fractionated in the gas phase, between the points of CO2 production (biological production of CO2 by soil organisms and roots) and measurement (subsurface or flux chamber samples). This assumption is

**1**

**Jocelyn Egan 2019-7-6 3:55 PM**

Jocelyn Egan 2019-7-6 3:55 PM Deleted: Towards this end we applied an

Jocelyn Egan 2019-7-6 7:31 PM Deleted:

Jocelyn Egan 2019-7-6 3:56 PM

Deleted: across a range of soil diffusivities and biological CO2 production rates, allowing us to control the radiocarbon ( $\Delta^{14}$ C) and stable isotope  $(\delta^{13}C)$  compositions of modeled soil CO2 production and atmospheric CO2. This approach allowed us to assess the bias that results from using the conventional correction method for estimating  $\Delta^{\rm 14}C$ of soil production. We found that the conventional correction is inappropriate for interpreting the radioisotopic composition of CO2 from biological production, because it does not account for both diffusive transport and diffusive mixing. The resultant  $\Delta^{14}$ C bias associated with the traditional correction is highest in soils with low biological production and/or high soil diffusion rates.

Jocelyn Egan 2019-7-6 7:33 PM Deleted: W

celyn Egan 2019-7-6 3:23 PM

Deleted: Understanding the age of soil-respired carbon is important for process-based carbon cycle studies in the face of climate change. By knowing the age of respired carbon, we can make predictions about the rate at which carbon stocks will be transformed under current and future climates. In the case of Arctic environments underlain by permafrost, this is especially important as "ancient carbon sources previously stored may become available to decomposers as the permafrost thaws.

Jocelyn Egan 2019-7-7 3:21 PM Deleted: calculation and

reasonable based on the short residence time of  $CO_2$  (minutes to days) in the soil profile before emission to the atmosphere. However, soil gas isotopic signatures depart in predictable ways from the signature of production because of physical fractionation. It has been recognized for decades that  $\delta^{13}C$  of  $CO_2$  at any point in the soil profile will never equal the isotopic signature of production, because of transport fractionations that alter produced  $CO_2$  before it is measured (Cerling et al.,

5 1991). This theory readily extends to  $^{14}C$ .

We argue here, that in the case of soil pore space 14C where mixing of air masses occurs, that the assumption that massdependent fractionation is twice as large for 14C as  $\delta^{13}$ C by biochemical and physical processes no longer holds. Using theory and physical modelling of soil gas transport to illustrate the issue with the current reporting convention correction, we propose an alternative approach for specific use cases.

**10 2 Theory**

20

To understand why the mass-dependent correction used in the Stuiver and Polach (1977) radiocarbon reporting convention may be a poor fit for soil gas studies, we can look at our current understanding of the stable isotopic composition,  $\delta^{13}$ C, of *soil CO*2 (pore space CO2, mole fraction with respect to dry air). We use delta notation to present the stable isotopic composition of CO2:

**15 $\delta^{13}C = \left(\frac{R_s}{R_{VPDR}} - 1\right) 1000,$**

(1)

where  $\delta^{13}C$  is the isotopic composition in ‰ (see Table 1 for a full list of abbreviations),  $R_s$  is the  ${}^{13}C/{}^{12}C$  ratio of the sample, and  $R_{VPDB}$  is the  ${}^{13}C/{}^{12}C$  ratio of the international standard, Vienna Pee Dee Belemnite.

From foundational work done by Cerling (1991) we know that the isotopic composition of soil CO2 is different from that of soil-respired CO2. Any change in  $\delta^{13}$ C of soil CO2 with depth is influenced by 1) mixing of atmospheric and biological (or biogeochemical) sources of isotopically-distinct CO2, which may occur via diffusion (no bulk gas flow; referred to as

- diffusive mixing for the remainder of the paper) or advection (bulk gas flow) and 2) kinetic fractionation by diffusion. The effect of these is illustrated in Fig. 1\_using a simulated soil gas profile. In panel (a) two depth profiles of  $\delta^{13}$ C of CO2 that were modelled in a steady-state environment are shown (the model will be described in Section 3). The profiles differ only in soil diffusivity; all other characteristics were held constant, including rates of production, and  $\delta^{13}$ C of CO2 in the atmosphere
- 25 (-8 ‰; circle) and biological production (-25 ‰; square with dashed line). In the resultant depth profile with higher soil diffusivity in panel (a), the  $\delta^{13}$ C of soil CO2 ranges from -8 to -15.1 ‰. In the depth profile representing a soil with lower diffusivity, the  $\delta^{13}$ C of soil CO2 ranges from -8 to -20.6 ‰. We stress again these two isotopic depth profiles differ only due to differences in transport as a result of their varying soil diffusivities. In the depth profile with lower soil diffusivity, atmospheric CO2 does not penetrate downwards as readily, so the profile shape is much steeper near the soil-atmosphere

2

**Jocelyn Egan 2019-7-7 3:23 PM**

[revised manuscript text omitted]

(3)

(4)

 $(C_s = 14780 \text{ ppm}, \delta_s^{14} = -39.3989 \text{ }_{\infty}, C_a = 380 \text{ ppm} \text{ and } \delta_a^{14} = 45.5276 \text{ }_{\infty}; \text{ see inset box, Fig. 2(b)}).$  As was the case for  $\delta^{13}C$  in Fig. 2(a), the modelled values of  $C_s$  and  $\delta_s^{14}$  at any depth will yield the true isotopic composition of production, -50  $\infty$  (dashed line), because this approach accounts for diffusion and diffusive mixing.

5 The typical approach that has been used for interpreting the 14C composition of soil CO2 and soil-respired CO2 (e.g., Trumbore, 2000) differs from the δ14C example above, because a δ13C correction is applied to account for mass-dependent isotopic fractionation of biochemical origin, ultimately converting δ14C to a variant called Δ14C (Stuiver and Polach, 1977). The derivation of the mass-dependent correction is provided in Stuiver and Robinson (1974), where observations are normalized to an arbitrary baseline value of -25 ‰ for δ13C (a value for terrestrial wood), and the 13C fractionation factors
 10 are squared to account for the 14C/12C fractionation factor as follows:

$$\begin{split} A_{SN} &= A_{S} \left[ \frac{R_{S}(-25)}{R_{S}} \right]^{2}, \\ &= A_{S} \frac{\left[ \left( 1 - \frac{25}{1000} \right)^{*} R_{VPDB} \right]^{2}}{\left[ \left( 1 + \frac{8^{13}C}{1000} \right)^{*} R_{VPDB} \right]^{2}}, \\ &= A_{S} \frac{\left[ \left( 1 - \frac{25}{1000} \right) \right]^{2}}{\left[ \left( 1 + \frac{8^{13}C}{1000} \right)^{2}, \end{split}$$

where  $A_{SN}$  is the normalized sample activity,  $A_s$  is the sample activity, and  $\delta^{13}C$  is the isotopic composition of the sample 15 (soil CO2 in our case). As explained in Stuiver and Robinson (1974), the 0.975 term sometimes used in forms of  $A_{SN}$  is equivalent to  $\left(1 - \frac{25}{1000}\right)$ , which we will retain for clarity. The equation for  $\Delta^{14}C$ , the  $\delta^{13}C$  corrected variant of  $\delta^{14}C$ , can then be created from Eq. (5) by substituting in delta notation for  $\Delta^{14}C$  of  $\Delta^{14}C = (A_{SN}/A_{abs} - 1)*1000$  as follows Stuiver and Robinson (1974):

$$\Delta^{14}C = \left[ \left( 1 + \frac{\delta^{14}c}{1000} \right) \frac{\left( 1 - \frac{25}{1000} \right)^2}{\left( 1 + \frac{\delta^{13}c}{1000} \right)^2} - 1 \right] 1000.$$
(6)

20 Combining Eqs. (3) and (6):

$$\Delta^{14} \mathcal{C}_{old} = \left[ \left( \frac{A_S}{A_{abs}} \right) \frac{\left( 1 - \frac{25}{1000} \right)^2}{\left( 1 + \frac{\delta^{13}C}{1000} \right)^2} - 1 \right] 1000.$$
(7)

For more information on the derivation of Eqs. (6) and (7) see Stuiver and Robinson (1974), page 88. In Eq. (7) we have added the subscript "old" to highlight that this is the common approach used to date for soil gas applications - we will introduce a "new" method with Eq. (15). The terms on the left-hand side of Eqs. (6) and (7) are identical. Note that  $A_{abs}$  in

25 our notation is equivalent to  $A_0$  in Stuiver and Robinson (1974).

5

Jocelyn Egan 2019-7-7 12:14 AM Deleted: traditional Unknown Field Code Changed Jocelyn Egan 2019-7-7 3:28 PM Deleted: (Stuiver and Polach, 1977),

(5)

Jocelyn Egan 2019-7-7 12:16 AM Deleted: traditional Jocelyn Egan 2019-7-7 12:15 AM Deleted: mass-dependent correction Eq. 7 uses  $\delta^{13}C$  as an input parameter to make a mass-dependent correction to obtain  $\Delta^{14}C$ , but the profiles of  $\delta^{13}C$  and  $\delta^{14}C$  of soil CO2 (Fig. 1) highlight that both vary within the soil because of diffusion and diffusive mixing. This makes it unclear what form of  $\delta^{13}C$  should actually be used in the place of the mass-dependent correction in the soil gas environment ( $\delta^{13}C$  of the soil CO2 is measured, but  $\delta^{13}C$  of biological production is not) as diffusive mixing is not a mass-dependent process.

5 When  $\Delta^{14}C_{old}$  is modelled through depth like  $\delta^{13}C$  and  $\delta^{14}C$  in Figs. 1 and 2 it also varies with depth as shown in Fig. 2(c). However, using a  $\Delta^{14}C$  variant of Davidson's  $\delta_J$  (as for  $\delta^{14}C$  in Fig. 2(b)) at the same 40 cm depth does not correctly reproduce the specified model value for the  $\Delta^{14}C$  of production of -50 ‰, like it did for  $\delta^{13}C$  and  $\delta^{14}C$  (Cs = 14780 ppm,  $\Delta_s$  = -48.4319 ‰, Ca = 380 ppm and  $\Delta_a = 10$  ‰; see inset box, 2(c)). We therefore, adapted the mass-dependent correction in  $\Delta^{14}C_{old}$  using Davidson's (1995) theory to demonstrate how and why it should be used for  $\Delta^{14}C$  soil gas applications.

**10 3 Methods - Model Description**

We used an analytical gas transport model to simulate a range of natural soil profiles of  ${}^{12}CO_2$ ,  ${}^{13}CO_2$  and  ${}^{14}CO_2$  in order to present soil gas transport theory. The model is based on Fick's second law of diffusion:

$$\theta \frac{\partial Conc}{\partial t} = \frac{\partial}{\partial z} \left( D\left(z,t\right) \frac{\partial Conc}{\partial z} \right) + P(z,t) , \qquad (8)$$

where  $\theta$  is the soil air-filled pore space, *Conc* is the concentration, *t* is time, D(z,t) is the soil gas diffusion function, and 15 P(z,t) is the biological production function, with the latter two dependent on both depth *z* and time *t*.

The model was run in steady-state:

20

$$\frac{\partial Conc}{\partial t} = 0, \tag{9}$$

and both diffusion and production rates were constant with depth:

|   | D(z)=D,                                       | (10) |
|---|-----------------------------------------------|------|
| ) | P(z) = P.                                     | (11) |
|   | The following boundary conditions were used:  |      |
|   | $C(z=0) = Conc_{atm},$                        | (12) |
|   | $\frac{\partial c}{\partial z}\Big _{z=L}=0,$ | (13) |

where  $Conc_{atm}$  is the concentration of CO2 in air just above the soil and *L* is the model lower spatial boundary, the point below which no production or diffusion occurs. Eq. (8) is solved analytically to yield the following equation:

$$Conc(z) = \frac{P/L}{D} \left( L \times z - \frac{z^2}{z} \right) + Conc_{atm}.$$
(14)

6

Jocelyn Egan 2019-7-7 12:18 AM Deleted: This traditional approach

| Jocelyn Egan 2019-7-7 12:19 AM                                                                                                                                                                                        |  |  |  |  |  |
|-----------------------------------------------------------------------------------------------------------------------------------------------------------------------------------------------------------------------|--|--|--|--|--|
| Deleted:                                                                                                                                                                                                              |  |  |  |  |  |
| Jocelyn Egan 2019-7-6 3:35 PM                                                                                                                                                                                         |  |  |  |  |  |
| Deleted: we                                                                                                                                                                                                           |  |  |  |  |  |
| Jocelyn Egan 2019-7-7 12:19 AM                                                                                                                                                                                        |  |  |  |  |  |
| Deleted: the traditional convention                                                                                                                                                                                   |  |  |  |  |  |
| Jocelyn Egan 2019-7-6 3:50 PM                                                                                                                                                                                         |  |  |  |  |  |
| Deleted: and tested this new convention in the same way as the traditional one, by simulating soil profiles and "sampling" them to extract the isotopic composition at depth.                                  |  |  |  |  |  |
| Jocelyn Egan 2019-7-6 3:33 PM                                                                                                                                                                                         |  |  |  |  |  |
| Deleted: We then used actual field observations, where natural soil depth profiles were sampled, to test the discrepancy between old and new correction methods used to report $\Delta^{14}$ C with real data. |  |  |  |  |  |
| Jocelyn Egan 2019-7-7 12:24 AM                                                                                                                                                                                        |  |  |  |  |  |
| Deleted: 3.1 Testing the Traditional Approach                                                                                                                                                                         |  |  |  |  |  |

In the model, isotopologues of CO2 are treated as independent gases, with their own specific concentration gradients and diffusion rates (Cerling et al., 1991; Nickerson and Risk, 2009; Risk and Kellman, 2008). We assume total CO2 to be 12CO2 because of its high abundance. The error associated with this assumption is less than 0.01% (Amundson et al., 1998). Eq. (14) is thus applied for 13CO2 and 14CO2. For the full derivation see Nickerson et al. (2014) Section 2.3.

5

The analytical gas transport model was applied across a range of soil diffusivity  $(1x10^{-7}, 1x10^{-6} \text{ and } 1x10^{-5} \text{ m}^2 \text{ s}^{-1})$ , and soil production rates (0.5, 1, 2, and 4 µmol CO2 m-3 s-1). The specific soil diffusivity and production rates used to generate each profile are stated in the figure caption of that profile. We used a  $\sqrt{\delta^{13}C}$  of biological production ( $-25, \infty$ ) and atmospheric CO2 ( $\Delta_{a_s}$  - 8  $\infty$ ), and  $\sqrt{\Lambda^{14}C}$  of biological production (-50,  $\infty$ ) and atmospheric CO2 ( $\Delta_{a_s}$  - 10  $\infty$ ) to represent realistic conditions

10

20

found in nature. The other model boundary conditions were as follows: L = 0.8 m, z = 0.025 m, and  $Conc_{atm} = 15833 \text{ µmol} \text{ m}^{-3}$  (~380 ppm). The output of the model under these applied conditions were profiles of  ${}^{12}\text{CO}_2$ ,  ${}^{13}\text{CO}_2$ , and  ${}^{14}\text{CO}_2$  for each depth (z) down to the bottom boundary (L).

**4 Results**

4.1 Adapted Correction for Interpreting Radiocarbon Values of Soil and Soil-respired CO2

15 Based on Davidson's (1995) theory and what we demonstrated with Fig. 2(c), rather than using the  $\delta^{13}$ C soil pore space as a mass-dependent correction, we suggest instead using the value  $\delta^{13}_{J}$  (Eq. (2)), the biological production of  $\delta^{13}$ C, in its place in the denominator of Eq. (7) as follows:

$$\Delta^{14} C_{new} = \left[ \left( \frac{A_S}{A_{abs}} \right) \frac{\left( \frac{1 - \frac{25}{1000}}{2} \right)^2}{\left( 1 + \frac{\delta_1^{13}}{1000} \right)^2} - 1 \right] 1000.$$

Values of  $\Delta^{14}C_{new}$  through depth represent transport-fractionation-corrected soil CO2 values of radiocarbon, and in comparison to  $\Delta^{14}C_{old}$ , they are corrected for mass-independent diffusive mixing.

A depth profile of  $\Delta^{14}C_{new}$  is presented in Fig. 3 (dashed line). To generate this soil depth profile we used the numbers from the simulated profiles in Fig. 2 and inserted them into Eq. 2 to determine  $\delta_J^{13}$  at each depth. These values were then used in Eq. 15 to obtain  $\Delta^{14}C_{new}$  of soil CO2 through depth. The  $\Delta^{14}C_{new}$  profile (dashed line) is more isotopically enriched compared

25 to the  $\Delta^{14}C_{old}$  profile (solid line) in Fig. 3. Values "sampled" from the  $\Delta^{14}C_{old}$  profile (the same as the one presented in Fig. 2 (c)) were not able to reproduce the specified model value for the  $\Delta^{14}C$  of production of -50 ‰ using a  $\Delta^{14}C$  variant of Davidson's  $\delta_{J}$ . To demonstrate that  $\Delta^{14}C_{new}$  does correct for gas-transport fractionations, it can be placed into  $\Delta_{J}^{14}$ , a  $\Delta^{14}C$  adaption of Davidson (1995) for 14C (Eq. (4)) as follows:

 $\Delta_J^{14} = \frac{C_s(\Delta^{14}C_{new} - 8.8) - C_a(\Delta_a^{14} - 8.8)}{1.0088(C_s - C_a)},$

7

**Jocelyn Egan 2019-7-7 11:23 PM Deleted: (Cerling et al., 1991; Nickerson and Risk, 2009b; Risk and Kellman, 2008) Unknown Field Code Changed Unknown Field Code Changed Jocelyn Egan 2019-7-7 1:02 PM Deleted: ) Deleted: Jocelyn Egan 2019-7-7 12:21 AM Deleted: - 30 %. Jocelyn Egan 2019-7-7 12:21 AM Deleted: 0 Deleted: , -15 % Jocelyn Egan 2019-7-7 12:23 AM Deleted: . and Jocelyn Egan 2019-7-7 12:22 AM Deleted: 0 Jocelyn Egan 2019-7-7 12:22 AM Deleted: , -200 ‰, -1 ‰, 1 ‰, 200 ‰, 500 ‰). In addition it used Jocelyn Egan 2019-7-7 12:23 AM **Deleted:** $\Delta^{14}C$ Deleted: of Jocelyn Egan 2019-7-7 12:23 AM **Deleted:** and $\delta^{13}C$ of atmospheric CO2 ( $\delta_a$ ; -8 ‰), Jocelyn Egan 2019-7-7 12:23 AM Deleted: ing Jocelyn Egan 2019-7-6 3:34 PM Deleted: 3.2 Testing the Traditional and ... [3] Alternative Corrections . Jocelyn Egan 2019-7-6 12:49 AM Formatted: Heading 3 Jocelyn Egan 2019-7-7 12:26 AM**

Jocelyn Egan 2019-7-6 12:49 AM

... [4]

(15)

(16)

where  $\Delta_j^{14}$  is the  $\Delta^{14}$ C composition of soil production,  $C_s$  and  $\Delta^{14}C_{new}$  are the mole fraction and  $\Delta^{14}$ C composition of the soil CO2, and Ca and Aa14 are the mole fraction and A14C composition of CO2 in the air just above the soil.

Unlike in the case of  $\Delta^{14}C_{\text{nld}}$  demonstrated in the inset box in Fig. 2 (c), using the same 40 cm depth from the  $\Delta^{14}C_{\text{new}}$  profile, we were able to reproduce the specified model value for the  $\Delta^{14}C$  of production of -50 ‰ ( $C_s = 14780$  ppm,  $\Delta_s = -39.3989$ ‰,  $C_s = 380$  ppm and  $\Delta_a = 45.5276$  ‰; see inset box, Fig. 3).

**4,2, Workarounds and Establishing New Best Practice,**

In the soil gas environment,  $\Delta^{14}C_{new}$  convention should be used to properly interpret soil-respired CO2 from soil CO2, as it

- 10 corrects for all related transport fractionations. For researchers who have soil CO2 data previously interpreted using  $\Delta^{14}C_{olde}$  the following steps will help correct for transport fractionations: 1) use  $\delta_s^{13}$  and  $\Delta^{14}C_{old}$  to back out the activity of the sample (As); 2) calculate the isotopic composition of production for  $\delta^{13}C$  using Eq. (2),  $\delta_j^{13}$ ; 3) use  $\delta_j^{13}$  and As in Eq. (7) to calculate  $\Delta^{14}C_{news}$ , and finally 4) determine the radiocarbon isotopic composition of production, using Eq. (16),  $\Delta_j^{14}$ .
- 15 Going forward, several changes to best practice are recommended. On a lab level, for new soil CO2 data, we propose that AMS laboratories report radiocarbon using Eq. (3),  $\delta^{14}C$ , the uncorrected radiocarbon variant, so that the first step above, i.e. use  $\delta_s^{13}$  and  $\Delta^{14}C_{old}$  to back out the activity of the sample (As), can be avoided, and researchers can proceed with steps 2-4. We also suggest that researchers measure  $\delta^{13}$  alongside  $\Delta^{14}C$ , so that they are not dependent on the AMS measured  $\delta^{13}$  for potential back-corrections.

**20**

5

The Davidson (1995)  $\delta_1$  method was the gradient approach we used in our study, but alternative gradient approaches such as those presented for  $\delta^{13}$ C by Goffin et al. (2014) and Nickerson et al. (2014), and for  $\Delta^{14}$ C by Phillips et al. (2013) would likely be similarly successful in producing depth-dependent compositions of production. They are, however, not quite as straightforward as the  $\delta_1$  method.

**25 6 Conclusions**

This traditional  $\Delta^{14}$ C solution, which uses  $\delta^{13}$ C of soil CO2 as a mass-dependent correction, is not appropriate for the soil gas environment, as it does not account for mass-independent mixing processes. We propose a new best practice for  $\Delta^{14}$ C work in the soil gas environment that accounts for gas transport fractionations and produces true estimates of  $\Delta^{14}$ C of production.

**30 Author Contributions**

Jocelyn Egan 2019-7-6 12:44 AM Deleted: 4.2 Field Experiment -45 Jocelyn Egan 2019-7-7 1:08 AM Deleted: 5 Jocelyn Egan 2019-7-7 1:08 AM Deleted: 4 Jocelyn Egan 2019-7-6 12:49 AM Deleted: -Jocelyn Egan 2019-7-7 3:34 PM Deleted: the Jocelyn Egan 2019-7-7 1:43 PM Deleted: calculation

| Jocelyn Egan 2019-7-7 1:43 PM                                          |           |
|------------------------------------------------------------------------|-----------|
| Deleted: , to prevent potential error rang
1-10 ‰ (Table 3). | ging from |
| Jocelyn Egan 2019-7-7 1:44 PM                                          |           |
| Deleted:                                                               | [6]       |

**celyn Egan 2019-7-6 3:36 PM**

**Deleted:** As our fieldwork and analysis has shown, there could be error of 100 % for researchers using the traditional  $\Delta^{14}C$  reporting convention, where soil  $CO_2$  is used to interpret sources and ages of production. In cases where we are trying to predict the turnover rate and ages of sources of  $CO_2$  in future climate scenarios, an error this large is unacceptable.

Jocelyn Egan 2019-7-7 3:35 PM

JE, DB and DR conceptualized the theory and method for proving the new solution for radiocarbon applications in the soil gas environment. JE carried out the modeling, validation, visualization and writing of the original draft. DB, DR and JE reviewed and edited the draft.

**5 Acknowledgements**

Thanks to Thure Cerling for helpful discussions on the manuscript. JE is grateful for support from a Research-in-Residence Award from the Interuniversity Training in Continental-scale Ecology Project, US National Science Foundation Macrosystems Biology Program under award EF-1137336. JE was also funded by the Natural Sciences and Engineering

10 Research Council of Canada (NSERC). Additional support was provided by the US Department of Energy, Office of Science, Office of Biological and Environmental Research, Terrestrial Ecosystem Science (TES) Program under award number DE-SC0010625.

Figure 1: Modelled steady-state diffusive vertical depth profiles for  $\delta^{13}$ C and  $\delta^{14}$ C of soil CO2. In the top panel the  $\delta^{13}$ C of 5 atmospheric CO2 (circle) is -8 ‰ and CO2 from biological production (square with dashed line;  $\delta_J$ ) is -25 ‰. In the bottom panel the  $\delta^{14}$ C of atmospheric CO2 (circle) is 45.5 ‰ and CO2 from biological production (square with dashed line) is -50 ‰. Both profiles in each of the panels have the same biological production rates and isotopic composition of biological production, but each profile has a different soil diffusivity as indicated.